# Knowledge Reasoning Language Model: Unifying Knowledge and Language for Inductive Knowledge Graph Reasoning

**Xingrui Zhuo**[1,2], **Jiapu Wang**[3], **Gongqing Wu**[1,2*], **Zhongyuan Wang**[4,5], **Jichen Zhang**[6], **Shirui Pan**[7], **Xindong Wu**[1,2*]

[1]The Key Laboratory of Knowledge Engineering with Big Data (the Ministry of Education of China), Hefei University of Technology, China
[2]School of Computer Science and Information Engineering, Hefei University of Technology, China
[3]Nanjing University of Science and Technology, China
[4]China Unicom Digital Technology Co., Ltd., Beijing, China
[5]China Unicom Internet of Things Co., Ltd., Nanjing, China
[6]Shandong Inspur Science Research Institute, Jinan, China
[7]Griffith University, Australia
`zxr@mail.hfut.edu.cn,jiapu.wang@njust.edu.cn,wugq@hfut.edu.cn,`
`wangzy766@chinaunicom.cn,zhangjichen@inspur.com,`
`s.pan@griffith.edu.au,xwu@hfut.edu.cn,`

## Abstract

Inductive Knowledge Graph Reasoning (KGR) aims to discover facts in open-domain KGs containing unknown entities and relations, which poses a challenge for KGR models in comprehending uncertain KG components. Existing studies have proposed Knowledge Graph Foundation Models (KGFMs) that learn structural invariances across KGs to handle this uncertainty. Recently, Large Language Models (LLMs) have demonstrated strong capabilities for open-domain knowledge reasoning. As a result, the latest research has focused on LLM-based KGFMs that integrate LLM knowledge with KG context for inductive KGR. However, the intrinsic knowledge of LLMs may be overshadowed by sparse KG context, leading to LLM knowledge distortion, which can cause irreversible damage to model reasoning. Moreover, existing LLM-based KGR methods still struggle to fully constrain generative hallucinations in LLMs, severely limiting the credibility of reasoning results. To address these limitations, we propose a Knowledge Reasoning Language Model (KRLM) that achieves unified coordination between LLM knowledge and KG context throughout the KGR process. Specifically, we design a Knowledge Reasoning Language (KRL) instruction format and a KRL tokenizer to align LLM knowledge with KG representations. Then, we propose a KRL attention layer that coordinates intrinsic LLM knowledge with additional KG context through a dynamic knowledge memory mechanism. Finally, a structure-aware next-entity predictor is proposed, which strictly constrains the reasoning results within a trustworthy knowledge domain. Extensive experimental results on 25 real-world inductive KGR datasets demonstrate the significant superiority of the proposed KRLM[1] in both zero-shot reasoning and fine-tuning scenarios.

## 1 Introduction

Knowledge Graph Reasoning (KGR) (Ji et al., 2022; Liang et al., 2024) is dedicated to uncovering latent facts within KGs, offering interpretable evidentiary support for knowledge-driven applications (Luo et al., 2024; Ding et al., 2024; Huang et al., 2026). Traditional KGR methods (*e.g.*, rule-based (Das et al., 2018) and embedding models (Bordes et al., 2013; Sun et al., 2019; Zhuo et al., 2025b)) primarily reason facts within static closed-domain KGs, making it difficult for the

---

*Corresponding author

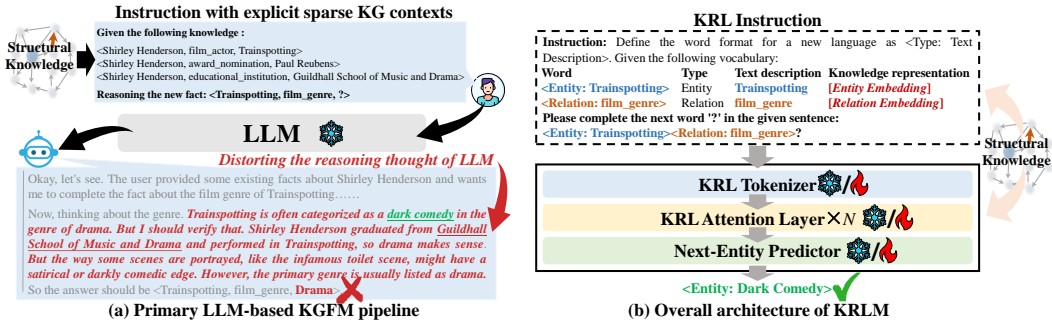

Figure 1: **(a)** Current LLM-based KGFMs overlook the necessity of establishing compatibility between sparse KG contexts and intrinsic knowledge in LLMs, which leads to knowledge distortion by LLMs. **(b)** Compared to explicit sparse KG context prompts, KRLM injects implicit knowledge representations into the reasoning instructions and LLM parameters, providing a more flexible environment for LLM to adapt to external knowledge.

model to adapt to the evolution of real-world KGs. Therefore, existing studies develop inductive KGR frameworks (Zhu et al., 2021) to reason facts with entities and relations newly added to KGs.

The core of inductive KGR is to generalize the structural characteristics of training KGs to represent unfamiliar entities and relations (Zhu et al., 2021; Teru et al., 2020). However, the inherent domain discrepancy across KGs leads to the incompatibility of structural characteristics during cross-KG deployment (Galkin et al., 2024), which limits the generalization of inductive KGR models. To cover this challenge, recent research has proposed KG Foundation Models (KGFMs) (Galkin et al., 2024; Huang et al., 2025; Zhang et al., 2024c) to capture the invariant representation of entities and relations across KGs. In general, this invariance enables any entity or relation to be represented by its relative structural context without relying on specific KG domains (Galkin et al., 2024). This property provides KGFMs with zero-shot learning capabilities, allowing them to handle open-domain KGR effectively.

Large Language Models (LLMs), pre-trained on large-scale textual corpora, have been demonstrated to achieve disruptive success on KGR (Chen et al., 2023; Wang et al., 2022; Zhang et al., 2024b; Wang et al., 2024a), which is attributed to their ability to master non-natural languages (Bolhuis et al., 2024; Han et al., 2024; Zhu et al., 2024a; Gao et al., 2024) (*e.g.*, structural knowledge-aware instructions (Kim et al., 2023; Wang et al., 2023)). Leveraging this advantage, the latest studies propose LLM-based KGFMs (Guo et al., 2024; Wang et al., 2024b) to conduct inductive KGR tasks. These methods, by utilizing the powerful context awareness and knowledge emergence (Pan et al., 2024) of LLMs, sufficiently capture implicit knowledge overlooked by primary KGFMs from structural KG context, thereby significantly improving models on open-world fact reasoning.

Previous research on LLM-based KGFMs usually explicitly recasts incomplete facts as KG context-aware instructions and conducts fact reasoning through LLM fine-tuning (Guo et al., 2024) or prompt-based reasoning (Wang et al., 2024b). Despite these accomplishments, existing LLM-based KGFMs still suffer from significant *knowledge distortion* (Li et al., 2024), *i.e.*, the sparse contextual evidence extracted from KGs may override the dense knowledge inherent in LLMs, which causes irreversible damage to LLM reasoning. This issue primarily arises from the inadequate coordination of the natural knowledge gap between KGs and LLMs, thereby hindering the generalizability of LLM-based KGFMs across diverse KGR downstream tasks.

Figure 1(a) illustrates the knowledge distortion challenge in LLM-based KGFMs. In general, current LLM-based KGFMs directly project sparse structural knowledge into a reasoning prompt, which poses a latent risk of misleading LLMs by incomplete reasoning evidence. For example, LLM incorrectly regards "*Guildhall School of Music and Drama*", the sole information related to "*film_genre*", as critical evidence. This toxic contextual association overrides the inherent knowledge of LLMs (*e.g.*, "*dark comedy*"), ultimately limiting model reasoning. In addition, although emergent knowledge endows LLMs with adaptive capacity for open-world fact reasoning, this characteristic actually

increases the risk of generating out-of-scope hallucinations (Guo et al., 2024; Pan et al., 2024). This result impacts the fairness and reliability of the model in evaluating across KGR tasks.

To address the aforementioned limitations, we propose a Knowledge Reasoning Language Model (KRLM) to alleviate the knowledge distortion by coordinating the inherent knowledge of LLMs and KGs throughout the entire KGR process. As shown in Figure 1(b), this knowledge coordination is achieved through two aspects: reasoning instruction design and model fine-tuning. Specifically, we first design a KRL-format instruction that aligns the intrinsic knowledge in LLMs (text description) with the implicit knowledge representation through a vocabulary table. Next, we construct a KRL tokenizer that converts entities and relations into unified KRL tokens, encapsulating both structural and textual knowledge. We then propose a KRL attention layer that integrates the context within KRL by coordinating the in-context learning module of a pre-trained LLM and a dynamic knowledge memory mechanism. Finally, a structure-aware next-entity predictor is proposed to tightly constrain the predicted facts to the given KG domain, ensuring the reliability and stability of the reasoning results. In addition, we adopt a collaborative training objective based on knowledge mutual distillation (Zhang et al., 2018; Hu et al., 2023) to further coordinate different knowledge.

Our main contributions can be summarized as follows:

- This paper proposes a novel Knowledge Reasoning Language Model (KRLM) for extensive KGR tasks. KRLM mitigates the knowledge distortion problem commonly faced by LLM-based KGFMs in diverse downstream KGR tasks.

- We design a unified tokenizer for various representation encapsulation in KRL, which infinite scalability of open-world entities/relations with constant-scale model parameter.

- We propose a KRL attention layer and a structure-aware next-entity predictor, which enables LLMs to effectively coordinate pre-trained intrinsic knowledge with external structural knowledge during the in-context learning process, ultimately allowing for reasoning with traceable facts.

- Extensive experimental results on 28 datasets demonstrate that the proposed method exhibits significant zero-shot learning and transfer capabilities in open-domain KGR scenarios.

## 2 RELATED WORK

In this section, we review the research roadmap of KGR, with a focus on comparing LLM-based KGR models with our proposed KRLM on open-domain KGR.

**A review of KGR**. KGR is mainly divided into transductive and inductive tasks. Traditional KGR methods (Das et al., 2018; Zhuo et al., 2024; Yang et al., 2017) are dedicated to reason latent facts in static KGs with finite sets of entity and relations. Nowadays, the dynamicity of real-world KGs have led to the proposal of inductive KGR methods for reasoning unseen entities or relations in facts. Previous inductive KGR methods (Zhu et al., 2021; Teru et al., 2020; Zhang & Yao, 2022; Galkin et al., 2022) can only generalize facts with new entities while unsuitable for unfamiliar relations. Consequently, several methods (Geng et al., 2023; Lee et al., 2023) take the relative ontological interaction of relations as a starting point to learn the structural invariance of relations in a KG, thereby improving the model's recognition of unknown relations. However, the most severe challenge faced by the featurization strategies of the above inductive KGR methods rely on specific domain features of KGs (*e.g.*, node degree or structural attribute similarity), which cannot be transferred to KGs in any domain. To address this challenge, Mikhail et al. (Galkin et al., 2024) propose an concept called "**knowledge graph foundation model**", which captures the structural invariance of entities and relations cross KGs. Inspired by this, numerous KGFMs (Huang et al., 2025; Cui et al., 2024; Zhang et al., 2024c) have been proposed in recent years, which have achieved remarkable cross domain inductive KGR through zero-shot learning.

**LLM-based KGR models**. Unlike the above KGR models that solely focuses on KG structure, LLMs can capture finer grained differences in KG context for distinguishing sub-KGs with similar structures (Zhuo et al., 2025a). Therefore, numerous studies have recently introduced LLMs to improve KGR models. For example, CSProm-KG (Chen et al., 2023) and MKGL (Guo et al., 2024) use the prefix-tuning (Li & Liang, 2021) and LoRA (Hu et al., 2022) technique, respectively, to transfer LLMs to KGR scenarios. KICGPT (Wei et al., 2023) and PROLINK (Wang et al., 2024b)

utilize a large-small model collaborative framework to integrate LLM planners and KG retrievers to achieve effective KGR. Among then, MKGL and PROLINK sufficiently the emergent knowledge capability of LLMs (Pan et al., 2024), which enables them to uncover more latent facts across open-domain KGs. This advantage makes them representative LLM-based KGFMs. However, given the natural representation gaps between the inherent knowledge of LLMs and the structural knowledge of KGs, existing LLM-based KGR methods typically face the problem of knowledge distortion, where sparse KG context used for fact reasoning may interfere with LLM reasoning, which limits the performance of LLM-based KGR models.

In contrast, the proposed KRLM comprehensively coordinates the inherent knowledge of LLMs and the implicit knowledge representation of KGs from the perspectives of instruction construction and model fine-tuning, overcoming the weakness of existing LLM-based KGFMs in unifying the internal knowledge of LLM and the external KG representation, and improving the zero-shot learning ability of LLM on cross-domain KGs during fine-tuning.

## 3 PRELIMINARIES

In this section, we introduce the background and main definitions related to this study.

**Knowledge graphs and inductive knowledge graph reasoning**. A knowledge graph is a multi-relational directed graph with entities as nodes and relations as edges. Formally, a KG can be represented as $\mathcal{G} = (\mathcal{E}, \mathcal{R}, \mathcal{T})$, where $\mathcal{E} = \{e_i\}_{i=1}^I$ and $\mathcal{R} = \{r_j\}_{j=1}^J$ denote the sets of entities and relations, respectively, and $\mathcal{T} = \{< e_h, r, e_t > | e_h, e_t \in \mathcal{E}, r \in \mathcal{R}\}$ is the set of triplets. Each triplet represents a fact composed of a head entity $e_h$, a tail entity $e_t$, and a relation $r$ that truly exists between them. Given a KG $\mathcal{G}_{train} = (\mathcal{E}_{train}, \mathcal{R}_{train}, \mathcal{T}_{train})$ for training a KGR model, inductive KGR tasks require the model to predict facts in an unobserved KG $\mathcal{G}_{test} = (\mathcal{E}_{test}, \mathcal{R}_{test}, \mathcal{T}_{test})$, where $\mathcal{E}_{test} \neq \mathcal{E}_{train}$ or $\mathcal{R}_{test} \neq \mathcal{R}_{train}$.

**Knowledge graph foundation models** learn the structural invariance from KGs, which addresses the domain shift between training and reasoning KGs in inductive KGR tasks. Typically, KGFMs employ two Graph Neural Networks ($\text{GNN}_r$ and $\text{GNN}_e$) to build KG structure learning models (Zhu et al., 2021; Teru et al., 2020). Given a query triplet $< e_h, r_q, ? > \in \mathcal{G}$, the overall framework of KGFMs can be summarized as:

$$\boldsymbol{R} = \text{GNN}_r(\{\mathbb{I}_{j=q} \cdot \mathbf{1}^d\}_{j=1}^J, \boldsymbol{R}^*, \mathcal{G}_r), \quad \boldsymbol{E} = \text{GNN}_e(\{\mathbb{I}_{i=h} \cdot \boldsymbol{r}_q\}_{i=1}^I, \boldsymbol{R}, \mathcal{G}), \tag{1}$$

where $\mathbb{I}$ is an assert function and $\mathbf{1}^d \in \mathbb{R}^d$ is the embedding of ones. KGFMs first construct a relational graph $\mathcal{G}_r = (\mathcal{R}, \mathcal{R}^*, \mathcal{T}^*)$ with $\mathcal{R}$ as a node set and $\mathcal{R}^*$ as an edge set, where $\mathcal{R}^*$ is the relative structure patterns of $\mathcal{R}$ in $\mathcal{G}$ (Galkin et al., 2024; Huang et al., 2025) and $\boldsymbol{R}^* \in \mathcal{R}^{|R^*| \times d}$ represents the type embedding of relative structural patterns. Afterwards, KGFMs use labeling tricks (Zhu et al., 2021) to obtain structurally invariant representations of all relations $\boldsymbol{R} \in \mathbb{R}^{J \times d}$. Then, driven by $\boldsymbol{r}_q \in \boldsymbol{R}$, the representation of $r_q$, KGFMs summarize the structurally invariant representations of all entities $\boldsymbol{E} \in \mathbb{R}^{I \times d}$. The detailed design of the relational graph and the KGFM architecture are provided in Appendixs C.1 and C.2, respectively.

**Knowledge reasoning language** is a new language form that contains both the inherent corpus knowledge in LLMs and the structural knowledge of KGs. As shown in Figure 2, a KRL instruction contains a global vocabulary that integrates the word-level forms, types, text descriptions, and knowledge representations of entities and relations. This intuitive contextual comparison can assist LLM understand unfamiliar elements in KRL instructions. When reasoning a fact, KRLM regards the word-level forms of entities and relations as unique tokens and adds their indices into the LLM tokenizer. Then, KRLM predicts a latent next word-level entity following the KRL instruction. Refer **Section 4** for processing details.

In addition, to alleviate the training costs may caused by the addition of word-level tokens for entities and relations, we design a low-parametric method based on Principal Attribute Aggregation (PAA), which enhances the representational completeness of word-level tokens through multi-view attribute aggregation functions (Guo et al., 2024) of pre-trained tokens, as detailed in **Section 4.1**.

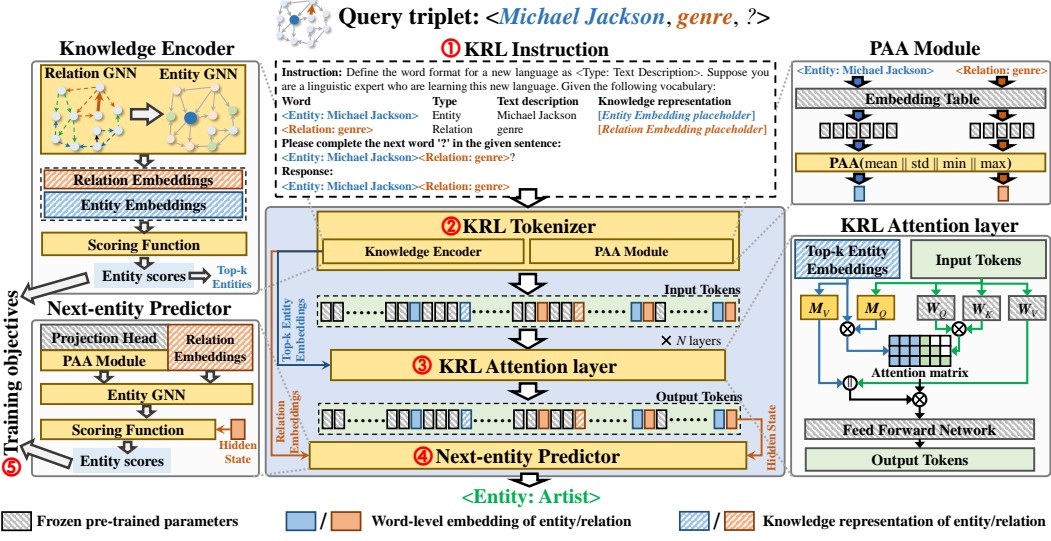

Figure 2: Overall framework of KRLM. Given a query triplet, we first convert it to ① a KRL instruction that integrates inherent knowledge of LLMs and KGs and obtain its token embedding sequence by ② a KRL tokenizer. These tokens are then input into ③ stacked KRL attention layers for capturing the in-context hidden states within KRL. Next, ④ a next-entity predictor is used to reason the entity word following KRL based on the last hidden state. ⑤ The training objective of KRLM is to coordinate the inherent knowledge of LLM with structural knowledge representation.

# 4 KNOWLEDGE REASONING LANGUAGE MODEL

In this section, we elaborate on the proposed KRLM in detail, which consists of three main components (Figure 2): a **KRL tokenizer** (**Section 4.1**) based on a knowledge encoder and a PAA module, a in-context learning module composed of stacked **KRL attention layers** (**Section 4.2**), and a GNN-based **next-entity predictor**(**Section 4.3**). In the following sections, we first provide the design of each module. Then, we illustrate the training strategy of KRLM (**Section 4.4**).

## 4.1 KRL TOKENIZER

As shown in Figure 2, a KRL instruction contains different categories of tokens. For the general tokens, we map them to the corresponding embeddings according to the pre-trained embedding table within a LLM. The word-level embeddings and knowledge representations of entities/relations in KRL are obtained by the PAA mechanism and the knowledge encoder, respectively. The detailed design specifics and in-depth discussions on KRL instructions are provided in **Appendix B**.

**The PAA mechanism** is used to obtain word-level embeddings of entities and relations. Here, we use an entity as a case to introduce the details of PAA.

Let $<$*Entity: Text description*$>$ be the word-level format of an entity, we can obtain its textual token embedding sequence $\{t_1, t_2, ..., t_L\} = \text{Emb}(\text{TKN}(< \textit{Entity: Text description} >))$, where $\text{TKN}(\cdot)$ and $\text{Emb}(\cdot)$ are the text tokenizer and token embedding table of a LLM, respectively. The PAA mechanism aggregates the different attributes of these token embeddings (*i.e.*, mean, max, min, and std attributes (Guo et al., 2024)) to obtain the word-level embedding of the entity $w_e = \text{PAA}(\{t_1, t_2, ..., t_L\})$:

$$\text{PAA}(\{t_1, t_2, ..., t_L\}) = \left[ \underset{\text{attr} \in \{\text{mean,max,min,std}\}}{||} \text{attr}(\{t_1^*, t_2^*, ..., t_L^*\}) \right] W_{\text{fusion}}, \quad (2)$$

where $||$ is a column-wise concatenation operation, $t_L \in \mathbb{R}^F$ is a $F$-dimensional token embedding in $\text{Emb}(\cdot)$, $t_L^* = t_L W_{\text{down}}$, $W_{\text{down}} \in \mathbb{R}^{F \times d}$ and $W_{\text{fusion}} \in \mathbb{R}^{4d \times d}$ are two trainable weight matrices. The PAA mechanism can construct new entity/relation word-level embeddings without restrictions

under fixed training parameters, which effectively saves memory costs and is beneficial for handling unknown entities/relations in inductive KGR tasks.

**The knowledge encoder** is a GNN-based KG structure learner that captures universal structural representations of entities and relations. Given a query triplet $< e_h, r_q, ? > \in \mathcal{G}$, we construct a knowledge encoder according to Eq. (1), where we can obtain $\boldsymbol{E}$ and $\boldsymbol{R}$, the knowledge representations of all entities and relations, respectively, based on $< e_h, r_q, ? >$. In brief, $\text{GNN}_e$ and $\text{GNN}_r$ in Eq. (1) are both designed to $S$-layer NBFNet (Zhu et al., 2021). The detailed design are provided in Appendix C.2.

In addition, to inject relevant structural context in the KRL attention layer (**Section 4.2**), we construct a MLP function $\mathcal{S}_{\text{struct}}(\cdot) : \mathbb{R}^{2d} \rightarrow \mathbb{R}^1$ to score the correlation between the structural knowledge of entity $e_i \in \mathcal{E}$ and the query triplet $< e_h, r_q, ? >$:

$$sc_{\text{struct}}^{(i)} = \mathcal{S}_{\text{struct}}([\boldsymbol{e}_i || \boldsymbol{r}_q]), \quad \boldsymbol{e}_i \in \boldsymbol{E}, \quad \boldsymbol{r}_q \in \boldsymbol{R}. \tag{3}$$

**The process of KRL tokenization** is as follows: Given an input embeddings sequence of KRL $\{\boldsymbol{w}_{e_h}, \boldsymbol{w}_{r_q}, \boldsymbol{e}_h, \boldsymbol{r}_q\} \cup \{\boldsymbol{t}_1, \boldsymbol{t}_2, ..., \boldsymbol{t}_m\}$, where $\boldsymbol{w}_{e_h}, \boldsymbol{w}_{r_q} \in \mathbb{R}^d$ are the word-level embeddings of $e_h$ and $r_q$ obtained by Eq. (2), $\boldsymbol{e}_h, \boldsymbol{r}_q \in \mathbb{R}^d$ are the knowledge representations of $e_h$ and $r_q$ obtained by Eq. (1), respectively, and $\{\boldsymbol{t}_1, \boldsymbol{t}_2, ..., \boldsymbol{t}_m\} \in \mathbb{R}^{m \times F}$ are the general text token embeddings of KRL containing the placeholders of $\{\boldsymbol{w}_{e_h}, \boldsymbol{w}_{r_q}, \boldsymbol{e}_h, \boldsymbol{r}_q\}$. We first unify $\{\boldsymbol{w}_{e_h}, \boldsymbol{w}_{r_q}, \boldsymbol{e}_h, \boldsymbol{r}_q\}$ into the dimension $F$ that can be input into LLM and replace the corresponding placeholders in $\{\boldsymbol{t}_1, \boldsymbol{t}_2, ..., \boldsymbol{t}_m\}$:

$$\widetilde{\boldsymbol{w}}_{e_h} = \mathcal{F}_{\text{word}}(\boldsymbol{w}_{e_h}), \widetilde{\boldsymbol{w}}_{r_q} = \mathcal{F}_{\text{word}}(\boldsymbol{w}_{r_q}), \widetilde{\boldsymbol{e}}_h = \mathcal{F}_{\text{struct}}(\boldsymbol{e}_h), \widetilde{\boldsymbol{r}}_q = \mathcal{F}_{\text{struct}}(\boldsymbol{r}_q)$$
$$\boldsymbol{T} = \{\boldsymbol{t}_1, ..., \boldsymbol{t}_a, \widetilde{\boldsymbol{w}}_{e_h}, \boldsymbol{t}_{a+1}, ..., \boldsymbol{t}_b, \widetilde{\boldsymbol{e}}_h, \boldsymbol{t}_{b+1}, ..., \boldsymbol{t}_c, \widetilde{\boldsymbol{r}}_q, \boldsymbol{t}_{c+1}, ..., \boldsymbol{t}_z, \widetilde{\boldsymbol{r}}_q, \boldsymbol{t}_{z+1}, ..., \widetilde{\boldsymbol{w}}_{e_h}, \widetilde{\boldsymbol{w}}_{r_q}\}, \tag{4}$$

where $\mathcal{F}_{\text{word}}(\cdot), \mathcal{F}_{\text{struct}}(\cdot) : \mathbb{R}^d \rightarrow \mathbb{R}^F$ are trainable linear layers that map word-level and knowledge embeddings of entities and relations to the LLM-dimensional space. $\boldsymbol{T} \in \mathbb{R}^{m \times F}$ are the input sequence with $m$ embeddings.

## 4.2 KRL ATTENTION LAYER

A KRL attention layer is an improvement on the standard LLM attention decoding module, which deploys a knowledge memory mechanism to dynamically coordinate the LLM intrinsic knowledge with the external KG representations in the in-context learning process. In this section, we elaborate on the LLM attention decoding layer to introduce the knowledge memory mechanism.

**The LLM attention decoding module** performs preliminary contextual learning on textual tokens, entity/relation word-level embeddings, and structural knowledge representations in KRL. To capture the multi-view context of KRL, we first obtain $\boldsymbol{T}$ by Eq. (4) and then input it into a LLM attention decoding module in the $n$-th KRL attention layer, where $n \in [1, N]$:

$$\boldsymbol{H}^{(0)} = \boldsymbol{T}, \quad \boldsymbol{H}^{(n)} = \text{softmax}(\frac{\boldsymbol{H}^{(n-1)}\boldsymbol{W}_Q^{(n)}[\boldsymbol{H}^{(n-1)}\boldsymbol{W}_K^{(n)}]^{\text{T}}}{\sqrt{F}} + \boldsymbol{W}_{\text{mask}})\boldsymbol{H}^{(n-1)}\boldsymbol{W}_V^{(n)}, \tag{5}$$

where $\boldsymbol{W}_Q^{(n)}, \boldsymbol{W}_K^{(n)}, \boldsymbol{W}_V^{(n)} \in \mathbb{R}^{F \times F}$ are frozen pre-trained weight matrices in the $n$-th layer. $\boldsymbol{W}_{\text{mask}} \in \mathbb{R}^{m \times m}$ is a casual mask matrix with a lower triangle value of 0 and the rest being $-\infty$.

**The knowledge memory mechanism** dynamically integrates structural knowledge contexts related to the query triplet into Eq. (5). Specifically, we use Eq. (3) to obtain the knowledge representations of top-$\mathcal{K}$ most relevant entity as a memory $\boldsymbol{E}_{\text{mem}} = \{\boldsymbol{e}_k | e_k \in \mathcal{E}[\text{TopK}(\{sc_{\text{struct}}^{(i)}\}_{i=1}^{I})], \boldsymbol{e}_k \in \boldsymbol{E}\} \in \mathbb{R}^{\mathcal{K} \times d}$ to guide the model learning richer KRL context, where $\text{TopK}(\cdot)$ obtains the indices of top-$\mathcal{K}$ entities and $\boldsymbol{E}$ is obtained by Eq. (1). Overall, the $n$-th KRL attention layer can be represented as:

$$\boldsymbol{H}^{(0)} = \boldsymbol{T}, \ \boldsymbol{A} = \text{softmax}(\frac{\boldsymbol{H}^{(n-1)}\boldsymbol{M}_Q^{(n)}\boldsymbol{E}_{\text{mem}}^{\text{T}} || (\boldsymbol{H}^{(n-1)}\boldsymbol{W}_Q^{(n)}[\boldsymbol{H}^{(n-1)}\boldsymbol{W}_K^{(n)}]^{\text{T}} + \boldsymbol{W}_{\text{mask}})}{\sqrt{F}}),$$
$$\boldsymbol{H}^{(n)} = \boldsymbol{A}[\boldsymbol{E}_{\text{mem}}\boldsymbol{M}_V^{(n)} || \boldsymbol{H}^{(n-1)}\boldsymbol{W}_V^{(n)}], n \in [1, N] \tag{6}$$

where $\boldsymbol{M}_Q^{(n)} \in \mathbb{R}^{F \times d}, \boldsymbol{M}_V^{(n)} \in \mathbb{R}^{d \times F}$ are trainable weight matrices in the $n$-th KRL attention layer. In specific settings, $\boldsymbol{H}^{(n)}$ needs to be further processed by a feed forward network of the corresponding layer in a LLM before it can be input into the next KRL attention layer. More discussion of the knowledge memory mechanism is attached in **Appendix D**.

### 4.3 NEXT-ENTITY PREDICTOR

In a standard LLM next-token predictor, the hidden state of the last instruction token is transformed into a probability distribution over the candidate tokens by applying a projection head $\boldsymbol{P}$. However, the inherent token vocabulary of a LLM does not completely overlap with the entity vocabulary of a KG, which can result in out-of-scope predictions and compromise the fairness of model evaluation. To address this issue, we propose a next-entity predictor that adapts the projection head $\boldsymbol{P}$ to a specific KG domain via a structural knowledge decoder. This approach constrains the reasoning results strictly within the entity vocabulary. Moreover, the knowledge decoder enables KRLM to further coordinate the inherent pre-trained knowledge in $\boldsymbol{P}$ with KG representation.

**Mapping the projection head to word-level embeddings**. We use the pre-trained projection head $\boldsymbol{P}$ in the next-token predictor of a LLM as the mapping vocabulary for the word-level embeddings of all entities. Given a word-level format *<Entity: Text description>* of an entity $e_h$, we obtain its mapping embedding $\boldsymbol{p}_h$ similar to Eq. (2):

$$\boldsymbol{p}_h = \text{PAA}(\boldsymbol{P}[\text{TKN}(< \textit{Entity: Text description} >)]), \tag{7}$$

where $\text{PAA}(\cdot)$ is a parameter-independent module that has the same structure as the one in Eq. (2).

**Knowledge decoder**. This module decodes the projection head $\boldsymbol{P}$ into the specific KG through the structural constraints of $\boldsymbol{p}_h$, avoiding the prediction of out-of-scope KG domain. In specific, we build $\text{GNN}_p$, a $S$-layer entity GNN with the same structure as $\text{GNN}_e$ Eq. (1) to achieve this goal:

$$\widetilde{\boldsymbol{P}} = \text{GNN}_p(\{\mathbb{I}_{i=h} \cdot \boldsymbol{p}_h\}_{i=1}^{I}, \boldsymbol{R}, \mathcal{G}) \tag{8}$$

where $\widetilde{\boldsymbol{P}} \in \mathbb{R}^{I \times d}$ is the decoded projection matrix. $\boldsymbol{R}$ is the knowledge representation of relations obtained by Eq. (1), which guides $\widetilde{\boldsymbol{P}}$ to perceive structural knowledge.

**Next-entity prediction**. Given word-level formats *<Entity: Text description>* and *<Relation: Text description>* of an entity $e_h$ and a relation $r_q$, respectively, we construct a MLP function $\mathcal{S}_{\text{KRLM}}(\cdot) : \mathbb{R}^{3d} \rightarrow \mathbb{R}^1$ to predict next entity scores of a KRL ending with "*<Entity: Text description><Relation: Text description>*":

$$sc_{\text{KRLM}}^{(i)} = \mathcal{S}_{\text{KRLM}}\left(\left[\widetilde{\boldsymbol{p}}_i || \boldsymbol{r}_q || g(\boldsymbol{H}^{(N)}[m])\right]\right), \tag{9}$$

where $\widetilde{\boldsymbol{p}}_i \in \widetilde{\boldsymbol{P}}$ is the projection embedding of the entity $e_i$; $\boldsymbol{r}_q \in \boldsymbol{R}$ is the knowledge embedding of $r_q$; $\boldsymbol{H}^{(N)} \in \mathbb{R}^{m \times F}$ is the result of the $N$-layer KRL attention layer (Section 4.2), where $m$ is the length of an input KRL; $\boldsymbol{H}^{(N)}[m]$ is the hidden state of the last token; and $g(\cdot) : \mathbb{R}^F \rightarrow \mathbb{R}^d$ is a linear layer.

When reasoning the next entity, we average the results of two scoring functions (Eqs. (3) and (9)) to obtain the final predicted scores of all candidate entities and regard the entity with the highest score as the predicted result.

### 4.4 TRAINING AND REASONING

Given a query triplet $q = <e_h, r_q, ?>$ with the ground truth $e_t$, the training objective is designed as:

$$
\begin{aligned}
\mathcal{L} = (1-\lambda)\underbrace{\left[-\log\left(sc_{\text{KRLM}}^{(t)}\right) + \frac{1}{|\mathcal{N}_{\text{neg}}(q)|}\sum_{e_n \in \mathcal{N}_{\text{neg}}(q)}\log\left(1 - sc_{\text{KRLM}}^{(n)}\right)\right] + \lambda\text{KL}(\mathcal{P}_{\text{struct}}||\mathcal{P}_{\text{KRLM}})}_{\text{structural distillation}} \\
+ (1-\lambda)\underbrace{\left[-\log\left(sc_{\text{struct}}^{(t)}\right) + \frac{1}{|\mathcal{N}_{\text{neg}}(q)|}\sum_{e_n \in \mathcal{N}_{\text{neg}}(q)}\log\left(1 - sc_{\text{struct}}^{(n)}\right)\right] + \lambda\text{KL}(\mathcal{P}_{\text{KRLM}}||\mathcal{P}_{\text{struct}})}_{\text{KRL distillation}},
\end{aligned}
\tag{10}
$$

where $sc_{\text{struct}}^{(t)}$ and $sc_{\text{KRLM}}^{(t)}$ are obtained by Eqs (3) and (9), respectively, $\mathcal{N}_{\text{neg}}(q)$ is a negative sample set of the query triplet $q$, $\lambda$ is a fixed weight used to balance the target loss and KL term, and $\text{KL}(\mathcal{P}||\mathcal{Q})$ is used to calculate the KL divergence between distributions $\mathcal{P}$ and $\mathcal{Q}$. $\mathcal{P}_{\text{struct}}$ and $\mathcal{P}_{\text{KRLM}}$ are two predicted score distributions of positive and negative targets.

Table 1: Average performance of each model on inductive datasets. "PT", "FT", and "E2E" mean "pre-training", "fine-tuning", and "end-to-end training from scratch" respectively. Black bold and underline indicate the best and second best results. "-" indicates that a model is not suitable for the KGR task, or the corresponding source does not have reproduction conditions.

| Inductive Datasets | | Supervised SOTA | ULTRA (PT) | ULTRA (FT) | MOTIF (PT) | MOTIF (FT) | TRIX (PT) | TRIX (FT) | MKGL | PROLINK (Llama2-7b) | KRLM (PT) | KRLM (FT) |
|---|---|---|---|---|---|---|---|---|---|---|---|---|
| IndE | Hit@10 | 0.675 | 0.703 | 0.724 | 0.721 | 0.740 | 0.732 | 0.734 | 0.726 | 0.733 | 0.738 | **0.751** |
| (12 datasets) | MRR | 0.527 | 0.549 | 0.566 | 0.557 | 0.582 | 0.579 | 0.583 | 0.578 | 0.562 | 0.583 | **0.590** |
| IndER | Hit@10 | 0.347 | 0.536 | 0.542 | 0.519 | 0.538 | 0.535 | 0.536 | - | 0.542 | 0.546 | **0.556** |
| (13 datasets) | MRR | 0.209 | 0.352 | 0.350 | 0.335 | 0.349 | 0.353 | 0.353 | - | 0.354 | 0.361 | **0.367** |

| Transductive Datasets | | ULTRA (PT) | MOTIF (PT) | TRIX (PT) | CSProm-KG (BERT) | KICGPT (GPT-3.5) | GPT-4 | KG-LLM (Llama2-7b) | MKGL | PROLINK (Llama2-7b) | KRLM (PT) | KRLM (E2E) |
|---|---|---|---|---|---|---|---|---|---|---|---|---|
| FB15k-237 | Hit@10 | 0.564 | 0.550 | 0.559 | 0.538 | 0.554 | 0.565 | - | **0.591** | - | 0.554 | 0.568 |
| | MRR | 0.368 | 0.357 | 0.366 | 0.358 | 0.412 | **0.420** | - | 0.410 | - | 0.381 | 0.394 |
| WN18RR | Hit@10 | 0.614 | 0.628 | 0.611 | **0.678** | 0.641 | - | 0.503 | 0.656 | - | 0.610 | 0.659 |
| | MRR | 0.480 | 0.529 | 0.514 | **0.575** | 0.549 | - | 0.427 | 0.552 | - | 0.506 | 0.552 |
| CoDEx-M | Hit@10 | 0.525 | 0.517 | 0.521 | - | - | - | - | - | - | 0.501 | **0.526** |
| | MRR | **0.372** | 0.361 | 0.365 | - | - | - | - | - | - | 0.349 | 0.367 |

Inspired by the mutual knowledge distillation frameworks (Zhang et al., 2018; Hu et al., 2023), Eq. (10) consists of two parts: *structural distillation* and *KRL distillation*. This approach allows KRLM to dynamically align textual context and structural knowledge in KRL during the training process, thereby promoting the coordination of different modal knowledge in KRLM. The detailed training algorithm and reasoning time complexity are provided in **Appendixes F and G**, respectively.

## 5 EXPERIMENTS

In this section, we demonstrate KRLM from the following research question: **RQ1**. Can KRLM effectively perform inductive KGR tasks on unseen KG under the zero-shot and fine-tuned conditions? **RQ2**. Does the effectiveness of each module in KRLM be confirmed, including the knowledge encoder, the PAA module, KRL attention layers, the knowledge decoder, and the training approach? **RQ3**. Is the hyperparameters set in KRLM effective?

### 5.1 DATASETS, BASELINES, AND EXPERIMENTAL SETTINGS

**Datasets**. To verify the ability of KRLM to reason facts on unseen KGs, we conduct evaluations on 28 datasets. According to the overlap level between the train KG and the test KG, these datasets can be divided into the following three categories:

- 12 **Ind**uctive **E**ntity (**IndE**) datasets from GraIL (Teru et al., 2020): FB-V1, FB-V2, FB-V3, FB-V4, NELL-V1, NELL-V2, NELL-V3, NELL-V4, WN-V1, WN-V2, WN-V3, and WN-V4.

- 13 **Ind**uctive **E**ntity and **R**elation (**IndER**) datasets from InGram (Lee et al., 2023): FB-25, FB-50, FB-75, FB-100, NL-0, NL-25, NL-50, NL-75, NL-100, WK-25, WK-50, WK-75, and WK-100.

- Three **Transductive** datasets for pre-training: FB15k-237 (Toutanova & Chen, 2015), WN18RR (Dettmers et al., 2018), CoDEx-M (Safavi & Koutra, 2020).

According to previous studies (Galkin et al., 2024), we pre-train KRLM using three transductive datasets and conduct both zero-shot and fine-tuning evaluations on IndE and IndER datasets. Detailed dataset descriptions and statistics are provided in **Appendix H**.

**Baselines**. We compare KRLM under three versions ("pre-training", "fine-tuning", and "end-to-end training from scratch") with three categories baselines that can handle inductive KGR tasks: (1) State-of-the-art supervised models reported by ULTRA (Galkin et al., 2024). We collect their detailed performance on each dataset in **Appendix H**. (2) KGFMs focusing on KG structural learning, including ULTRA (Galkin et al., 2024), MOTIF (Huang et al., 2025), and TRIX (Zhang et al., 2024c). (3) Latest LLM-based models, including MKGL (Guo et al., 2024) and PROLINK (Wang et al., 2024b). In addition, we introduce four LM-based KGR methods, CSProm-KG (Chen et al., 2023), KICGPT (Wei et al., 2023), GPT-4 (Zhu et al., 2024b), and KG-LLM (Yao et al., 2025) designed for end-to-end transductive KGR training/evaluation.

**Evaluation settings**. Based on previous work (Galkin et al., 2024), we adopt Mean Recurrent Rank (MRR) and top-10 Hit rate (Hit@10) as evaluation metrics. For each test triplet $< e_h, r_q, e_t >$, a model simultaneously predict head and tail entities, i.e. $< e_h, r_q, ? >$ and $< e_t, -r_q, ? >$, where $-r_q$ is the inverse relation of $r_q$. In the zero-shot evaluation, we use the pre-trained model with the best validation checkpoint to obtain MRR and Hit@10 on each dataset. In the fine-tuning condition, we further train the best validation checkpoint on each dataset for evaluation.

**Implementation settings**. We pre-train and fine-tune KRLM using 4 A100 (40GB) GPUs with the batch size is 4 per GPU. The total training epochs is set to 20 for pre-training. The optimizer is default to AdamW with a 5e-4 learning rate, a 1% warmup step setting and a 4-step gradient accumulation. The more detailed settings of model hyperparameters are provided in **Appendix I**.

## 5.2 MAIN RESULTS (RQ1)

In this section, we report the performance of KRLM on different KGR tasks and compare it with the SOTA baselines mentioned in Section 5.1.

**Inductive KGR tasks**. Table 1 and Figure 3 show the overall performance of KRLM on inductive datasets (the detailed experimental

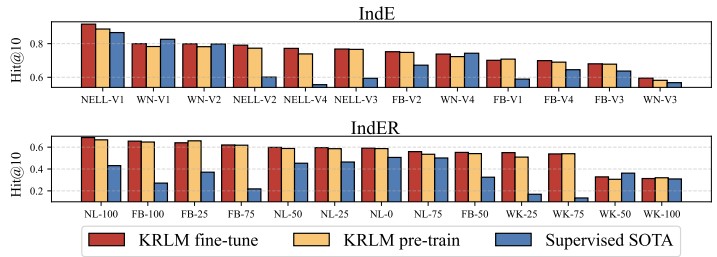

Figure 3: Comparison of our KRLM with supervised SOTA baselines on every inductive dataset.

results are provided in **Appendix J.1**). Obviously, KGFM achieves the best average performance in the fine-tuning scenario. Besides, KRLM outperforms 87% of the baselines in zero-shot scenarios and even surpasses some fine-tuned KGFMs. This success can be attributed to KRLM's ability to leverage the pre-trained intrinsic knowledge of LLMs as an extension of the invariant knowledge representation in KGFMs, which enables the model to more effectively distinguish unfamiliar entities and relations in unknown KGs. Further experimental analysis of LLM-based KGFMs reveals that MKGL fixes the number of the relation vocabulary, making it unsuitable for the IndER task and limiting its generality. In contrast, the competitive PROLINK utilizes a LLM to plan reasoning conditions and execute pre-trained ULTRA to reason facts. However, PROLINK overlooks the incompatibility between sparse KG context and LLM inherent knowledge, leading to knowledge distortion and slightly inferior performance on some datasets compared to KRLM. More detailed analysis of KRLM are attached in the Appendixes J.1 and J.7.

**Transductive KGR tasks**. The transductive KGR performance of KRLM and baselines are provided in Table 1. The results show that there is no significant positive correlation between the KGR performance of a model in the closed domain (transductive) and the open domain (inductive), which may be related to the tendency of a model to overfit during training in closed domain KGR scenarios.

Table 2: Hit@10 of each ablation variant. "E2E" means "end-to-end training". "KEn", "KMe", and "KDe" indicate the knowledge encoder, knowledge memory, and knowledge decoder in KRLM, respectively. "Atten" and "Mean" represent replacing the PAA module with attentive pooling and mean pooling, respectively. "KD" and "KL" is the KRL distillation and KL divergence part in Eq. (14), respectively.

| Datasets | KRLM (E2E) | Main Component | | | PAA Module | | Loss | | |
|---|---|---|---|---|---|---|---|---|---|
| | | -KEn | -KMe | -KDe | Atten | Mean | -KD | -KL | -KD-KL |
| **FB-V1** | **0.705** | 0.614 | 0.691 | 0.674 | 0.696 | 0.692 | 0.699 | 0.672 | 0.665 |
| **WN-V1** | **0.801** | 0.710 | 0.780 | 0.764 | 0.789 | 0.787 | 0.782 | 0.798 | 0.761 |
| **NL-0** | **0.591** | 0.537 | 0.583 | 0.570 | 0.588 | 0.584 | 0.554 | 0.533 | 0.535 |
| **NL-100** | **0.688** | 0.640 | 0.667 | 0.669 | 0.685 | 0.683 | 0.666 | 0.678 | 0.660 |

## 5.3 ABLATION EXPERIMENTS (RQ2)

This section mainly discusses the effectiveness of various modules in KRLM. The designed ablation variants and experimental results are shown in Table 2. Overall, the effectiveness of each ablation variant is inferior to that of the complete KRLM, especially in some important structural knowledge

learning modules such as "KEn", "KDe", and "KD". **Appendix J.2** provides detailed experimental settings and more results of ablation experiments.

## 5.4 PARAMETER ANALYSIS (RQ3)

This section discusses the influence of the main hyperparameters in KRLM. As shown in Figure 4, the scale $\mathcal{K}$ of knowledge memory in the KRL attention layer is set from 10 to 70. When $\mathcal{K}$ is set to 50 or above, there is no significant improvement in model. Therefore, we set $\mathcal{K} = 50$ in the experiments. In addition, to ensure the expression consistency of structured knowledge in the model, the layer numbers for the three GNNs in KRLM is uniformly set to $S$. Figure 4 demonstrates that the model is generally optimal when $S = 6$, and too few or too many layers may lead to underfitting or oversmoothing of the GNN model. The detailed parameter analysis of $\lambda$ in Eq. (10) is attached in **Appendix J.3**.

## 6 CONCLUSION

This paper first discusses the knowledge distortion challenge faced by LLM-based KGFMs in inductive KGR tasks, *i.e.*, these models are difficult to co-ordinate internal knowledge of LLMs and external KG context, where sparse KG context may override LLM's internal knowledge, thereby seriously damaging the credibility of reasoning

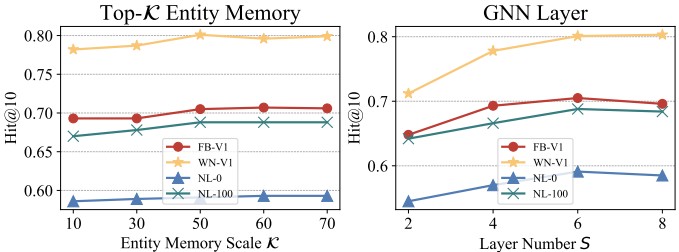

Figure 4: Performance of KRLM with different hyperparameters.

results. Based on this, we propose a novel Knowledge Reasoning Language Model (KRLM), which comprehensively enhances the inherent knowledge collaboration between LLMs and KGs from four aspects: fine-tuning instruction construction, in-context learning, next-token prediction, and model training. Extensive experiments confirm the superiority of KRLM in terms of both end-to-end fine-tuning and zero-shot transfer scenarios. **Appendix K** provides the limitations of KRLM and possible future expansion directions.

## 7 ETHICS STATEMENT

We confirm that our work has been conducted in accordance with the ICLR Code of Ethics (https://iclr.cc/public/CodeOfEthics). The study does not involve human subjects, sensitive personal data, or experiments that may cause harm to individuals or groups. The datasets used are publicly available and no personally identifiable information is included. Our methodology and findings are intended for academic purposes and do not pose foreseeable risks of misuse. We have carefully considered issues of fairness, bias, and privacy, and to the best of our knowledge, our research maintains integrity and complies with all applicable ethical standards.

## 8 REPRODUCIBILITY STATEMENT

We confirm that our study has reproducibility. Specifically, we have first submitted our desensitized project on anonymous GitHub (https://anonymous.4open.science/r/KRLM-EA36). The detailed pseudocode of the algorithm is provided in **Appendix F**. In addition, we provide specific details of the experimental conclusions in the main text, including dataset partitioning (**Appendix H**), hyperparameter settings (**Appendix I**), and ablation variant settings (**Appendix J.2**).

## ACKNOWLEDGMENTS

This work was supported by the National Natural Science Foundation of China under Grants 62120106008 and 62472136; the Program for Innovative Research Team at the University of the Ministry of Education under Grant IRT_17R32; Anhui Provincial Science and Technology Fortification Plan, China, under Grant 202423k09020015; and Hefei Key Technology R&D Champion-Based Selection Project under Grant 2024SGJ010.

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

## A    THE USE OF LARGE LANGUAGE MODELS

We used a large language model (LLM) solely as an editing assistant to improve the grammar, clarity, and concision of the manuscript. All technical contributions, experimental design, data processing, evaluation, and conclusions reported in the paper were authored and verified by the human authors. LLM-suggested edits were reviewed and accepted or modified by the authors; no numerical results, figures, or analyses were generated or approved solely by the LLM.

## B    DESIGN DETAILS OF KRL INSTRUCTIONS

Given a query triplet $(h, r, ?)$, we first provide its schema of a KRL instruction below:

---

**Schema of the KRL Instruction**

**Instruction:** Define the word format for a new language as <Type: Text Description>. Suppose you are a linguistic expert who are learning this new language. Given the following vocabulary:

**Word|Type|Text description|Knowledge representation**
*<Entity: name of $h$>*|Entity|description of $h$|[KG embedding of $h$]
*<Relation: name of $r$>*|Relation|description of $r$|[KG embedding of $r$]

Please complete the next word '?' in the given sentence:
*<Entity: name of $h$><Relation: name of $r$>*?

**Response:***<Entity: name of $h$><Relation: name of $r$>*

---

A KRL instruction consists of three types of tokens: word-level embeddings, KG embeddings, and LLM-pretrained tokens.

- **Word-level embeddings** refer to the principal attribute aggregation result of the text strings of entities and relations after looking up the LLM pretrained token table (refer to Eq. (2)). Given an entity $h$ expressed as the string "<Entity: name of $h$>", we feed this string into the LLM's tokenizer to obtain an embedding sequence $[\mathbf{t}_1, \mathbf{t}_2, ..., \mathbf{t}_n] \in \mathbb{R}^{n \times F}$. We then apply four pooling operations, mean, std, max, and min, to obtain $\mathbf{t}_{mean}, \mathbf{t}_{std}, \mathbf{t}_{max}, \mathbf{t}_{min} \in \mathbb{R}^F$. A trainable MLP layer encodes the concatenation of these pooled vectors into a representation of dimension $F$, which serves as the word-level embedding of $h$, denoted as *<Entity: name of $h$>*. This design avoids expanding the LLM's pretrained embedding table to accommodate new entities, thereby improving the model's scalability and generalization ability.

- **The KG embeddings** are obtained from a GNN-based KG reasoning model. To enable zero-shot generalization on unseen KGs, we adopt ULTRA (Galkin et al., 2024), a GNN-based KG foundation model, to produce structural embeddings for entities and relations. These embeddings are then projected through a trainable MLP layer to match the LLM hidden dimension $F$ and injected into the KRL instruction as [KG embedding of $h$] and [KG embedding of $r$].

- **LLM pretrained tokens**. These are standard tokens in the KRL instruction that fall outside the above two categories and are directly provided by the pretrained LLM.

Because of the vocabulary table mapping the word-level tokens, KG embeddings, and LLM pretrained tokens, the KRL instruction is more concise explicit KG-context prompts, resulting in an average length of only 118.75±5.14 in the 28 KG datasets.

## C    MODELING DETAILS OF KGFMS

### C.1    RELATIONAL GRAPH CONSTRUCTION

Unlike a typical KG $\mathcal{G} = (\mathcal{E}, \mathcal{R}, \mathcal{T})$, a relational graph is used to describe the relative states between relations. According to the design of ULTRA (Galkin et al., 2024), the relative state of relations

in a relational graph is related to the entity attributes they share. For example, given two triplets $< h_1, r_1, t_1 >$ and $< e_2, r_2, t_1 >$, $r_1$ and $r_2$ share the same tail entity $t_1$, so the relative state from $r_1$ to $r_2$ is "*tail-to-tail*" (*t2t*). According to this setting, we can map $\mathcal{G}$ into four relational sub-graphs that only contain a single relative state: $\mathcal{G}_{h2t} = (\mathcal{R}, \{r^*_{h2t}\}, \mathcal{T}^*_{h2t})$, $\mathcal{G}_{h2h} = (\mathcal{R}, \{r^*_{h2h}\}, \mathcal{T}^*_{h2h})$, $\mathcal{G}_{t2h} = (\mathcal{R}, \{r^*_{t2h}\}, \mathcal{T}^*_{t2h})$, and $\mathcal{G}_{t2t} = (\mathcal{R}, \{r^*_{t2t}\}, \mathcal{T}^*_{t2t})$, where $r^*_{h2t}$, $r^*_{h2h}$, $r^*_{t2h}$, and $r^*_{t2t}$ indicate four relative states "*head-to-tail*", "*head-to-head*", "*tail-to-head*", and "*tail-to-tail*", respectively.

Finally, we can obtain the relational graph $\mathcal{G}_r = (\mathcal{R}, \mathcal{R}^*, \mathcal{T}^*)$ in Eq. (1) by integrating $\mathcal{G}_{h2t}$, $\mathcal{G}_{h2h}$, $\mathcal{G}_{t2h}$, and $\mathcal{G}_{t2t}$, where $\mathcal{R}^* = \{r^*_{h2t}, r^*_{h2h}, r^*_{t2h}, r^*_{t2t}\}$ and $\mathcal{T}^* = \mathcal{T}^*_{h2t} \cup \mathcal{T}^*_{h2h} \cup \mathcal{T}^*_{t2h} \cup \mathcal{T}^*_{t2t}$.

## C.2 KGFM ARCHITECTURE

As shown in Eq. (1), KGFM contain two structure learning modules ($\text{GNN}_e$ and $\text{GNN}_r$) for entities and relations. Given a query triplet $< e_h, r_q, ? > \in \mathcal{G}$ and $\boldsymbol{r}_j^{(0)} = \mathbb{I}_{j=q} \cdot \mathbf{1}^d$, we first design a $S$-layer GNN model $\text{GNN}_r$ for learning the invariance of the relational structure according to Eq. (1):

$$\boldsymbol{r}_q^{(s)} = \sigma(\text{Update}([\boldsymbol{r}_q^{(s-1)} || \text{Agg}(\text{Mess}(\boldsymbol{r}_j^{(s-1)}, \boldsymbol{r}^*) | r_j \in \mathcal{N}_{\mathcal{G}_r}(r_q), \boldsymbol{r}^* \in \boldsymbol{R}^*)])), \quad s \in [1, S], \quad (11)$$

where $\text{Mess}(\cdot)$ is a non-parametric DistMult message function (Yang et al., 2015), $\text{Agg}(\cdot)$ represents the sum aggregation operation, $\text{Update}(\cdot) : \mathbb{R}^{2d} \to \mathbb{R}^d$ is a trainable linear layer, and $\sigma(\cdot)$ is a ReLU activation function. $\mathcal{G}_r$ is a relational graph defined in Eq. (1). The edges in $\mathcal{G}_r$ are directed as "*head-to-tail*", "*tail-to-head*", "*head-to-head*", and "*tail-to-tail*" based on the shared entities (either the head entity or tail entity) between the two relations in $\mathcal{G}$ (Galkin et al., 2024) (The detailed design are provided in Appendix C.1). Therefore, the edge embeddings are set to a trainable matrix $\boldsymbol{R}^* \in \mathbb{R}^{4 \times d}$ to model the relative structures between two relations.

According to Eq. (11), we obtain the knowledge representation of relations $\boldsymbol{R} = \{\boldsymbol{r}_j^{(S)}\}_{j=1}^J$. Similarly, let $\boldsymbol{e}_i^{(0)} = \mathbb{I}_{i=h} \cdot \boldsymbol{R}[q]$, we construct a $S$-layer GNN model $\text{GNN}_e$ for entity structure learning:

$$\boldsymbol{e}_h^{(s)} = \sigma(\text{Update}([\boldsymbol{e}_h^{(s-1)} || \text{Agg}(\text{Mess}(\boldsymbol{e}_i^{(s-1)}, f^{(s)}(\boldsymbol{r})) | e_i \in \mathcal{N}_{\mathcal{G}}(e_h), \boldsymbol{r} \in \boldsymbol{R})])), \ s \in [1, S], \quad (12)$$

where $f^{(s)} : \mathbb{R}^d \to \mathbb{R}^d$ is a non-linear function composed of a two-layer MLP with a relu function, which can transform the structural embeddings of relations into representations that adapt to the learning of entity structures in each layer of $\text{GNN}_e$. Finally, we obtain the knowledge representation of entities $\boldsymbol{E} = \{\boldsymbol{e}_i^{(S)}\}_{i=1}^I$ by Eq. (12).

## D DISCUSSION OF THE KRL ATTENTION LAYER

This section elaborates on the effectiveness of the KRL attention mechanism from the perspective of the last token in the KRL instruction. Overall, we hope that the hidden state of the last token in KRL can simultaneously contain textual and structural knowledge contexts in KRL, which provide a prerequisite for subsequent next-entity prediction.

Let the hidden state sequence of tokens obtained by the $n$-1 th KRL attention layer is $\boldsymbol{H}^{(n-1)} = \{\boldsymbol{h}_1^{(n-1)}, \boldsymbol{h}_2^{(n-1)}, ..., \boldsymbol{h}_m^{(n-1)}\}$. According to Eq. (5), without introducing the dynamic knowledge memory, the hidden state of the last token obtained by the $n$-th KRL attention layer is:

$$\boldsymbol{h}_m^{(n)} = \sum_{i=1}^m \alpha_i \boldsymbol{h}_i^{(n-1)} \boldsymbol{W}_V^{(n)}, \quad \alpha_i = \frac{\exp\left(< \boldsymbol{h}_m^{(n-1)} \boldsymbol{W}_Q^{(n)}; \boldsymbol{h}_i^{(n-1)} \boldsymbol{W}_K^{(n)} >\right)}{\sqrt{F} \sum_{j=1}^m \exp\left(< \boldsymbol{h}_m^{(n-1)} \boldsymbol{W}_Q^{(n)}; \boldsymbol{h}_j^{(n-1)} \boldsymbol{W}_K^{(n)} >\right)}, \quad (13)$$

where $< \cdot; \cdot >$ is an inner product operation. Eq. (13) can be seen as in-context learning of tokens within a KRL instruction (including textual tokens and structural knowledge representations), where $\alpha_i$ represents the scaling degree of contextual semantics for the last token.

However, the independent structural knowledge representation of the entity and relation in a KRL instruction is too thin compared to the widely existing textual tokens, which can easily cause the model to undervalue critical KG context when learning KRL instructions. To address this issue, we propose a dynamic knowledge memory mechanism that injects extra KG structural context related to the entity and relation in KRL into the in-context learning process in a KRL attention layer. Let

$\{\boldsymbol{e}_k\}_{k=1}^{\mathcal{K}}$ be a knowledge memory containing top-$\mathcal{K}$ entity embeddings obtained by Eqs. (1) and (3). According to Eq. (6), we can reconstruct Eq. (13) into Eq. (14):

$$\boldsymbol{h}_m^{(n)} = \sum_{i=1}^{m} \alpha_i \boldsymbol{h}_i^{(n-1)} \boldsymbol{W}_V^{(n)} + \sum_{k=1}^{\mathcal{K}} \beta_k \boldsymbol{e}_k \boldsymbol{M}_V^{(n)},$$

$$\alpha_i = \frac{\exp\left(< \boldsymbol{h}_m^{(n-1)} \boldsymbol{W}_Q^{(n)}; \boldsymbol{h}_i^{(n-1)} \boldsymbol{W}_K^{(n)} >\right)}{\sqrt{F}[\sum_{j=1}^{m} \exp\left(< \boldsymbol{h}_m^{(n-1)} \boldsymbol{W}_Q^{(n)}; \boldsymbol{h}_j^{(n-1)} \boldsymbol{W}_K^{(n)} >\right) + \sum_{k=1}^{\mathcal{K}} \exp\left(< \boldsymbol{h}_m^{(n-1)} \boldsymbol{M}_Q^{(n)}; \boldsymbol{e}_k >\right)]}, \quad (14)$$

$$\beta_k = \frac{\exp\left(< \boldsymbol{h}_m^{(n-1)} \boldsymbol{M}_Q^{(n)}; \boldsymbol{e}_k >\right)}{\sqrt{F}[\sum_{j=1}^{m} \exp\left(< \boldsymbol{h}_m^{(n-1)} \boldsymbol{W}_Q^{(n)}; \boldsymbol{h}_j^{(n-1)} \boldsymbol{W}_K^{(n)} >\right) + \sum_{z=1}^{\mathcal{K}} \exp\left(< \boldsymbol{h}_m^{(n-1)} \boldsymbol{M}_Q^{(n)}; \boldsymbol{e}_z >\right)]}.$$

By utilizing additional KG context, Eq. (14) coordinates the influence of LLM internal knowledge and external KG context on $\boldsymbol{h}_m^{(n)}$ through semantic space scaling and translation. In specific, Eq. (14) utilizes the knowledge memory to scale the contextual importance coefficient $\alpha_i$ of each token in KRL, which alleviates the contextual impact of large-scale textual tokens on rare entity/relation structural representations in KRL. In addition, the knowledge memory contributes an effective semantic translation as an independent parameter term $\sum_{k=1}^{\mathcal{K}} \beta_k \boldsymbol{e}_k \boldsymbol{M}_V^{(n)}$, which enhances the perception of structural knowledge context by $\boldsymbol{h}_m^{(n)}$ and thus assists in subsequent next-entity prediction.

## E DISCUSSION OF THE NEXT-ENTITY PREDICTOR

The next-entity predictor uses the hidden state of the last token (<*Relation: name of $r$*>) in the KRL instruction to predict the word-level token (<*Entity: name of $h$*>) of the target entity. This converts KG reasoning into an LLM-style next-token prediction, *i.e.*, next-entity prediction. This design avoids the risk of generating out-of-scope entities commonly observed in existing LLM-based KGR models. The structural constraints of our approach are reflected in two aspects:

**(I) Entity-space constraint:** Most prior LLM-based KGR methods inherit the LLM's next-token prediction mechanism, generating entities as sequences of vocabulary tokens. Since the LLM vocabulary (e.g., Llama2-7B has 32k tokens) is typically much larger than the number of entities in a KG benchmark and an entity name may require multiple tokens, LLMs may generate the textual name of an entity that falls outside the gold entity set. (This does not necessarily mean the generated entity is factually wrong, but it makes evaluation unfair.)

To address this, KRLM aggregates the MLP head $\mathbf{P} \in \mathbb{R}^{4096 \times 32000}$ in the next-token predictor of Llama2-7b into a compressed one $\mathbf{P} \in \mathbb{R}^{4096 \times |\mathcal{E}|}$ whose size matches the KG's entity set $\mathcal{E}$. The hidden state of the last KRL token is then compared with this compressed MLP head to select the top-1 target entity. This guarantees that predictions always lie within the entity set and therefore remain evaluable.

**(II) Structural context constraint:** Under the **entity-space constraint**, the compressed MLP head stores each target entity's word-level embedding, allowing basic in-domain entity prediction. However, we further want the MLP head to incorporate the KG structural context of a given query triplet $(h, r, ?)$ to assist in model prediction.

Consequently, as described in Eq. (8) in our paper, we feed the word-level embedding of the head entity $h$ into NBFNet (Zhu et al., 2021), a GNN-based KG encoder, to propagate messages over the KG and obtain contextual embeddings for all entities. These embeddings form an $h$-specific MLP head, which is then used for predicting the target entity. To verify its effectiveness, we include the "-KDe" ablation in Table 2 in our paper, which demonstrates that adding **structural-context constraint** significantly outperforms using only **entity-space constraint**.

## F TRAINING ALGORITHM

Algorithm 1 provides a complete pre-training process for KRLM. In each training round, the head entity $e_h$ and relation $r_q$ in a query triplet are firstly transformed into structural knowledge represen-

---

**Algorithm 1** Pre-training framework of KRLM

---

**Input:** Query triplet set $\mathcal{T}_q$; KG $\mathcal{G}$; relational graph $\mathcal{G}_r$; trainable model parameters $\Theta$; learning rate $\eta$; max training step $s$; batch size $b$.
**Output:** Optimized parameters $\Theta$.

1: $step = 0$
2: **for** $step < s$ **do**
3:     Obtain $\mathcal{T}_q^* \subseteq \mathcal{T}_q$ that contains $b$ randomly selected query triplets
4:     $\mathcal{L}_{total} = 0$
5:     **for** $< e_h, r_q, ? >$ in $\mathcal{T}_q^*$ **do**
6:         Obtain $e_h, r_q$ according to Eq. (1) and obtain $w_{e_h}, w_{r_q}$ according to Eq. (2)
7:         Construct the KRL token embedding sequence $T$ by Eq. (4)
8:         Select top-$\mathcal{K}$ entity embedding related to $< e_h, r_q, ? >$ by Eq. (3)
9:         Obtain $H^{(N)}$ by Eq. (6) and extract the hidden state $H^{(N)}[m]$ of the last KRL token
10:        Mapping the projection head in LLM to the KG domain by Eqs. (7) and (8)
11:        Obtain the predicted entity score according to Eq. (9)
12:        Calculate the loss $\mathcal{L}$ using Eq. (10)
13:        $\mathcal{L}_{total} \leftarrow \mathcal{L}_{total} + \mathcal{L}$
14:     **end for**
15:     Optimize $\Theta$ using $\mathcal{L}_{total}$ with the Adam gradient descent method
16:     $step \leftarrow step + 1$
17: **end for**
18: **return** $\Theta$

---

Table 3: Comparison of training costs between KRLM and MKGL.

| Model
(Llama2-7b as backbone) | Trainable parameters | Training time per epoch |
|---|---|---|
| MKGL | 18 M (16.78 M for LoRA) | 1 h 8 min / 4 X A100 GPU |
| KRLM (Ours) | 18.49 M
(16.78 M for the KRL attention layer) | 1 h 20 min / 4 X A100 GPU |

tations ($e_h$ and $r_q$) and word-level embeddings ($w_{e_h}$ and $w_{r_q}$) using Eqs. (1) and (2), respectively, and ultimately integrated into a KRL instruction (Steps 6-7). Next, we select top-$\mathcal{K}$ entities related to the query triplet (Step 8) and input them together with the KRL instruction into the stacked KRL attention layers for in-context learning. Then, we extract the hidden state of the last KRL token and calculate the predicted score of the next entity of the KRL instruction (Steps 9-11). Finally, the training loss is calculated according the predicted scores, which is used to optimize the trainable parameters in KRLM.

# G COMPUTATIONAL COMPLEXITY

## G.1 TRAINING COST

We calculated the trainable parameters of MKGL and our KRLM, as well as the training time on the FB15k237 dataset with a uniform batch of 4 per GPU. The statistical results are shown in Table 3.

KRLM requires embedding GNN in the tokenizer and next-token predictor of LLM, which slightly increases the parameters. However, it is consistent with MKGL in the main fine-tuning parameters of LLM (KRL attention layer V.S. LoRA). To ensure generalization, KRLM requires additional cost to construct a relational graph for real-time perturbed KGs in each batch, resulting in a training time of about 12 minutes longer per epoch than MKGL.

Although KRLM incurs additional training costs, it offers substantially stronger generalization compared to MKGL. Specifically, KRLM requires only a single pre-training phase on a large-scale KG, after which it can perform training-free zero-shot reasoning on entirely new KGs (refer to KRLM (PT) in Tables 1, 12, and 13 in our submitted paper). In contrast, MKGL is not a fully generalizable KGFM in the strict sense. While it can effectively recognize unseen entities within each inductive

Table 4: TFLOPs, memory footprint, and wall-clock time of KRLM for pre-training and fine-tuning.

| Metrics | Pre-training (3 transductive dataset) | Fine-tuning (FB V1) | Fine-tuning (FB25) |
|---|---|---|---|
| TFLOPs of forward propagation | 3.3436±0.4540 | 3.2755±0.5208 | 3.3312±0.4859 |
| Training Memory footprint | 36.12 GB | 32.57 GB | 32.67 GB |
| Wall-clock time | 3h10m per epoch×20 epochs | 7m28s per epoch×3 epochs | 12m13s per epoch×10 epochs |

dataset, it cannot transfer zero-shot across different inductive datasets. Consequently, MKGL must be retrained for every new dataset, which significantly increases its deployment overhead.

Table 4 shows the TFLOPs, memory footprint, and wall-clock time of KRLM for pre-training and fine-tuning under the condition of batch_size = 4 per GPU × 4 GPUs.

During training, there is natural step-to-step variability in both the number of input tokens. To obtain a stable and representative estimate, we compute the average forward TFLOPs over 100 steps. (Backward propagation and optimizer updates theoretically introduce an additional 2-3 × TFLOPs). For fine-tuning efficiency, we further include results on the largest inductive dataset (FB25) and the smallest inductive dataset (FB-V1) to provide an upper-lower bound range of computational overhead.

### G.2   INFERENCE COMPLEXITY

The inference complexity of KRLM can be analyzed from two parts. From the perspective of the knowledge encoder and decoder, the time complexity is upper-bounded by the entity GNN ($GNN_e(\cdot)$ and $GNN_p(\cdot)$), as the number of nodes $|\mathcal{R}|$ involved in $GNN_r(\cdot)$ is much smaller than the number of KG entities $|\mathcal{E}|$ that $GNN_e(\cdot)$ and $GNN_p(\cdot)$ need to handle. For an entity GNN, the reasoning time complexity of each layer is usually linearly related to the number of edges (triplets) (Galkin et al., 2024; Zhu et al., 2021) $O(|\mathcal{T}|d + |\mathcal{E}|d^2)$. Therefore, for a $S$-layer entity GNN, its overall time complexity is $O(S(|\mathcal{T}|d + |\mathcal{E}|d^2))$. Furthermore, thanks to the efficient relational messaging kernel implemented by the Pytorch-geometric library, the complexity of an entity GNN is optimized to $O(S|\mathcal{E}|d)$ that is linear with the number of nodes, which has been applied to the related ULTRA-like KGFMs (Galkin et al., 2024; Huang et al., 2025; Zhang et al., 2024c).

the reasoning time complexity in LLM is concentrated in the KRL attention layer. Set the token length of a KRL instruction and the scale of the knowledge memory to be $m$ and $\mathcal{K}$, respectively, the reasoning time complexity in KRL attention layer can be divided into the self-attention matrix calculation in LLM attention decodeing module ($O(m^2F)$) and the knowledge memory ($O(m\mathcal{K}d)$), and the final attentive pooling operation ($O(m(m+\mathcal{K})F)$), where $F$ and $d$ are the hidden dimensions of LLM and $GNN_e(\cdot)$, respectively. Because $m \gg \mathcal{K}$, the total complexity of a $N$-layer KRL attention module can be represented as $O(Nm(m + \mathcal{K})F)$.

To visually demonstrate the inference latency of KRLM, we selected two datasets with the highest (FB15k237) and lowest (NELL-V1) graph densities within our experimental scope as benchmarks and included MKGL and PROLINK, the latest LLM-based KGFMs, as comparative baselines. Table 5 reports the inference time of both LLM-based (KRLM, MKGL, PROLINK) and ULTRA-like (ULTRA, MOTIF, TRXI) KGFMs. For consistency, we set the test batch size to 16 and used Llama-2-7b as the backbone for all LLM-based KGFMs, conducting experiments on a single NVIDIA A100 GPU.

ULTRA-like KGFMs require loading the entire KG as the source for inference, while LLM-based KGFMs follow the ULTRA+LLM hybrid framework. Consequently, all publicly accessible KGFMs we used are inevitably affected by the scale of the underlying KG. In addition, since the original PROLINK paper does not release data-processing scripts for FB15k237, we only counted its inference time on NELL-V1.

Table 5 reports the detailed inference costs, including inference time (seconds per batch) and GPU memory consumption. Existing KGFMs exhibit sensitivity to the KG size. For KRLM and MKGL, their inference time differs by approximately one second between FB15k237 and NELL-V1, which are acceptable to humans. However, PROLINK needs a long prompt to guide Llama2-7b to generate the potential target entity types of a query according to the relational context, which leads to it needing to spend a longer inference time and larger memory on small-scale NELL-V1.

Table 5: Inference time (second) and GPU memory consumption of KRLM and baselines.

| Dataset | KRLM (Ours) | MKGL | PROLINK | ULTRA | MOTIF | TRIX |
|---------|-------------|------|---------|-------|-------|------|
| FB15k237 | 2.23±0.03 [30.11GB] | 1.98±0.04 [27.75GB] | - | 0.14±0.01 [2.6GB] | 0.25±0.01 [2.63GB] | 0.22±0.01 [2.6GB] |
| NELL-V1 | 1.18±0.07 [29.32GB] | 0.99±0.06 [26.82GB] | 4.35±0.04 [36.42GB] | 0.01±0.00 [2.5GB] | 0.02±0.00 [2.5GB] | 0.01±0.00 [2.5GB] |

Table 6: Inference time (second) of each in KRLM.

| Dataset | KRL tokenizer | KRL attention layers | Knowledge decoder |
|---------|---------------|----------------------|-------------------|
| FB15k237 | 1.2413±0.0179 | 1.1206±0.2408 | 0.0961±0.0200 |
| NELL-V1 | 0.0293±0.0019 | 1.0554±0.0535 | 0.0862±0.0047 |

In addition, we have also counted the inference time of each component in KRLM Table 6. We found that the main module that affects the inference latency of KRLM on different scales of KGs is the KRL tokenizer, because it contains an ULTRA module, which needs to read the complete KG for structural context learning of entities and relations.

## H DATASETS

To verify the ability of KRLM to reason facts on unseen KGs, we conduct evaluations on 28 datasets. According to the overlap level between train KG $\mathcal{G}_{train} = (\mathcal{E}_{train}, \mathcal{R}_{train}, \mathcal{T}_{train})$ and test KG $\mathcal{G}_{test} = (\mathcal{E}_{test}, \mathcal{R}_{test}, \mathcal{T}_{test})$, these datasets can be divided into the following three categories:

- **Ind**uctive **E**ntity (**IndE**) datasets that $\mathcal{E}_{test} \neq \mathcal{E}_{train}$ and $\mathcal{R}_{test} = \mathcal{R}_{train}$, including 12 datasets from GraIL (Teru et al., 2020): FB-V1, FB-V2, FB-V3, FB-V4, NELL-V1, NELL-V2, NELL-V3, NELL-V4, WN-V1, WN-V2, WN-V3, and WN-V4.

- **Ind**uctive **E**ntity and **R**elation (**IndER**) datasets that $\mathcal{E}_{test} \neq \mathcal{E}_{train}$ and $\mathcal{R}_{test} \neq \mathcal{R}_{train}$, including 13 datasets from InGram (Lee et al., 2023): FB-25, FB-50, FB-75, FB-100, NL-0, NL-25, NL-50, NL-75, NL-100, WK-25, WK-50, WK-75, and WK-100.

- **Transductive** datasets for pre-training that $\mathcal{E}_{test} = \mathcal{E}_{train}$ and $\mathcal{R}_{test} = \mathcal{R}_{train}$: FB15k-237 (Toutanova & Chen, 2015), WN18RR (Dettmers et al., 2018), CoDEx-M (Safavi & Koutra, 2020).

These dataset are used to evaluate the model in zero-shot/fine-tuning scenarios. Tables (7), (8), and (9) provide detailed elemental statistics for these datasets. In addition, in response to the "**Supervised SOTA**" methds in Section 5.2, we provide the supervised KGR models that achieved the best performance for each dataset in Tables (8) and (9).

## I EXPERIMENTAL HYPERPARAMETER SETTINGS

In Section 5.2, we evaluate three forms of KRLM, *e.i.*, "Pre-Training" (PT), "Fine-Tuning" (FT), and "End-to-End training from scratch" (E2E). The hyperparameters of KRLM-PT and KRLM-E2E are uniformly set to the values in Table 10. During the pre-training process, we mix the three transductive KGR datasets from Table 7 as the training corpus and train KRLM from scratch for 20 epochs, each containing 10000 steps. In PT and E2E modes, except for the pre-trained parameters of Llama2-7b used for the backbone LLM of KRLM, the parameters of all other modules are randomly initialized using the nn.Linear() function of the Pytorch library. We allocate query triplets with batch

Table 7: Transductive KGR datasets used for model pre-training. "**#Train**", "**#Valid**", and "**#Test**" indicate the training, validation, and testing triplet numbers in each dataset, respectively.

| Datasets | Entities | Relations | #Train | #Valid | #Test |
|----------|----------|-----------|--------|--------|-------|
| FB15k-237 (Toutanova & Chen, 2015) | 14541 | 237 | 272115 | 17535 | 20466 |
| WN18RR (Dettmers et al., 2018) | 40943 | 11 | 86835 | 3034 | 3134 |
| CoDEx-M (Safavi & Koutra, 2020) | 17050 | 51 | 185584 | 10310 | 10311 |

Table 8: IndE KGR datasets used for zero-shot and fine-tuning evaluation. "**Triplets**" represents the number of total triplets contained in a training/validation/testing graph. "**#Valid**" and "**#Test**" are the number of evaluation triplets in the validation and testing graph, respectively.

| Datasets | Relations | Training graph | | Validation Graph | | | Testing Graph | | | Supervised SOTA |
|---|---|---|---|---|---|---|---|---|---|---|
| | | Entities | Triplets | Entities | Triplets | #Valid | Entities | Triplets | #Test | |
| FB-V1 (Teru et al., 2020) | 180 | 1594 | 4245 | 1594 | 4245 | 489 | 1093 | 1993 | 411 | A*Net (Zhu et al., 2022) |
| FB-V2 (Teru et al., 2020) | 200 | 2608 | 9739 | 2608 | 9739 | 1166 | 1660 | 4145 | 947 | NBFNet (Zhu et al., 2021) |
| FB-V3 (Teru et al., 2020) | 215 | 3668 | 17986 | 3668 | 17986 | 2194 | 2501 | 7406 | 1731 | NBFNet (Zhu et al., 2021) |
| FB-V4 (Teru et al., 2020) | 219 | 4707 | 27203 | 4707 | 27203 | 3352 | 3051 | 11714 | 2840 | A*Net (Zhu et al., 2022) |
| NELL-V1 (Teru et al., 2020) | 14 | 3103 | 4687 | 3103 | 4687 | 414 | 225 | 833 | 201 | RED-GNN (Zhang & Yao, 2022) |
| NELL-V2 (Teru et al., 2020) | 88 | 2564 | 8219 | 2564 | 8219 | 922 | 2086 | 4586 | 935 | RED-GNN (Zhang & Yao, 2022) |
| NELL-V3 (Teru et al., 2020) | 142 | 4647 | 16393 | 4647 | 16393 | 1851 | 3566 | 8048 | 1620 | RED-GNN (Zhang & Yao, 2022) |
| NELL-V4 (Teru et al., 2020) | 76 | 2092 | 7546 | 2092 | 7546 | 876 | 2795 | 7073 | 1447 | RED-GNN (Zhang & Yao, 2022) |
| WN-V1 (Teru et al., 2020) | 9 | 2746 | 5410 | 2746 | 5410 | 630 | 922 | 1618 | 373 | NBFNet (Zhu et al., 2021) |
| WN-V2 (Teru et al., 2020) | 10 | 6954 | 15262 | 6954 | 15262 | 1838 | 2757 | 4011 | 852 | NBFNet (Zhu et al., 2021) |
| WN-V3 (Teru et al., 2020) | 11 | 12078 | 25901 | 12078 | 25901 | 3097 | 5084 | 6327 | 1143 | NBFNet (Zhu et al., 2021) |
| WN-V4 (Teru et al., 2020) | 9 | 3861 | 7940 | 3861 | 7940 | 934 | 7084 | 12334 | 2823 | A*Net (Zhu et al., 2022) |

Table 9: IndER KGR datasets used for zero-shot and fine-tuning evaluation. "**Triplets**" represents the number of total triplets contained in a training/validation/testing graph. "**#Valid**" and "**#Test**" are the number of evaluation triplets in the validation and testing graph, respectively.

| Datasets | Training graph | | | Validation Graph | | | | Testing Graph | | | | Supervised SOTA |
|---|---|---|---|---|---|---|---|---|---|---|---|---|
| | Entities | Relations | Triplets | Entities | Relations | Triplets | #Valid | Entities | Relations | Triplets | #Test | |
| FB-25 (Lee et al., 2023) | 5190 | 163 | 91571 | 4097 | 216 | 17147 | 5716 | 4097 | 216 | 17147 | 5716 | InGram (Lee et al., 2023) |
| FB-50 (Lee et al., 2023) | 5190 | 153 | 85375 | 4445 | 205 | 11636 | 3879 | 4445 | 205 | 11636 | 3879 | InGram (Lee et al., 2023) |
| FB-75 (Lee et al., 2023) | 4659 | 134 | 62809 | 2792 | 186 | 9316 | 3106 | 2792 | 186 | 9316 | 3106 | InGram (Lee et al., 2023) |
| FB-100 (Lee et al., 2023) | 4659 | 134 | 62809 | 2624 | 77 | 6987 | 2329 | 2624 | 77 | 6987 | 2329 | InGram (Lee et al., 2023) |
| WK-25 (Lee et al., 2023) | 12659 | 47 | 41873 | 3228 | 74 | 3391 | 1130 | 3228 | 74 | 3391 | 1131 | InGram (Lee et al., 2023) |
| WK-50 (Lee et al., 2023) | 12022 | 72 | 82481 | 9328 | 93 | 9672 | 3224 | 9328 | 93 | 9672 | 3225 | InGram (Lee et al., 2023) |
| WK-75 (Lee et al., 2023) | 6853 | 52 | 28741 | 2722 | 65 | 3430 | 1143 | 2722 | 65 | 3430 | 1144 | InGram (Lee et al., 2023) |
| WK-100 (Lee et al., 2023) | 9784 | 67 | 49875 | 12136 | 37 | 13487 | 4496 | 12136 | 37 | 13487 | 4496 | InGram (Lee et al., 2023) |
| NL-0 (Lee et al., 2023) | 1814 | 134 | 7796 | 2026 | 112 | 2287 | 763 | 2026 | 112 | 2287 | 763 | InGram (Lee et al., 2023) |
| NL-25 (Lee et al., 2023) | 4396 | 106 | 17578 | 2146 | 120 | 2230 | 743 | 2146 | 120 | 2230 | 744 | InGram (Lee et al., 2023) |
| NL-50 (Lee et al., 2023) | 4396 | 106 | 17578 | 2335 | 119 | 2576 | 859 | 2335 | 119 | 2576 | 859 | InGram (Lee et al., 2023) |
| NL-75 (Lee et al., 2023) | 2607 | 96 | 11058 | 1578 | 116 | 1818 | 606 | 1578 | 116 | 1818 | 607 | InGram (Lee et al., 2023) |
| NL-100 (Lee et al., 2023) | 1258 | 55 | 7832 | 1709 | 53 | 2378 | 793 | 1709 | 53 | 2378 | 793 | InGram (Lee et al., 2023) |

size of 4 per GPU for KRLM in each step. One batch of triplets only belongs to one training KG, and their sampling probability is proportional to the total number of triplets contained in that training KG.

After pre-training KRLM, we obtain the best validation checkpoint of KRLM-PT for fine-tuning KRLM-FT on each dataset. The main training hyperparameters of KRLM-FT are the same as those in Table 10. However, to adapt the model to the vastly different number of training triplets in different datasets (ranging from a few thousand to nearly one hundred thousand), we set different training epoch values for different datasets shown in Table 11.

When we train KRLM-E2E on a single transductive KGR dataset, the main hyperparameters of the model are the same as those in Table 10, but the training epochs are changed to 10. In each epoch, the model needs to learn all training triplets in the dataset.

# J    DETAILS EXPERIMENTAL RESULTS

## J.1    DETAILS EXPERIMENTAL RESULTS ON INDUCTIVE DATASETS

Tables 12 and 13 correspond to the detailed experimental results of each method in Table 1 on the IndE and IndER datasets, respectively.

Obviously, the current supervised SOTA baselines can only achieve mediocre performance on almost all inductive datasets, which is attributed to their modeling limitations that make it difficult for them to capture sufficient transferable structure semantics of entities and relations. In addition, considering that these baselines ignore the knowledge structure invariance cross KG domains, they lack of zero-shot reasoning ability across KGs. Therefore, we can only train them from scratch on each dataset during evaluation, which increases the spatiotemporal overhead of model deployment.

Table 10: Hyperperameters of KRLM used in pre-training and end-to-end training from scratch.

| Module | Component | Parameter |
|---|---|---|
| **Knowledge Encoder** | Entity GNN $\text{GNN}_e(\cdot)$ | Layer number $S = 6$
Hidden dim $d = 64$
Message function $\text{Mess}(\cdot) = \text{DistMult}$
Aggregation function $\text{Agg}(\cdot) = \text{Sum}$
Updating function $\text{Update}(\cdot) = \text{Linear}(128, 64)$ |
| | Relation GNN $\text{GNN}_r(\cdot)$ | Layer number $S = 6$
Hidden dim $d = 64$
Message function $\text{Mess}(\cdot) = \text{DistMult}$
Aggregation function $\text{Agg}(\cdot) = \text{Sum}$
Updating function $\text{Update}(\cdot) = \text{Linear}(128, 64)$ |
| | Score function $\mathcal{S}_{\text{struct}}(\cdot)$ | $\text{Linear}(128, 64)$
$\text{ReLU}(\cdot)$
$\text{Linear}(64, 1)$ |
| **KRL Attention Layer** | Llama2-7b backbone | Layer number $N = 32$
Hidden dim $F = 4096$ |
| | Mapping layer $\mathcal{F}_{\text{word}}(\cdot)$ | $\text{Linear}(64, 4096)$ |
| | Mapping layer $\mathcal{F}_{\text{struct}}(\cdot)$ | $\text{Linear}(64, 4096)$ |
| | Scale of knowledge memory | $\mathcal{K} = 50$ |
| **Next-entity Predictor** | Knowledge Decoder $\text{GNN}_p(\cdot)$ | Layer number $S = 6$
Hidden dim $d = 64$
Message function $\text{Mess}(\cdot) = \text{DistMult}$
Aggregation function $\text{Agg}(\cdot) = \text{Sum}$
Updating function $\text{Update}(\cdot) = \text{Linear}(128, 64)$ |
| | Mapping layer $g(\cdot)$ | $\text{Linear}(4096, 64)$ |
| | Score function $\mathcal{S}_{\text{KRLM}}(\cdot)$ | $\text{Linear}(192, 64)$
$\text{ReLU}(\cdot)$
$\text{Linear}(64, 1)$ |
| **Training** | Optimizer | AdamW |
| | Learning rate $\eta$ | 5e-4 |
| | Batch size $b$ | 4 per GPU |
| | Training epochs | 20 |
| | Steps in each epoch | 10000 |
| | Number of negative samples | 256 |
| | KL weight $\lambda$ | 0.5 |

Table 11: Training epochs and steps of KRLM-FT on different inductive datasets. For example, (3, all) means that we fine-tune KRLM on a dataset within 3 epochs and the model needs to learn all the triplets in the training KG.

| Datasets | KRLM-FT |
|---|---|
| FB V1 | (3, all) |
| FB V2 | (3, all) |
| FB V3 | (5, all) |
| FB V4 | (5, all) |
| NELL V1 | (3, all) |
| NELL V2 | (3, all) |
| NELL V3 | (5, all) |
| NELL V4 | (3, all) |
| WN V1 | (3, all) |
| WN V2 | (5, all) |
| WN V3 | (5, all) |
| WN V4 | (3, all) |
| FB-25 | (10, all) |
| FB-50 | (10, all) |
| FB-75 | (10, all) |
| FB-100 | (10, all) |
| NL-0 | (3, all) |
| NL-25 | (5, all) |
| NL-50 | (5, all) |
| NL-75 | (5, all) |
| NL-100 | (3, all) |
| WK-25 | (10, all) |
| WK-50 | (10, all) |
| WK-75 | (10, all) |
| WK-100 | (10, all) |

Table 12: Detailed performance of each model on IndE datasets. "PT" and "FT" mean "pre-training" and "fine-tuning", respectively. Black bold indicates the best result.

| Inductive Datasets | | Supervised SOTA | ULTRA (PT) | ULTRA (FT) | MOTIF (PT) | MOTIF (FT) | TRIX (PT) | TRIX (FT) | MKGL | PROLINK (Llama2-7b) | KRLM (PT) | KRLM (FT) |
|---|---|---|---|---|---|---|---|---|---|---|---|---|
| FB-V1 | Hit@10 | 0.589 | 0.656 | 0.670 | 0.692 | 0.702 | 0.682 | 0.682 | 0.595 | 0.692 | **0.708** | 0.701 |
| | MRR | 0.457 | 0.498 | 0.509 | 0.503 | 0.53 | 0.515 | 0.515 | 0.475 | 0.498 | 0.537 | **0.541** |
| FB-V2 | Hit@10 | 0.672 | 0.700 | 0.710 | 0.716 | 0.744 | 0.730 | 0.730 | 0.681 | 0.745 | 0.748 | **0.752** |
| | MRR | 0.510 | 0.512 | 0.524 | 0.511 | 0.557 | 0.525 | 0.525 | 0.508 | 0.514 | 0.555 | **0.557** |
| FB-V3 | Hit@10 | 0.637 | 0.654 | 0.663 | **0.692** | 0.684 | 0.669 | 0.669 | 0.643 | 0.683 | 0.678 | 0.680 |
| | MRR | 0.476 | 0.491 | 0.504 | 0.500 | 0.519 | 0.501 | 0.501 | 0.486 | 0.485 | 0.514 | **0.522** |
| FB-V4 | Hit@10 | 0.645 | 0.677 | 0.684 | 0.677 | 0.695 | 0.687 | 0.687 | 0.645 | 0.676 | 0.690 | **0.699** |
| | MRR | 0.466 | 0.486 | 0.496 | 0.487 | **0.508** | 0.493 | 0.493 | 0.471 | 0.498 | 0.503 | 0.504 |
| NELL-V1 | Hit@10 | 0.866 | 0.913 | 0.878 | 0.871 | 0.873 | 0.898 | 0.899 | 0.886 | 0.883 | 0.887 | **0.916** |
| | MRR | 0.637 | 0.785 | 0.757 | 0.674 | 0.712 | **0.806** | 0.804 | 0.749 | 0.726 | 0.652 | 0.682 |
| NELL-V2 | Hit@10 | 0.601 | 0.707 | 0.761 | 0.769 | 0.765 | 0.768 | 0.764 | 0.767 | 0.787 | 0.773 | **0.791** |
| | MRR | 0.419 | 0.526 | 0.575 | 0.564 | 0.566 | 0.569 | 0.571 | 0.570 | 0.581 | **0.589** | 0.583 |
| NELL-V3 | Hit@10 | 0.594 | 0.702 | 0.755 | 0.724 | 0.764 | 0.743 | 0.759 | 0.759 | 0.762 | 0.766 | **0.768** |
| | MRR | 0.436 | 0.515 | 0.563 | 0.533 | 0.580 | 0.558 | 0.571 | 0.571 | 0.589 | 0.594 | **0.598** |
| NELL-V4 | Hit@10 | 0.556 | 0.712 | 0.733 | 0.711 | 0.740 | 0.765 | 0.772 | 0.769 | 0.769 | 0.739 | **0.772** |
| | MRR | 0.363 | 0.479 | 0.469 | 0.503 | 0.507 | 0.538 | 0.551 | 0.535 | 0.533 | 0.544 | **0.554** |
| WN-V1 | Hit@10 | **0.826** | 0.768 | 0.793 | 0.778 | 0.806 | 0.791 | 0.798 | 0.822 | 0.788 | 0.783 | 0.800 |
| | MRR | 0.741 | 0.648 | 0.685 | 0.682 | 0.703 | 0.699 | 0.705 | **0.746** | 0.644 | 0.705 | 0.711 |
| WN-V2 | Hit@10 | 0.798 | 0.765 | 0.779 | 0.771 | 0.781 | 0.781 | 0.780 | 0.799 | 0.777 | 0.782 | **0.799** |
| | MRR | 0.704 | 0.663 | 0.679 | 0.663 | 0.680 | 0.678 | 0.682 | **0.712** | 0.669 | 0.696 | 0.700 |
| WN-V3 | Hit@10 | 0.568 | 0.476 | 0.546 | 0.538 | 0.590 | 0.541 | 0.543 | **0.599** | 0.496 | 0.582 | 0.595 |
| | MRR | 0.452 | 0.376 | 0.411 | 0.420 | 0.466 | 0.418 | 0.425 | 0.456 | 0.388 | 0.447 | **0.469** |
| WN-V4 | Hit@10 | **0.743** | 0.705 | 0.720 | 0.718 | 0.733 | 0.723 | 0.722 | 0.741 | 0.733 | 0.723 | 0.738 |
| | MRR | 0.661 | 0.611 | 0.614 | 0.640 | 0.659 | 0.648 | 0.650 | 0.664 | 0.623 | 0.655 | **0.665** |

Table 13: Detailed performance of each model on IndER datasets. "PT" and "FT" mean "pre-training" and "fine-tuning", respectively. Black bold indicates the best result. "-" indicates that a model is not suitable for this KGR task.

| Inductive Datasets | | Supervised SOTA | ULTRA (PT) | ULTRA (FT) | MOTIF (PT) | MOTIF (FT) | TRIX (PT) | TRIX (FT) | MKGL | PROLINK (Llama2-7b) | KRLM (PT) | KRLM (FT) |
|---|---|---|---|---|---|---|---|---|---|---|---|---|
| FB-25 | Hit@10 | 0.371 | 0.640 | 0.635 | 0.640 | 0.635 | 0.650 | 0.650 | - | 0.648 | **0.658** | 0.640 |
| | MRR | 0.223 | 0.388 | 0.383 | 0.384 | 0.388 | 0.393 | 0.393 | - | 0.391 | **0.404** | 0.398 |
| FB-50 | Hit@10 | 0.325 | 0.543 | 0.538 | 0.546 | 0.544 | 0.547 | 0.547 | - | 0.549 | 0.541 | **0.552** |
| | MRR | 0.189 | 0.338 | 0.334 | 0.338 | 0.340 | 0.334 | 0.334 | - | 0.336 | 0.339 | **0.345** |
| FB-75 | Hit@10 | 0.218 | 0.604 | 0.598 | 0.614 | 0.607 | 0.611 | 0.611 | - | 0.616 | 0.618 | **0.620** |
| | MRR | 0.117 | 0.403 | 0.400 | 0.399 | 0.399 | 0.401 | 0.401 | - | 0.407 | 0.409 | **0.414** |
| FB-100 | Hit@10 | 0.271 | 0.642 | 0.643 | 0.628 | 0.642 | 0.635 | 0.633 | - | 0.635 | 0.647 | **0.655** |
| | MRR | 0.133 | 0.449 | 0.444 | 0.428 | 0.439 | 0.436 | 0.436 | - | 0.452 | 0.445 | **0.455** |
| NL-0 | Hit@10 | 0.506 | 0.523 | 0.551 | 0.497 | 0.556 | 0.549 | 0.549 | - | 0.550 | 0.587 | **0.591** |
| | MRR | 0.309 | 0.342 | 0.329 | 0.324 | 0.328 | 0.385 | 0.385 | - | 0.352 | 0.375 | **0.399** |
| NL-25 | Hit@10 | 0.464 | 0.569 | 0.596 | 0.498 | 0.580 | 0.589 | 0.589 | - | 0.589 | 0.586 | **0.596** |
| | MRR | 0.261 | 0.395 | **0.407** | 0.348 | 0.390 | 0.377 | 0.377 | - | 0.396 | 0.394 | 0.401 |
| NL-50 | Hit@10 | 0.453 | 0.570 | 0.595 | 0.532 | 0.573 | 0.548 | 0.555 | - | 0.579 | 0.588 | **0.598** |
| | MRR | 0.281 | 0.407 | 0.418 | 0.373 | 0.414 | 0.404 | 0.405 | - | 0.411 | 0.412 | **0.432** |
| NL-75 | Hit@10 | 0.501 | 0.547 | **0.570** | 0.512 | 0.548 | 0.525 | 0.525 | - | 0.552 | 0.535 | 0.559 |
| | MRR | 0.334 | 0.368 | **0.374** | 0.314 | 0.360 | 0.351 | 0.351 | - | 0.346 | 0.361 | 0.367 |
| NL-100 | Hit@10 | 0.431 | 0.651 | 0.684 | 0.647 | 0.682 | 0.676 | **0.691** | - | 0.684 | 0.667 | 0.688 |
| | MRR | 0.269 | 0.471 | 0.458 | 0.438 | 0.464 | 0.486 | 0.482 | - | 0.471 | **0.493** | 0.489 |
| WK-25 | Hit@10 | 0.169 | 0.532 | 0.535 | 0.493 | 0.505 | 0.496 | 0.493 | - | 0.539 | 0.509 | **0.550** |
| | MRR | 0.107 | 0.316 | 0.321 | 0.311 | 0.317 | 0.305 | 0.300 | - | 0.323 | 0.324 | **0.332** |
| WK-50 | Hit@10 | **0.362** | 0.324 | 0.280 | 0.314 | 0.304 | 0.313 | 0.313 | - | 0.286 | 0.306 | 0.328 |
| | MRR | **0.247** | 0.166 | 0.140 | 0.163 | 0.160 | 0.166 | 0.166 | - | 0.168 | 0.160 | 0.168 |
| WK-75 | Hit@10 | 0.135 | 0.537 | 0.53 | 0.540 | 0.535 | 0.513 | 0.513 | - | 0.535 | **0.540** | 0.538 |
| | MRR | 0.068 | 0.365 | 0.380 | 0.366 | 0.371 | 0.368 | 0.368 | - | 0.370 | **0.390** | 0.384 |
| WK-100 | Hit@10 | 0.309 | 0.286 | 0.286 | 0.282 | 0.284 | 0.299 | 0.299 | - | 0.283 | **0.320** | 0.313 |
| | MRR | 0.186 | 0.164 | 0.168 | 0.164 | 0.173 | 0.188 | 0.188 | - | 0.179 | **0.192** | 0.189 |

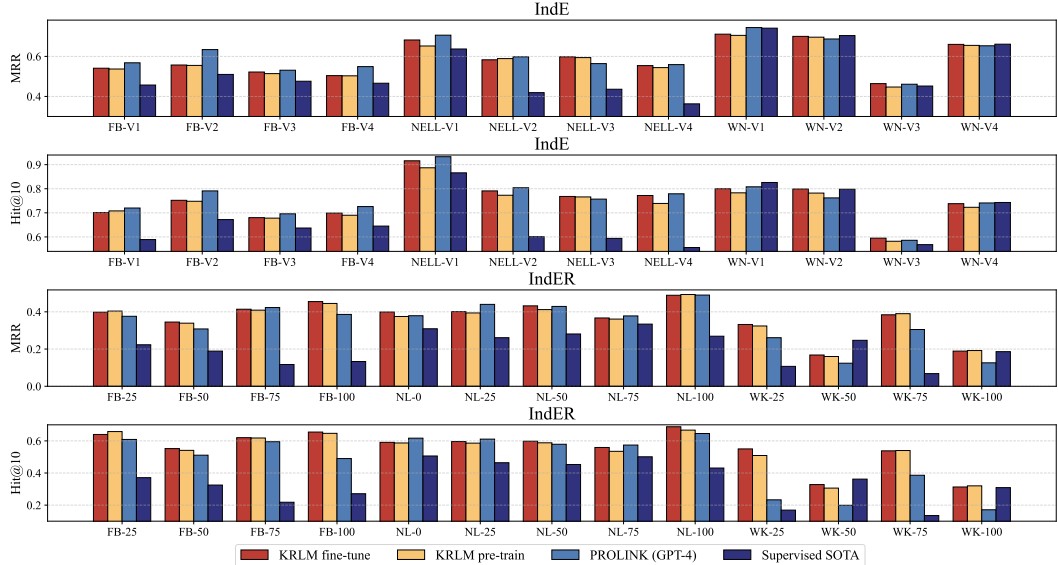

Figure 5: Comparison of our KRLM with more powerful GPT-4. Due to the interference of knowledge distortion, PROLINK using GPT-4 is also unable to effectively handle the inherent knowledge gap between LLMs and KGs. On the IndER datasets with a larger open-domain scope, this reasoning error is more pronounced.

ULTRA is a typical KGFM that proposes a transferable KG reasoning framework driven by relation structure invariance. This approach endows ULTRA with the ability to recognize unfamiliar entities and relations in unseen KGs, thereby enabling reasoning of facts on out-of-domain KGs. Based on this advantage, ULTRA can even perform significantly better than supervised SOTA baselines in zero-shot reasoning configuration, *i.e.*, ULTRA (PT).

MOTIF and TRIX are improvements based on ULTRA. For example, MOTIF extends the four types of relation interactions in the relational graph to hyperedges within three hops (Huang et al., 2025), thereby expanding the structural context of the relations. TRIX iteratively propagates messages between interacting the entity GNN and the relational GNN, enabling the model to perceive more rigorous structural representations and alleviating ULTRA's confusion problem with structurally similar heterogeneous triplets.

The above KGFMs only rely on the sparse structural semantics of KGs, which can easily make the model ignore deeper underlying knowledge. MKGL and PROLINK use the internal knowledge of LLM to extend the structural semantics of KGs, making the reasoning evidence space denser and thus improving the performance of the model. However, MKGL cannot be considered strictly a LLM-based KGFM, as it requires a fixed number of relations based on specific KGs during modeling. Therefore, although MKGL can achieve the best results by training from scratch on some IndE KGR datasets (*e.g.*, WN-V2 and WN-V3), it cannot achieve zero-shot reasoning across KGs and is not suitable for the IndER KGR scenario.

PROLINK adopts a framework that combines large and small models. First, PROLINK uses Llama to plan reasoning paths, and then candidate reasoning paths are mapped to KG space through a pre-trained KGFM (such as ULTRA). This apporach achieves remarkable performance and generalization. However, PROLINK struggles to effectively address the inherent knowledge gap between LLMs and KGFMs, which makes it difficult for PROLINK to effectively overcome the limitations of knowledge distortion on model inference even when using GPT-4 (Figure 5).

In contrast, our proposed KRLM alleviates the LLM knowledge distortion problem caused by the inherent knowledge gap between LLM and KG by coordinating LLM internal knowledge and KG structured knowledge in various modules of LLM.

## J.2 Details Ablation Analysis

Section 5.3 analyzes the effectiveness of various components of KRLM. To alleviate the time overhead caused by multiple pretraining from scratch on large-scale transductive datasets, our ablation experiments perform end-to-end training from scratch on several small inductive datasets (FB-V1, WN-V1, NL-0, and NL-100).

Table 2 provides 8 ablation variants, and the following are their design details:

- **-KEn**. This variant removes the knowledge encoder mentioned from Section 4.1. This encoder is an extremely important module in KRLM, which involves updating special token embeddings in subsequent KRL instructions (Eq. (4)), sampling knowledge memory in KRL attention layer (Eq. (6)), and applying relational knowledge representation in netx entity predictor ((Eqs. (8) and (9)). Therefore, in the absence of a knowledge encoder, we need to remove the knowledge representation token placeholders of entities and relations from KRL instructions, replace the KRL attention layer with the LoRA fine-tuning framework (referring to the LoRA parameter settings in MKGL (Guo et al., 2024)), remove the knowledge decoder from the next-entity predictor (Eq. (8)), replace $\widetilde{p}_i$ in Eq. (9) by $p_i$ in Eq. (8), and remove the relation representation $r_q$ from Eq. (9).

- **-KMe**. This variant removes the knowledge memory mechanism from Section 4.2 and replaces the KRL attention layer with the LoRA fine-tuning framework (referring to the LoRA parameter settings in MKGL (Guo et al., 2024)).

- **-KDe**. This variant removes the knowledge decoder from Section 4.3, replaces $\widetilde{p}_i$ in Eq. (9) by $p_i$ in Eq. (7), and removes $r_q$ from Eq. (9).

- **Atten**. This variant replaces the PAA module in Eqs. (2) and (7) with the attention pooling method, which uses trainable attention weights to average the textual tokens of entities/relations.

- **Mean**. This variant replaces the PAA module in Eqs. (2) and (7) with the mean pooling method, which directly averages the textual tokens of entities/relations.

- **-KD**. This variant removes the KRL distillation module from Eq. (10) and only retains the structural distillation module.

- **-KL**. This variant abandons the knowledge distillation function in Eq. (10), which only retains two cross-entropy losses and removes the calculation process of KL divergence.

- **-KD-KL**. This variant simultaneously removes KRL distillation and KL divergence from Eq. (10), *i.e.*, only uses the simplest single cross-entropy loss.

The results in Table 2 indicate that the knowledge encoder ("**-KEn**") plays an important role in KRLM, as it introduces implicit structural context into LLM, which is more effective in driving knowledge coordination between LLM and KG compared to the explicit knowledge injection method of existing LLM-based KGFMs (Wang et al., 2024b).

The role of a knowledge decoder is to strictly constrain the reasoning results of LLM so that they do not exceed the domain of a specific KG. Therefore, after removing the knowledge decoder ("**-KDe**"), the reasoning of KRLM degenerates into the next-token prediction mechanism of LLM, making it difficult for the model to perceive KG structural knowledge throughout the entire reasoning process, thereby limiting its performance.

The purpose of knowledge distillation in training loss is to coordinate the knowledge in LLMs and KGs from the response side of KRLM. Therefore, the variant "**-KD-KL**" using the simplest cross entropy loss cannot achieve this function, resulting in poor performance. Variants "**-KD**" and "**-KL**" use one-side distillation and double cross-entropy loss coordination methods, respectively, which makes it difficult for them to maximize the interoperability between different knowledge and limits their performance.

The remaining variants ("**-KMe**", "**Atten**", and "**Mean**") mainly focus on the application of different modal knowledge in KRLM, with the significance of enhancing the knowledge context awareness of the hidden state of the last KRL token output by KRLM. Therefore, removing these modules also reduce the reasoning of KRLM, but the impact is not as significant as the variants analyzed above that focus on the coordination of LLM and KG knowledge.

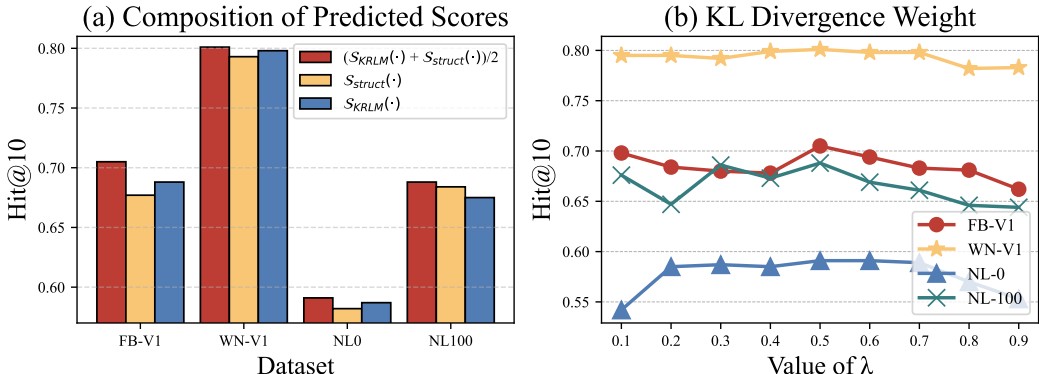

Figure 6: (a) Comparison of different approaches for obtaining predicted scores. (b) Experiments on the proportion of distillation terms in Eq. (9).

In addition to the ablation experiments in Section 5.3, we also compare the impact of different prediction score acquisition methods on the final reasoning of the model. Figure 6(a) shows three methods for obtaining prediction scores. Our KRLM uses a combination of Eqs. (3) and (9), i.e. $\frac{\mathcal{S}_{KRLM}(\cdot) + \mathcal{S}_{struct}(\cdot)}{2}$, to obtain the final prediction scores. $\mathcal{S}_{KRLM}(\cdot)$ and $\mathcal{S}_{struct}(\cdot)$ represent obtaining the final predicted scores of entities using only Eqs. (9) and (3), respectively. Obviously, using a single scoring function can lower the final prediction results of the model. The main reason may be that although we use knowledge mutual distillation in Eq. (10) to align the predicted distributions of KRLM and the knowledge encoder, they still have a preference for their respective modal knowledge. Therefore, to fully integrate the model's expected ratings of entities in different modalities, we use simple average aggregation to achieve effective prediction.

### J.3 ANALYSIS OF THE WEIGHT OF KNOWLEDGE DISTILLATION

Figure 6(b) provides the performance of KRLM for different values of $\lambda$ in Eq. (10). Although the influence of the weight of KL divergence term on model training is not emphasized in relevant literature (Zhang et al., 2018), our experiment still demonstrates the importance of balancing target loss and KL divergence. Therefore, in practical implementation, we uniformly set $\lambda = 0.5$.

### J.4 ANALYSIS ON SPARSE KG REASONING

As shown in Tables 7, 8, and 9, among all the datasets involved in the experiment, FB15k237 had the highest graph density ($1.29 \times 10^{-3}$), while the graph density of the other inductive datasets was concentrated between $10^{-4}$ and $10^{-5}$. Tables 12 and 13 show the Hits@10 and MRR of each method on 25 inductive datasets, where our KRLM achieves SOTA on most of them, demonstrating KRLM's inference advantage on sparse KGs.

In addition, we collect three sparse KG datasets (Lv et al., 2020) derived from FB15k237 (FB15k237_10, FB15k237_20, and FB15k237_50), and conduct further zero-shot sparse-KG reasoning experiments with KRLM on these datasets. The detailed results are presented in Table 14.

Overall, existing KGFM models perform significantly better than supervised SOTA KG reasoning models on sparse KGs, but they do not show clear advantages on dense ones. We attribute this to the relational GNN module in KGFM (Eq. (1)), which is able to induce more generalizable structural semantics from the KG and thus provides additional information for reasoning over sparse KGs. After injecting the inherent knowledge of LLMs, LLM-based KGFMs can further supply dense semantic support to sparse KGs, leading to additional performance gains.

### J.5 QUANTITATIVE EVALUATION OF KNOWLEDGE DISTORTION

We begin by defining the evaluation metric for knowledge distortion, namely the Distortion Rate (DR).

Table 14: Detailed performance of each model on sparse KG datasets. "PT" means a model is pre-trained by the three transductive dataset show in Table 7. Black bold indicates the best result.

| Datasets (density) | | Supervised SOTA | ULTRA (PT) | MOTIF (PT) | TRIX (PT) | MKGL | PROLINK (Llama2-7b) | KRLM (PT) |
|---|---|---|---|---|---|---|---|---|
| FB15k237_10 | Hit@10 | 0.337 | 0.398 | 0.384 | 0.393 | - | 0.383 | **0.409** |
| $(2.11 \times 10^{-4})$ | MRR | 0.219 | **0.248** | 0.236 | 0.246 | - | 0.238 | 0.243 |
| FB15k237_20 | Hit@10 | 0.391 | **0.436** | 0.422 | 0.430 | - | 0.404 | 0.424 |
| $(3.14 \times 10^{-4})$ | MRR | 0.247 | **0.272** | 0.259 | 0.269 | - | 0.262 | 0.269 |
| FB15k237_50 | Hit@10 | 0.458 | 0.526 | 0.508 | 0.521 | - | 0.529 | **0.526** |
| $(6.79 \times 10^{-4})$ | MRR | 0.293 | 0.324 | 0.312 | 0.321 | - | 0.324 | **0.328** |
| FB15k237 | Hit@10 | **0.599** | 0.564 | 0.550 | 0.559 | 0.591 | - | 0.554 |
| $(1.29 \times 10^{-3})$ | MRR | **0.415** | 0.368 | 0.357 | 0.366 | 0.410 | - | 0.381 |

Table 15: Detailed performance of each model on sparse KG datasets for knowledge distortion. "PT" means a model is pre-trained by the three transductive dataset show in Table 7. Black bold indicates the best result.

| FB15k237_10 testing triplets under different background KGs | | ULTRA (PT) | PROLINK (Llama2-7b as backbone LLM) | KRLM (PT) |
|---|---|---|---|---|
| FB15k237_10 | Hit@10 | 0.398 | 0.383 | 0.409 |
| | MRR | 0.248 | 0.238 | 0.243 |
| | DR | 471.42 | 612.78 | **297.01** |
| FB15k237 | Hit@10 | 0.668 | 0.668 | 0.665 |
| | MRR | 0.469 | 0.471 | 0.479 |

DR is used to measure the misjudgment rate of the model before and after changes in KG structure, reflecting the model's ability to autonomously coordinate with KG context. For a query triplet $q = (h, r, ?) \in \mathcal{T}$, let $t$ be the ground truth. Suppose the model assigns a ranking score $s_1^{(q)}$ to $t$ on a clean KG and a score $s_2^{(q)}$ on a noisy KG. If $s_2^{(q)} > s_1^{(q)}$, the distortion rate for this query is recorded as $s_2^{(q)} - s_1^{(q)}$. The overall DR of the model on the noisy KG is given by $\frac{\sum_{q \in \mathcal{T}} \max(0, s_2^{(q)} - s_1^{(q)})}{|\mathcal{T}|}$, with lower values indicating better performance.

According to Appendix J.4, we use FB15k237_10 (Lv et al., 2020) as a sparse dataset extracted from FB15k237. Then, we test the query triplets of FB15k237_10 using the background KGs of FB15k237 and FB15k237_10, respectively. Table 14 reports the performance of structural learning-based (ULTRA) and LLM-based (PROLINK) KGFMs under the pre-trained setting.

Evidently, sparse KGs significantly constrain the reasoning of models due to the limited contextual evidence they can provide, leading to failures on query triplets that would otherwise be manageable. In this scenario, the structural learning capability of GNN modules becomes particularly crucial, enabling ULTRA and KRLM to capture implicit structural contexts in sparse KGs and thereby mitigate reasoning errors. In contrast, PROLINK's explicit prompt-based contextual learning mechanism struggles to extract information highly relevant to the ground truth from the limited number of available KG paths.

## J.6    ADAPTIVE ANALYSIS ON DIFFERENT LLM BACKBONES

We select Llama-2-7b-chat-hf as the backbone in our KRLM to ensure consistency with LLM-based baselines, thereby allowing us to more clearly demonstrate the effectiveness of our proposed method.

To verify the adaptability of the proposed components, we additionally select Mistral-7B-Instruct-v3.0 and Llama-3.1-8B-Instruct as alternative LLM backbones to examine the generality of the knowledge coordination mechanism in our KRLM. We conduct end-to-end training from scratch on four lightweight inductive datasets. The Hit@10 results of all models are summarized in Table 16. The results show that our knowledge coordination mechanism is broadly applicable across different LLM backbones, and it consistently yields improvements over most LLM-based KGFMs.

Table 16: Hit@10 of KRLM under different LLM backbones.

| Dataset | Supervised SOTA | ULTRA | MKGL (Llama-2-7b) | PROLINK (Llama-2-7b) | KRLM (Llama-2-7b) | KRLM (Mistral-7b) | KRLM (Llama-3.1-8b) |
|---|---|---|---|---|---|---|---|
| FB-V1 | 0.589 | 0.670 | 0.595 | 0.692 | **0.705** | **0.696** | **0.708** |
| WN-V1 | **0.826** | 0.793 | **0.822** | 0.788 | 0.801 | 0.805 | 0.808 |
| NL-0 | 0.506 | 0.551 | - | 0.550 | **0.591** | **0.585** | **0.595** |
| NL-100 | 0.431 | 0.684 | - | 0.684 | **0.688** | **0.692** | **0.689** |

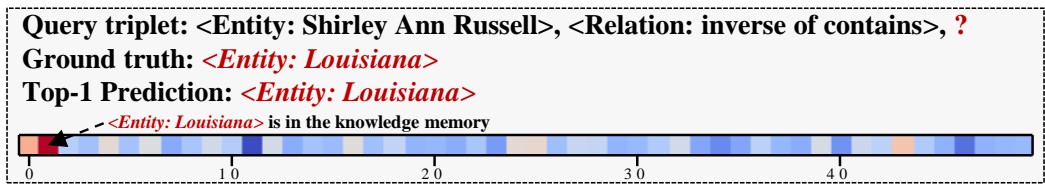

(a) KRLM hits ground truth. KRLM can mine potential correct results from knowledge memory

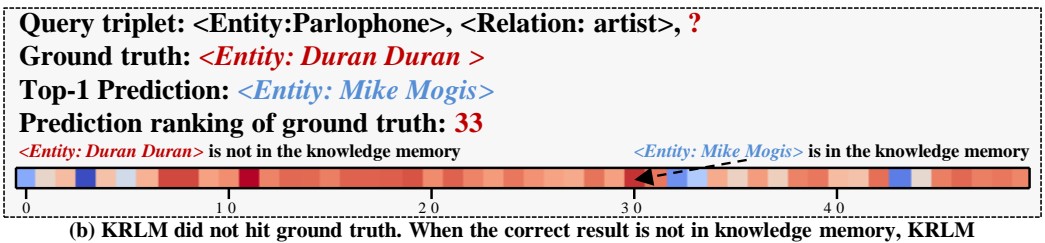

(b) KRLM did not hit ground truth. When the correct result is not in knowledge memory, KRLM attempts to aggregate the context of ground truth from other candidate entities

Figure 7: Visualization of the attention weights over 50 candidate entities in the knowledge memory within a KRL attention layer, illustrating cases where KRLM reasoning succeeds and fails, respectively. (a) KRLM assigns the highest attention weights to the potential answers it finds in the knowledge memory. (b) If the memory lacks potential answers, KRLM attempt to aggregate a broader set of candidate entities to obtain the knowledge context of the ground-truth.

### J.7 CASE STUDY AND ERROR ANALYSIS

This section further analyzes the reasoning mechanism of KRLM from the perspectives of error analysis and case study.

Let's begin with a visual case study. Figure 7 shows the attention weights of candidate entities within the knowledge memory in a KRL attention layer under correct/incorrect reasoning scenarios. Intuitively, when the knowledge memory contains the ground truth entity (included in the top-50 entities selected by Eq. (3)), KRLM tends to highlight its attention weight (shown in Figure 7(a)), even though it is not given the highest score by Eq. (3) among the top-50 entities. This means that KRLM does not rely solely on the scoring mechanism of Eq. (3), it can further filter information in the knowledge memory based on more complex in-context learning in subsequent modules.

In contrast, if the knowledge memory lacks the ground truth, KRLM automatically broadens its attention to include additional candidate entities. As shown in Figure 7(b), this yields far more high-attention weights than in Figure 7(a). By expanding its focus, the model gathers as much reasoning evidence as possible from a wider knowledge context. Although KRLM still fails to infer the ground truth correctly in Figure 7(b), it nonetheless boosts the ranking of the ground truth dramatically (from beyond 50th place to 33rd place).

Furthermore, we explore the universality of the above phenomenon based on the case study in Figure 7. We classify all triplets into two groups, "#Easy" and "#Hard", depending on whether their ground-truth entities are present in the knowledge memory. Table 10 presents the performance of KRLM for each group. Obviously, KRLM tends to correctly reason for "#Easy" triplets in the vast majority of cases, while the Hit@10 of reasoning for "#Hard" triplets tends to approach 1%, which

Table 17: Reasoning results of KRLM (PT) for different categories of query triplets in each dataset. "#Easy" means that the ground truth of a triplet is collected into the knowledge memory, while "#Hard" means the opposite.

| Datasets | Hit@10 | | MRR | |
|---|---|---|---|---|
| | #Easy | #Hard | #Easy | #Hard |
| FB-V1 | **0.857** | 0.007 | **0.658** | 0.010 |
| FB-V2 | **0.888** | 0.074 | **0.660** | 0.022 |
| FB-V3 | **0.892** | 0.009 | **0.674** | 0.011 |
| FB-V4 | **0.878** | 0.016 | **0.639** | 0.013 |
| NELL-V1 | 0.876 | **0.950** | **0.832** | 0.701 |
| NELL-V2 | **0.866** | 0.047 | **0.661** | 0.022 |
| NELL-V3 | **0.887** | 0.179 | **0.699** | 0.084 |
| NELL-V4 | **0.842** | 0.057 | **0.635** | 0.018 |
| WN-V1 | **0.932** | 0.000 | **0.827** | 0.003 |
| WN-V2 | **0.923** | 0.008 | **0.816** | 0.005 |
| WN-V3 | **0.850** | 0.004 | **0.650** | 0.006 |
| WN-V4 | **0.924** | 0.001 | **0.829** | 0.003 |
| FB-25 | **0.835** | 0.022 | **0.515** | 0.018 |
| FB-50 | **0.776** | 0.024 | **0.490** | 0.018 |
| FB-75 | **0.827** | 0.070 | **0.564** | 0.028 |
| FB-100 | **0.856** | 0.068 | **0.598** | 0.027 |
| NL-0 | **0.758** | 0.022 | **0.502** | 0.027 |
| NL-25 | **0.763** | 0.292 | **0.536** | 0.087 |
| NL-50 | **0.801** | 0.016 | **0.565** | 0.020 |
| NL-75 | **0.715** | 0.010 | **0.465** | 0.010 |
| NL-100 | **0.867** | 0.031 | **0.607** | 0.019 |
| WK-25 | **0.778** | 0.005 | **0.491** | 0.016 |
| WK-50 | **0.631** | 0.003 | **0.338** | 0.006 |
| WK-75 | **0.839** | 0.044 | **0.621** | 0.023 |
| WK-100 | **0.688** | 0.006 | **0.427** | 0.007 |

is also the main source of errors made by KRLM. The above analysis indirectly reflects the impact of candidate entity recall methods in the knowledge memory on KRLM reasoning.

## K    LIMITATIONS AND FUTURE WORK

KRLM provides a novel modeling paradigm for existing LLM-based KGR research, which involves injecting KG representations into LLM components in different forms. However, the limitations of KRLM in terms of reasoning cost hinder its application in a wider range of knowledge-based reasoning environments (see **Appendix G** for analysis of reasoning complexity).

In the future, we plan to inject KG context into LLMs from the perspective of knowledge editing (Meng et al., 2023; Zhang et al., 2024a; Fang et al., 2025) such as the null-space projection (Fang et al., 2025), this method only requires minimal computational overhead. In addition, as knowledge editing directly affects the parameter-level knowledge in LLMs, it has the potential to make KG context and LLM internal knowledge self-consistent.

Another way to alleviate the compute bottleneck is to use ULTRA (Galkin et al., 2024) as a relation tokenizer and employ a smaller LLM, fine-tuned to treat relation embeddings as atomic tokens, as a rule generator. The generated candidate KG rules can then be processed using a neuro-symbolic embedding model for lightweight fuzzy-logical reasoning. This method can enhance the interpretability of the model while optimizing inference time.

