# OpenReview forum: "Knowledge Reasoning Language Model: Unifying Knowledge and Language for Inductive Knowledge Graph Reasoning"
_ICLR.cc/2026/Conference — ICLR 2026 Poster_

### Official Review · Reviewer_LqgZ · 2025-10-28

**Soundness:** 2
**Presentation:** 2
**Contribution:** 2
**Rating:** 4
**Confidence:** 3

**Summary:**

This paper presents the Knowledge Reasoning Language Model (KRLM), a novel framework for inductive knowledge graph reasoning (KGR). The primary goal is to address the "knowledge distortion" problem in existing LLM-based approaches, where sparse contextual information from a knowledge graph (KG) can interfere with or override the dense intrinsic knowledge of the LLM. KRLM aims to create a more unified coordination between the LLM's internal knowledge and the external KG's structural knowledge.

**Strengths:**

The paper clearly identifies and articulates the "knowledge distortion" problem, which is a significant and practical challenge in combining LLMs with KGs. The goal of unifying the two knowledge sources is highly relevant.

The proposed KRLM is a sophisticated system where each component is designed with a clear purpose that ties back to the central goal of knowledge coordination. The KRL instruction format, the memory-augmented attention, the structured predictor, and the mutual distillation loss all work in concert. This is a significant engineering and research effort.

**Weaknesses:**

The proposed KRLM architecture is very complex. It involves multiple GNNs (for relations, entities, and the projection head), a modified attention mechanism, and a custom tokenizer, all integrated with a base LLM. While the results are strong, this complexity raises questions about its practical scalability and efficiency. The computational complexity analysis in Appendix E confirms that this is a heavy model. Could the authors comment on the trade-offs between this complexity and the performance gains? Is it possible to achieve similar benefits with a simpler architecture?

The experiments are based on Llama2-7b. While this is a reasonable choice, the field of LLMs is moving incredibly fast. How dependent are the architectural choices and performance gains on this specific base model? Would the same knowledge coordination mechanisms be as effective or even necessary with more advanced, capable models (e.g., Llama-3, Mistral, etc.) that might have better inherent reasoning and context-handling abilities?

**Questions:**

see weaknesses

---

> ### Author Response · Authors · 2025-11-22
> **Response to Weakness**
>
> > **Weakness:**
> >- Model complexity & efficiency concerns: KRLM integrates multiple GNNs, modified attention, and a custom tokenizer, resulting in high computational cost and raising questions about scalability and whether similar gains could be achieved with a simpler architecture.
> >- Dependence on Llama2-7B & generalizability: The approach is evaluated only on Llama2-7B, leaving uncertainty about how well the design and its performance improvements transfer to newer, more capable LLMs (e.g., Llama-3, Mistral).
>
> **Response:** Thank you for your careful and insightful review. We provide our responses regarding both the model complexity and the choice of LLM backbone as follows:
>
> **(I) Model Complexity Analysis:** We acknowledge that, in theory, KRLM inherits the combined computational complexity of both GNN message-passing and LLM self-attention. However, in practical deployment, this cost can be substantially mitigated.
>
> - **Efficient GNN implementation:** In our design, the message-passing mechanism of the GNN module is implemented using sparse-matrix operations provided by the PyTorch-geometric library. **This reduces the computational complexity from $O(|\mathcal{T}|d+|\mathcal{E}|d^2)$ to $O(|\mathcal{E}|d)$ [1], where $|\mathcal{T}|$ denotes the number of triplets (typically much larger than the number of entities $|\mathcal{E}|$)**. This optimization significantly lowers the overall computational cost in practice.
>
> - **Inference-time efficiency comparison.** To further demonstrate that **KRLM does not incur substantial overhead relative to LLM-based KGFMs, we report the actual inference-time performance of KRLM alongside several representative baselines**. We selected two datasets with the highest (FB15k237) and lowest (NELL-V1) graph densities within our experimental scope as benchmarks. The following table reports the inference time (seconds per batch) of both latest LLM-based (KRLM, MKGL, PROLINK) and structural learning-like (ULTRA, MOTIF, TRXI) KGFMs. For consistency, we set the test batch size to 16 and used Llama-2-7b as the backbone for all LLM-based KGFMs, conducting experiments on a single NVIDIA A100 GPU.
>
> |Dataset|KRLM (Ours)|MKGL|PROLINK|ULTRA|MOTIF|TRIX|
> |-|-|-|-|-|-|-|
> |FB15k237|2.2308±0.0329|1.9843±0.0411|The dataset is not provided by the preprocessing code|0.1436±0.0090|0.2508±0.0123|0.2212±0.00987|
> |NELL-V1|1.1817±0.0672|0.9916±0.0620|4.3536±0.0411|0.0129±0.0001|0.0211±0.0001|0.0132±0.0000|
>
> - **Future improvement strategy:** Alleviating compute bottlenecks has been one of our recent research priorities. **One promising direction is to use the pre-trained ULTRA [1] as a unified relation tokenizer and employ a smaller LLM, fine-tuned to treat KG rules as new relation token-based language, as a rule generator. The generated KG rules can then be processed using a neuro-symbolic model for lightweight fuzzy-logical reasoning**. At present, we have built a primary rule generator prototype on WN18RR using **Qwen2.5-0.5B**, achieving an average accuracy of around 70% in rule generation. We believe this provides a promising path toward mitigating the compute bottlenecks faced by LLMs in large-scale KGs.
>
> If you are interested in this topic, we would be delighted to further discuss it!
>
> [1] Galkin M, Yuan X, Mostafa H, et al. Towards Foundation Models for Knowledge Graph Reasoning. ICLR 2024.
>
> **(II) Different choices of LLM backbones:** We select Llama-2-7B as the backbone in our experiments to ensure consistency with LLM-based baselines, thereby allowing us to more clearly demonstrate the effectiveness of our proposed method.
>
> In response to this comment, we additionally select **Mistral-7B-Instruct-v3.0** and **Llama-3.1-8B-Instruct** as alternative LLM backbones to examine the generality of the knowledge coordination mechanism in our KRLM. Due to time constraints, we conduct end-to-end training from scratch on four lightweight inductive datasets. The Hit@10 results of all models are summarized in the table below:
>
> |Dataset|Supervised SOTA|ULTRA|MKGL (Llama-2-7b-chat-hf)|PROLINK (Llama-2-7b-chat-hf)|KRLM (Llama-2-7b-chat-hf)|KRLM (Mistral-7b-instruct-v0.3)|KRLM (Llama-3.1-8b-instruct)|
> |-|-|-|-|-|-|-|-|
> |FB-V1|0.589|0.670|0.595|0.692|**0.705**|**0.696**|**0.708**|
> |WN-V1|**0.826**|0.793|**0.822**|0.788|0.801|0.805|0.808|
> |NL-0|0.506|0.551|-|0.550|**0.591**|**0.585**|**0.595**|
> |NL-100|0.431|0.684|-|0.684|**0.688**|**0.692**|**0.689**|
>
> The results show that our knowledge coordination mechanism is broadly applicable across different LLM backbones, and it consistently yields improvements over most LLM-based KGFMs.
>
> We have uploaded the source codes about Mistral-7B and Llama-3.1-8B to the anonymous repository referenced in the paper, under the files `mistral_model.py` and `llama3_model.py`, to facilitate reproducibility.
>
> We hope the above clarifications address the reviewer’s concerns. Thank you again for your support and understanding!

---

### Official Review · Reviewer_ajSR · 2025-10-29

**Soundness:** 3
**Presentation:** 2
**Contribution:** 3
**Rating:** 6
**Confidence:** 4

**Summary:**

This paper presents KRLM, an LLM-based foundation model for inductive knowledge graph (KG) reasoning. It aims at mitigating the issue of knowledge distortion. KRLM introduces several key components: a custom knowledge reasoning language (KRL) instruction format and tokenizer, a KRL attention layer that dynamically integrates intrinsic LLM knowledge with KG context through a memory mechanism, and a structure-aware next-entity predictor. Extensive experiments across 28 datasets demonstrate that KRLM consistently outperforms state-of-the-art models in both zero-shot and fine-tuning settings.

**Strengths:**

- The paper proposes a well-integrated architecture that effectively aligns LLM representations with KG structure.

- The proposed method shows consistent performance gains across a wide range of datasets, particularly in inductive settings, with thorough comparisons to both structural and LLM-based baselines.

**Weaknesses:**

- The contribution is primarily empirical; a stronger theoretical justification for how KRLM’s architectural choices address knowledge distortion would enhance the paper’s depth.

- While KRLM employs a memory-efficient tokenizer, the paper lacks discussion on computational efficiency, including training time and parameter overhead compared to prior models.

- The model heavily relies on KRL-format instructions, yet the design choices—such as instruction length, style, and vocabulary—are not systematically analyzed.

**Questions:**

- The concept of “knowledge distortion”, while intuitive, is loosely defined. Can it be quantified or diagnosed more rigorously? A deeper analysis would help ground this notion.

- Table 1 emphasizes accuracy-based metrics (e.g., MRR, Hits@10), but omits key efficiency metrics such as FLOPs, memory footprint, and wall-clock time for fine-tuning vs. pretraining. These are crucial for assessing practical viability.

- The paper would benefit from qualitative examples illustrating how KRLM produces more faithful or interpretable reasoning compared to baseline LLMs.

---

> ### Author Response · Authors · 2025-11-21
> **Response to Weakness 1**
>
> > **Weakness:** The contribution is primarily empirical; a stronger theoretical justification for how KRLM’s architectural choices address knowledge distortion would enhance the paper’s depth.
>
> **Response:** Thank you for the thoughtful comment. Our core contribution lies in proposing **a new coupling mechanism that is essential for unifying LLM knowledge and KG structural reasoning**. We highlight the core innovation as follows:
>
> **(I) Research motivation:** Our work is motivated by the need to mitigate the **knowledge distortion** problem commonly seen in LLM-based KGR methods that rely on explicit KG-path prompts. Such models often treat limited-hop KG paths as if they were complete evidence, which may cause the LLM to overlook its broader pre-trained priors, leading to biased or incorrect reasoning. To address this challenge, our core idea is to modify the LLM architecture so that it can **query broader KG structural context** and **flexibly fuse KG context with LLM internal priors**, enabling autonomous coordination between external KG entries and the LLM intrinsic knowledge.
>
> **(II) Core algorithm contribution:** To enable this autonomous coordination, we propose a **knowledge memory based KRL attention mechanism**. Building on the standard LLM architecture, we inject GNN-encoded entity structural embeddings into each self-attention layer as a trainable **knowledge memory**. **Compared to the explicit retrieval of KG paths, KG embeddings provided by GNN contain richer implicit context**. During inference, **the KRL attention layer simultaneously extracts instruction context and queries relevant KG structural information stored in the memory**. The attention mechanism then performs a soft fusion between LLM priors and KG structural context, effectively reducing knowledge distortion during KGR.
>
> **(III) Theoretical Analysis of Effectiveness:** We first present the modeling formulation of the $n$-th KRL attention layer:
>
> $\mathbf{H}^{(0)}=\mathbf{T}$,
>
> $\mathbf{A}=\text{softmax}(\frac{\mathbf{H}^{(n-1)}\mathbf{M}_{Q}^{(n)}\mathbf{E}_m^T||(\mathbf{H}^{(n-1)}\mathbf{W}_Q^{(n)}[\mathbf{H}^{(n-1)}\mathbf{W}_K^{(n)}]^T+\mathbf{MASK})}{\sqrt{F}})$,
>
> $\mathbf{H}^{(n)}=\mathbf{A}[\mathbf{E}_{m}\mathbf{M}_V^{(n)}||\mathbf{H}^{(n-1)}\mathbf{W}_V^{(n)}]$.
>
> Here, $\mathbf{T}\in\mathbb{R}^{L\times F}$ denotes the token embedding sequence of the KRL instruction. $\mathbf{W}_Q^{(n)}$, $\mathbf{W}_K^{(n)}$, and $\mathbf{W}_V^{(n)}$ are the pretrained weights of the LLM, and $\mathbf{MASK}$ is the causal attention mask. $\mathbf{E}_m$ is the knowledge memory composed of the KG embeddings of top-$K$ entities, $\mathbf{M}_Q^{(n)}$ and $\mathbf{M}_V^{(n)}$ are the encoding parameters of $\mathbf{E}_m$, enabling the knowledge memory to be deeply coupled with the LLM’s context learning of the KRL instruction.
>
> Since the prediction of the target entity is based on the last token hidden state $\mathbf{h}_L^{(n)}\in\mathbf{H}^{(n)}$, we rewrite the above formulation with $\mathbf{h}_L^{(n)}$ as the focus:
>
> $\mathbf{h}_L^{(n)}=\sum _{l=1} ^{L}\alpha _l\mathbf{h}_l^{(n-1)}\mathbf{W}_V^{(n)}+\sum _{k=1}^K\beta_k\mathbf{e}_k\mathbf{M}_V^{(n)}$,
>
> $\alpha_l=\frac{\exp(<\mathbf{h} _L ^{(n-1)}\mathbf{W} _Q ^{(n)};\mathbf{h} _l ^{(n-1)}\mathbf{W} _K ^{(n)}>)}{\sqrt{F}[\sum _{j=1} ^L\exp(<\mathbf{h} _L ^{(n-1)}\mathbf{W} _Q ^{(n)};\mathbf{h} _j ^{(n-1)}\mathbf{W} _K ^{(n)}>)+\sum _{k=1} ^K\exp(<\mathbf{h} _L ^{(n-1)}\mathbf{M} _Q ^{(n)};\mathbf{e} _k>)]}$, $\mathbf{e}_k\in\mathbf{E}_m$,
>
> $\beta_k=\frac{\exp(<\mathbf{h} _L ^{(n-1)}\mathbf{M} _Q ^{(n)};\mathbf{e} _k>)}{\sqrt{F}[\sum _{j=1} ^L\exp(<\mathbf{h} _L ^{(n-1)}\mathbf{W} _Q ^{(n)};\mathbf{h} _j ^{(n-1)}\mathbf{W} _K ^{(n)}>)+\sum _{z=1} ^K\exp(<\mathbf{h} _L ^{(n-1)}\mathbf{M} _Q ^{(n)};\mathbf{e} _z>)]}$, $\mathbf{e}_z\in\mathbf{E}_m$,
>
> From the perspective of contextual learning, $\alpha_l$ shows how $\mathbf{E}_m$ participates in scaling the pre-trained LLM’s attention allocation over the instruction context, thereby modulating the relative contribution of each contextual token to $\mathbf{h}_L^{(n)}$.
>
> Meanwhile, the term $\sum _{k=1}^K\beta_k\mathbf{e}_k\mathbf{M}_V^{(n)}$ serves as supplementary semantic information that is injected into the final representation, which functions similarly to a bias semantic correction derived from external KG knowledge.
>
> **This analysis theoretically explains how the KRL attention layer coordinates the structural knowledge in $\mathbf{E}_m$ with the pretrained priors in LLM: By allowing $\mathbf{E}_m$ to influence both the attention-weight scaling in contextual computation and the semantic correction of the aggregated representation, the model effectively couples external KG entries with LLM internal priors, thereby mitigating knowledge distortion.**
>
> **A more detailed theoretical discussion is provided in Appendix D of the paper, hoping to address the reviewer's concerns!**

---

> ### Author Response · Authors · 2025-11-21
> **Response to Weakness 2**
>
> > **Weakness:** While KRLM employs a memory-efficient tokenizer, the paper lacks discussion on computational efficiency, including training time and parameter overhead compared to prior models.
>
> **Response:** Thank you for pointing out the importance of computational efficiency. We apologize for the lack of explicit discussion in the original submission.
>
> According to this comment, we have calculated the trainable parameters of MKGL and our KRLM, as well as the training time on the FB15k237 dataset with a uniform batch of 4 per GPU. The statistical results are shown in the following table.
>
> |Model (Llama2-7b as backbone)|trainable parameters|training time per epoch|
> |-|-|-|
> |MKGL|18 M (16.78 M for LoRA)|1 h 8 min / 4 X A100 GPU|
> |KRLM (Ours)| 18.49 M (16.78 M for the knowledge memory in the KRL attention layer) |1 h 20 min / 4 X A100 GPU|
>
> KRLM requires embedding GNN in the tokenizer and next-token predictor of LLM, which slightly increases the parameters. However, it is consistent with MKGL in the main fine-tuning parameters of LLM (KRL attention layer V.S. LoRA). To ensure generalization, KRLM requires additional cost to construct a relational graph for real-time perturbed KGs in each batch, resulting in a training time of about 12 minutes longer per epoch than MKGL.
>
> **Although KRLM incurs additional training costs, it offers substantially stronger generalization compared to MKGL**. Specifically, KRLM requires only a single pre-training phase on a large-scale KG, after which it can perform **training-free zero-shot reasoning** on entirely new KGs (**refer to KRLM (PT) in Tables 1, 12, and 13 in our submitted paper**). In contrast, MKGL is not a fully generalizable KGFM in the strict sense. While it can effectively recognize unseen entities within each inductive dataset, it cannot transfer zero-shot across different inductive datasets. Consequently, **MKGL must be retrained for every new dataset, which significantly increases its deployment overhead**.

---

> ### Author Response · Authors · 2025-11-21
> **Response to Weakness 3**
>
> > **Weakness:** The model heavily relies on KRL-format instructions, yet the design choices—such as instruction length, style, and vocabulary—are not systematically analyzed.
>
> **Response:** Thank you for this valuable comment. We have added a more systematic analysis of our designed KRL instruction from the perspectives of the style, vocabulary, length, and construction time cost.
>
> **(I) Style and vocabulary of the KRL instruction:** Given a query triplet $(h, r, ?)$, we first provide its format of a KRL instruction below to facilitate the reviewer’s understanding:
>
> **>> Instruction begin**
>
> **Instruction:** Define the word format for a new language as <Type: Text Description>. Suppose you are a linguistic expert who are learning this new language. Given the following vocabulary:
>
> |Word|Type|Text description|Knowledge representation|
> |-|-|-|-|
> |`<Entity: name of h>`|Entity|description of $h$|[KG embedding of $h$]|
> |`<Relation: name of r>`|Relation|description of $r$|[KG embedding of $r$]|
>
> Please complete the next word '?' in the given sentence:
>
> `<Entity: name of h>` `<Relation: name of r>`?
>
> **Response:** `<Entity: name of h>` `<Relation: name of r>`
>
> **>> Instruction end**
>
> A KRL instruction consists of three types of tokens: **word-level embeddings**, **KG embeddings**, and **LLM-pretrained tokens**.
>
> **① Word-level embeddings** refer to the principal attribute aggregation result of the text strings of entities and relations after looking up the LLM pretrained token table. Given an entity $h$ expressed as the string *“<Entity: name of h>”*, we feed this string into the LLM’s tokenizer to obtain an embedding sequence $[\mathbf{t}_ 1,\mathbf{t}_ 2,...,\mathbf{t}_ n]\in\mathbb{R}^{n\times F}$. We then apply four pooling operations, mean, std, max, and min, to obtain $\mathbf{t}_ {mean}$, $\mathbf{t}_ {std}$, $\mathbf{t}_ {max}$, $\mathbf{t}_ {min}\in\mathbb{R}^F$. A trainable MLP layer encodes the concatenation of these pooled vectors into a representation of dimension $F$, which serves as the word-level embedding of $h$, denoted as `<Entity: name of h>`. **This design avoids expanding the LLM’s pretrained embedding table to accommodate new entities, thereby improving the model’s scalability and generalization ability**.
>
> **② The KG embeddings of entities/relations** are obtained from a GNN-based KG reasoning model. To enable zero-shot generalization on unseen KGs, we adopt ULTRA [1], a GNN-based KG foundation model, to produce structural embeddings for entities and relations. These embeddings are then projected through a trainable MLP layer to match the LLM hidden dimension $F$ and injected into the KRL instruction as [KG embedding of $h$] and [KG embedding of $r$].
>
> **③ LLM pretrained tokens**. These are standard tokens in the KRL instruction that fall outside the above two categories and are directly provided by the pretrained LLM.
>
> **(II) Analysis of the instruction length:** The following table shows the average number of instruction tokens of MKGL and PROLINK, two latest LLM-based KGFMs (Llama2-7b as the backbone), on 25 inductive datasets in this paper, to directly reflect the token efficiency of our KRL instruction.
>
> |Model|Length|
> |-|-|
> |**KRLM (ours)**|118.75$\pm$5.14|
> |**MKGL**|114.10$\pm$4.38|
> |**PROLINK**|414.04$\pm$8.80|
>
> Because PROLINK requires extensive relational context to prompt the LLM in generating candidate target entity types, its total token counts are significantly higher than those of other methods.
>
> **KRLM adopts a compact word-level token vocabulary table, which makes the KRL instruction prompt more concise**. The instruction format of MKGL is similar to that of KRLM. However, the vocabulary used in MKGL’s instructions does not include KG embedding information. Therefore, MKGL uses slightly fewer tokens than KRLM.
>
> **(III) Latency of instruction construction:** We select two datasets with the highest (FB15k237) and lowest (NELL-V1) graph densities as benchmarks. The following table reports the inference time (seconds per batch) of each component in KRLM. For consistency, we set the test batch size to 16 and conduct experiments on a single NVIDIA A100 GPU.
>
> The overall inference time of KRLM differs by approximately one second between FB15k237 and NELL-V1, which are acceptable to humans. **However, we found that the main module that affects the inference latency is the KRL tokenizer, because it contains an ULTRA module, which needs to read the whole KG for structural context learning of entities and relations**.
>
> | Dataset | KRLM (Overall) |KRL tokenizer|LLM with KRL attention layers|Next-entity predictor|
> |-|-|-|-|-|
> |FB15k237|2.2308±0.0329|1.2413±0.0179|1.1206±0.2408|0.0961±0.0200|
> |NELL-V1|1.1817±0.0672|0.0293±0.0019|1.0554±0.0535|0.0862±0.0047|
>
> [1] Galkin M, Yuan X, Mostafa H, et al. Towards Foundation Models for Knowledge Graph Reasoning. ICLR 2024.

---

> ### Author Response · Authors · 2025-11-21
> **Response to Question 1**
>
> > **Question:** The concept of “knowledge distortion”, while intuitive, is loosely defined. Can it be quantified or diagnosed more rigorously? A deeper analysis would help ground this notion
>
> **Response:** Thank you for your insightful suggestion. According to this comment, we first clarify the definition and evaluation criteria of knowledge distortion:
>
> **Knowledge distortion** refers to the **systematic degradation of a model’s reasoning** when the model is provided with sparse KG evidence, despite being able to answer the same query correctly under a dense or adequate KG context.
>
> **Distortion rate (DR):** Given a query triplet $q=(h, r, ?)\in\mathcal{T}$, let $t$ be the ground truth. Suppose the model assigns a ranking score $s_1^{(q)}$ to $t$ on a dense KG and a score $s_2^{(q)}$ on a sparse KG. If $s_2^{(q)}>s_1^{(q)}$, the distortion rate for this query is recorded as $s_2^{(q)}-s_1^{(q)}$. The overall DR of the model on the sparse KG is given by $\frac{\sum_{q\in\mathcal{T}}\max(0,s_2^{(q)}-s_1^{(q)})}{|\mathcal{T}|}$, with lower values indicating better performance.
>
> **Experimental evaluation:** We have collected the graph densities ($\frac{\text{triplet number}}{\text{entity number}\times(\text{entity number}-1)}$) of all KG datasets used in our experiments. Among them, **FB15k-237** exhibits the highest density of $1.29\times10^{-3}$. To further examine the behavior of our KRLM on sparse KGs, we additionally collected **FB15k237_10** [1], a sparse dataset extracted from FB15k237, with the density of $2.11\times10^{-4}$. **Then, we test the query triplets of FB15k237_10 using the background KGs of FB15k237 and FB15k237_10, respectively**. The table below reports the performance of structural learning-based (ULTRA) and LLM-based (PROLINK, KRLM) KGFMs under the pre-trained setting.
>
> |FB15k237_10 testing triplets under different background KGs|ULTRA|PROLINK (Llama2-7b as 10-shot prompt backbone)|KRLM (ours)|
> |-|-|-|-|
> |FB15k-237|MRR: 0.469; Hits@10: 0.668|MRR: 0.471; Hits@10: 0.668|MRR: 0.479; Hits@10: 0.665|
> |FB15k237_10|MRR: 0.248; Hits@10: 0.398; DR: 471.42|MRR: 0.238; Hits@10: 0.383; DR: 612.78|MRR: 0.243; Hits@10: 0.409; **DR: 297.01**|
>
> Evidently, sparse KGs significantly constrain model reasoning due to the limited contextual evidence they provide, leading to failures on query triplets that would otherwise be manageable. In this scenario, PROLINK’s explicit prompt-based contextual learning mechanism struggles to extract highly relevant information from the limited number of available KG paths, leading to the highest DR value. ULTRA uses a stacked GNN module to capture broader KG context, thereby alleviating to some extent the interference of sparse KGs on model inference. However, because GNNs are inherently sensitive to structural information, severe sparsification of the KG still leads to a substantial degradation in ULTRA’s performance.
>
> In contrast, our KRLM has the smallest DR, which we believe is due to its reasonable coordination of KG context and LLM priors, enabling it to utilize LLM's implicit knowledge to stabilize the inference performance of the model in situations where KG inference evidence is insufficient.
>
> [1] Lv X, Han X, Hou L, et al. Dynamic Anticipation and Completion for Multi-Hop Reasoning over Sparse Knowledge Graph. EMNLP. 2020: 5694-5703.

---

> ### Author Response · Authors · 2025-11-21
> **Response to Question 2**
>
> > **Question:** Table 1 emphasizes accuracy-based metrics (e.g., MRR, Hits@10), but omits key efficiency metrics such as FLOPs, memory footprint, and wall-clock time for fine-tuning vs. pretraining. These are crucial for assessing practical viability.
>
> **Response:** Thank you for your insightful suggestion. We report efficiency metrics from both the pre-training and fine-tuning perspectives, as summarized below.
>
> During training, there is natural step-to-step variability in both the number of input tokens and the size of KG dropout. To obtain a stable and representative estimate, we compute the **average forward TFLOPs over 100 steps**. (Backward propagation and optimizer updates theoretically introduce an additional 2~3 × TFLOPs.)
>
> For fine-tuning efficiency, we further include results on the **largest inductive dataset (FB25)** and the **smallest inductive dataset (FB-V1)** to provide an upper-lower bound range of computational overhead.
>
> The table below reports KRLM’s TFLOPs, memory footprint, and wall-clock time under the configuration of **batch size = 4 per GPU × 4 GPUs**. We hope these measurements sufficiently address the reviewer’s concerns regarding computational efficiency.
>
> |Metrics|Pre-train (3 transductive dataset)|Fine-tune (FB V1)|Fine-tune (FB25)|
> |-|-|-|-|
> |TFLOPs of forward propagation| 3.3436$\pm$0.4540 | 3.2755$\pm$0.5208 | 3.3312$\pm$0.4859 |
> |Training Memory footprint|36.12 GB|32.57 GB|32.67 GB|
> |Wall-clock time| 3h10m per epoch $\times$ 20 epochs |7m28s per epoch $\times$ 3 epochs|12m13s per epoch $\times$ 10 epochs|

---

> ### Author Response · Authors · 2025-11-21
> **Response to Question 3**
>
> > **Question:** The paper would benefit from qualitative examples illustrating how KRLM produces more faithful or interpretable reasoning compared to baseline LLMs.
>
> **Response:** This is a very constructive comment. Below is a response for faithful and interpretable reasoning of LLM-based KGFMs, based on both the current design of our study and the future improvements:
>
> **(I) Faithful reasoning:** In our work, **the faithfulness of our KRLM is reflected in the fact that all inferred entities are strictly constrained to the given KG domain**. In the *Introduction*, we point out that existing LLM-based KGR models may generate *out-of-scope* answers. Such answers are not necessarily factually incorrect, but they fall outside the candidate entity space, making fair evaluation difficult.
>
> To resolve this problem, we draw inspiration from the operation in the next-token predictor of LLM, especially the use of the last-token hidden state and the MLP head. Specifically, we aggregate the token embeddings of each entity’s textual string into a unique *word-level embedding*. Then, using the KRL instruction’s last-token hidden state, we query the entity word-embedding table and select the entity with the highest score as the inference output, i.e., a *next-entity prediction*. This operation constrains the prediction space of KRLM strictly within the set of allowed entities, fundamentally preventing the LLM from producing entities outside the predefined KG domain.
>
> To intuitively demonstrate how existing LLM-based KGR methods may produce out-of-scope answers, we construct a toy example using the triplet `<sport: baseball, sport fans in country, country: china>` in the NELL-V2 dataset and prompt GPT-4 as follows:
>
> > **[Instruction]**
> >
> > *Refer the following triplets:*
> >
> > `<sport: baseball, sport has sports team position, sports team position: rules>`
> >
> > `<sport: baseball, sport uses equipment, sports equipment: golf_club>`
> >
> > `<sport: baseball, sport uses stadium, stadium or event venue: target_field>`
> >
> > `<sport: baseball, sport has sports team position, sports team position: line>`
> >
> > *Reason the possible tail entity of the incompleted triplet:*
> >
> > `<sport: baseball, sport fans in country, ?>`
> >
> > **[Response]**
> >
> > Baseball is:
> >
> > - historically rooted in **the United States** (MLB, origin of many rules, largest fanbase);
> > - extremely popular in **Japan** (NPB league, major cultural following);
> > - a national passion in several Latin American countries, especially **Dominican Republic**, **Cuba**, **Venezuela**, and **Puerto Rico**.
> >
> > If the knowledge graph expects **one** primary tail entity, the strongest and most canonical answer is:
> >
> > **United States**
> >
> > If multiple are allowed:
> >
> > - **Japan**
> > - **Dominican_Republic**
> > - **Cuba**
> > - **Venezuela**
> > - **Puerto Rico**
> > - **South Korea**
>
> In the example above, the country entities predicted by GPT-4 are all objectively correct. However, only **United States** and **Venezuela** exist in NELL-V2. Moreover, the answer does not include the ground-truth entity `country: china`. Thus, although GPT-4 can rely on its broad pre-training knowledge to provide a wider range of reasonable answers, its hit rate within the NELL-V2 entity space is effectively zero, which is an unfair outcome for evaluation.
>
> In contrast, because KRLM’s next-entity prediction mechanism strictly constrains the model to predict only entities contained in NELL-V2, it ensures that no out-of-scope entities are produced, preserving the fairness of the evaluation. For the query `<sport: baseball, sport fans in country, ?>`, KRLM’s top-5 predictions are `country: u_s_`, `country: united_states`, `country: china`, `country: brazil`, and `country: england`. All of these entities exist within NELL-V2, and the ground-truth entity `country: china` is ranked third.
>
> **(II) Interpretable reasoning:** From the analysis of existing research directions, **interpretable KG reasoning models generally involve explicit multi-hop reasoning paths**, such as path exploration based on reinforcement learning and neuro-symbolic KG reasoning. However, **we apologize that the existing modeling mechanism of KRLM is unable to support its feedback explicit inference path**, which is also the direction we hope to improve in the future.
>
> **One promising direction is to use a structural-based KGFM as a unified relation tokenizer and employ a smaller LLM, fine-tuned to treat KG rules as new relation token-based language, as a rule generator**. The generated rule language can then be processed using a neuro-symbolic embedding model for lightweight fuzzy-logical reasoning. At present, we have built a primary rule generator prototype on WN18RR using Qwen2.5-0.5B, achieving an average accuracy of around 70% in rule generation. We believe this provides a promising path toward interpretable KG reasoning using LLM.
>
> If you are interested in this topic, we would be delighted to further discuss it!

---

### Official Review · Reviewer_dW1V · 2025-10-31

**Soundness:** 3
**Presentation:** 2
**Contribution:** 3
**Rating:** 4
**Confidence:** 4

**Summary:**

The paper proposes KRLM, a Knowledge Reasoning Language Model for inductive knowledge graph reasoning where entities/relations may be unseen. KRLM aims to coordinate intrinsic LLM knowledge with sparse KG context to mitigate knowledge distortion and reduce hallucinations. The method introduces (i) a KRL instruction format and a KRL tokenizer that map entities and relations into unified tokens aligned with LLM text representations; (ii) a KRL attention layer with a dynamic knowledge memory to balance LLM priors and KG evidence during reasoning; and (iii) a structure-aware next-entity predictor that constrains decoding to a trusted knowledge domain. Across 25 real-world inductive KGR datasets, KRLM is reported to outperform prior methods in both zero-shot and fine-tuned settings.

**Strengths:**

1.The paper focuses on an important pain point: leveraging LLM prior knowledge without letting sparse KG signals distort or override it, while also constraining generative hallucinations.

2.The KRL instruction + tokenizer provide a concrete alignment mechanism between symbolic KG elements and LLM token space, which is practical and reusable beyond a single dataset.

3.The KRL attention layer with dynamic memory is a sensible architectural step toward balancing textual priors and structured context, and the design is easy to ablate.

4.Results on 25 inductive benchmarks (zero-shot and fine-tuning) suggest robustness across datasets and settings, which strengthens external validity.

**Weaknesses:**

1.While the KRL interface and attention layer are coherent, the overall contribution currently reads as a well-engineered integration of known ingredients. The paper needs to sharpen what is theoretically or algorithmically new.

2.There is no efficiency study, yet the method introduces extra tokens (KRL), a memory mechanism, and constrained decoding. It is needed to provide per-query token counts, latency, and memory usage broken down by components, and compare to strong LLM-based KGR and KGFM baselines under matched budgets.

3.Ambiguity around the “dynamic knowledge memory.” It’s not clear how the memory is constructed, updated, and queried. How are conflicts between LLM priors and KG entries resolved?

4.The paper should specify how KRL embeddings, attention layers, and predictor parameters are initialized (random, vocabulary-tied, or from pretrained adapters), and whether the backbone LLM is fully fine-tuned, LoRA-adapted, or frozen. Include sensitivity to model scale and initialization choices.

5.The central claim, coordinated KRL reduces LLM knowledge distortion, needs direct measurement. It is needed to add controlled ablation studies that vary KG sparsity, conflicting triples, and noisy relations, reporting distortion/hallucination metrics.

**Questions:**

How exactly does the structure-aware predictor enforce structural constraints?

---

> ### Author Response · Authors · 2025-11-19
> **Response to Weakness 1**
>
> > **Weakness:** While the KRL interface and attention layer are coherent, the overall contribution currently reads as a well-engineered integration of known ingredients. The paper needs to sharpen what is theoretically or algorithmically new.
>
> **Response:** Thank you for the thoughtful comment. While KRLM integrates existing LLM and GNN components, our key contributions are not a simple combination of known techniques. Instead, KRLM proposes **a new coupling mechanism that is essential for unifying LLM knowledge and KG structural reasoning**. We highlight the core innovation as follows:
>
> **(I) Research motivation:** Our work is motivated by the need to mitigate **knowledge distortion** commonly seen in LLM-based KGR methods that rely on explicit KG-path prompts. Such models often treat limited-hop KG paths as if they were complete evidence, which may cause the LLM to overlook its broader pre-trained priors, leading to biased or incorrect reasoning, especially in scenarios where KG context is sparse. To address this challenge, our core idea is to modify the LLM architecture so that it can **query broader KG structural context** and **flexibly fuse KG context with LLM internal priors**, enabling autonomous coordination between external KG entries and the LLM intrinsic knowledge.
>
> **(II) Core algorithm contribution:** To enable this autonomous coordination, we propose a **knowledge memory-based KRL attention mechanism**. Building on the standard LLM architecture, we inject GNN-encoded entity structural embeddings into each self-attention layer as a trainable **knowledge memory**. During inference, the LLM simultaneously extracts instruction context and implicitly queries relevant KG structural information stored in this memory. The attention mechanism then performs a soft fusion between LLM priors and KG structural context, effectively reducing knowledge distortion during KGR.
>
> **(III) Theoretical Analysis:** We first present the modeling formulation of the $n$-th KRL attention layer:
>
> $\mathbf{H}^{(0)}=\mathbf{T}$,
>
> $\mathbf{A}=\text{softmax}(\frac{\mathbf{H}^{(n-1)}\mathbf{M}_{Q}^{(n)}\mathbf{E}_m^T||(\mathbf{H}^{(n-1)}\mathbf{W}_Q^{(n)}[\mathbf{H}^{(n-1)}\mathbf{W}_K^{(n)}]^T+\mathbf{MASK})}{\sqrt{F}})$,
>
> $\mathbf{H}^{(n)}=\mathbf{A}[\mathbf{E}_{m}\mathbf{M}_V^{(n)}||\mathbf{H}^{(n-1)}\mathbf{W}_V^{(n)}]$.
>
> Here, $\mathbf{T}\in\mathbb{R}^{L\times F}$ denotes the token embedding sequence of the KRL instruction. $\mathbf{W}_Q^{(n)}$, $\mathbf{W}_K^{(n)}$, and $\mathbf{W}_V^{(n)}$ are the pretrained weights of the LLM, and $\mathbf{MASK}$ is the causal attention mask. $\mathbf{E}_m$ is the knowledge memory composed of the KG embeddings of top-$K$ entities, $\mathbf{M}_Q^{(n)}$ and $\mathbf{M}_V^{(n)}$ are the encoding parameters of $\mathbf{E}_m$, enabling the knowledge memory to be deeply coupled with the LLM’s context learning of the KRL instruction.
>
> Since the prediction of the target entity is based on the last token hidden state $\mathbf{h}_L^{(n)}\in\mathbf{H}^{(n)}$, we rewrite the above formulation with $\mathbf{h}_L^{(n)}$ as the focus:
>
> $\mathbf{h}_L^{(n)}=\sum _{l=1} ^{L}\alpha _l\mathbf{h}_l^{(n-1)}\mathbf{W}_V^{(n)}+\sum _{k=1}^K\beta_k\mathbf{e}_k\mathbf{M}_V^{(n)}$,
>
> $\alpha_l=\frac{\exp(<\mathbf{h} _L ^{(n-1)}\mathbf{W} _Q ^{(n)};\mathbf{h} _l ^{(n-1)}\mathbf{W} _K ^{(n)}>)}{\sqrt{F}[\sum _{j=1} ^L\exp(<\mathbf{h} _L ^{(n-1)}\mathbf{W} _Q ^{(n)};\mathbf{h} _j ^{(n-1)}\mathbf{W} _K ^{(n)}>)+\sum _{k=1} ^K\exp(<\mathbf{h} _L ^{(n-1)}\mathbf{M} _Q ^{(n)};\mathbf{e} _k>)]}$, $\mathbf{e}_k\in\mathbf{E}_m$,
>
> $\beta_k=\frac{\exp(<\mathbf{h} _L ^{(n-1)}\mathbf{M} _Q ^{(n)};\mathbf{e} _k>)}{\sqrt{F}[\sum _{j=1} ^L\exp(<\mathbf{h} _L ^{(n-1)}\mathbf{W} _Q ^{(n)};\mathbf{h} _j ^{(n-1)}\mathbf{W} _K ^{(n)}>)+\sum _{z=1} ^K\exp(<\mathbf{h} _L ^{(n-1)}\mathbf{M} _Q ^{(n)};\mathbf{e} _z>)]}$, $\mathbf{e}_z\in\mathbf{E}_m$,
>
> From the perspective of contextual learning, $\alpha_l$ shows how $\mathbf{E}_m$ participates in scaling the pre-trained LLM’s attention allocation over the instruction context, thereby modulating the relative contribution of each contextual token to $\mathbf{h}_L^{(n)}$.
>
> Meanwhile, the term $\sum _{k=1}^K\beta_k\mathbf{e}_k\mathbf{M}_V^{(n)}$ serves as supplementary semantic information that is injected into the final representation, which functions similarly to a bias semantic correction derived from external KG knowledge.
>
> **This analysis theoretically explains how the KRL attention layer coordinates the structural knowledge in $\mathbf{E}_m$ with the pretrained priors in LLM: By allowing $\mathbf{E}_m$ to influence both the attention-weight scaling in contextual computation and the semantic correction of the aggregated representation, the model effectively couples external KG entries with LLM internal priors, thereby mitigating knowledge distortion.**
>
> A more detailed theoretical discussion is provided in **Appendix D** of the paper, hoping to address the reviewer's concerns!

---

> ### Author Response · Authors · 2025-11-19
> **Response to Weakness 2**
>
> > **Weakness:** There is no efficiency study. It is needed to provide per-query token counts, latency, and memory usage broken down by components, and compare to strong LLM-based KGR and KGFM baselines under matched budgets.
>
> **Response:** Thank you for your valuable comments. Below, we provide an efficiency comparison between our KRLM and existing LLM-based KGFM methods in terms of per-query token counts, latency, and memory usage.
>
> **(I) Per-query token counts:** The following table shows the average number of prompt/response tokens of MKGL and PROLINK, two latest LLM-based KGFMs (Llama2-7b as the backbone), on 25 inductive datasets in this paper, to directly reflect the token efficiency of our KRLM.
>
> |Model|Prompt tokens|Response Tokens|
> |-|-|-|
> |**KRLM (ours)**|118.75$\pm$5.14|1|
> |**MKGL**|114.10$\pm$4.38|1|
> |**PROLINK**|414.04$\pm$8.80|26.15$\pm$5.62|
>
> Because PROLINK requires extensive relational context to prompt the LLM in generating candidate target entity types, its total token counts are significantly higher than those of other methods.
>
> **KRLM adopts a compact word-level token vocabulary, which makes the KRL instruction prompt more concise**. Specifically, given a query triplet $(h, r, ?)$, we treat $h$ and $r$ as **new world-level atomic tokens** (i.e., `<Entity: name of h>` and `<Relation: name of r>`). A dedicated vocabulary is constructed to map each world-level token to its corresponding textual description (pre-trained tokens of LLM) and KG embedding. This design provides the LLM with a clear and consistent semantic context for learning these new world-level tokens.
>
> Meanwhile, KRLM reformulates KG reasoning as a **next-entity prediction task**. After feeding the KRL instruction into the LLM, KRLM obtains the hidden state of the final token `<Relation: name of r>` and computes its similarity with the world-level tokens of all candidate entities, analogous to LLM-style next-token prediction. The model then selects the candidate with the highest score as the reasoning output. Therefore, the number of response tokens from KRLM is always **exactly 1**.
>
> For clarity, we provide the general format of a KRL instruction below to facilitate reviewer’s understanding:
>
> **>> Instruction begin**
>
> **Instruction:** Define the word format for a new language as <Type: Text Description>. Suppose you are a linguistic expert who are learning this new language. Given the following vocabulary:
>
> |Word|Type|Text description|Knowledge representation|
> |-|-|-|-|
> |`<Entity: name of h>`|Entity|description of $h$|[KG embedding of $h$]|
> |`<Relation: name of r>`|Relation|description of $r$|[KG embedding of $r$]|
>
> Please complete the next word '?' in the given sentence:
>
> `<Entity: name of h>` `<Relation: name of r>`?
>
> **Response:** `<Entity: name of h>` `<Relation: name of r>`
>
> **>> Instruction end**
>
> The prompt format and reasoning paradigm of MKGL are similar to those of our KRLM. However, the vocabulary used in MKGL’s instructions does not include KG embedding information. Therefore, MKGL uses slightly fewer tokens than KRLM.
>
> **(II) Latency and memory usage in KG reasoning:** We selected two datasets with the highest (FB15k237) and lowest (NELL-V1) graph densities within our experimental scope as benchmarks. The following table reports the inference time (seconds per batch) and GPU memory consumption of both latest LLM-based (KRLM, MKGL, PROLINK) and structural learning-like (ULTRA, MOTIF, TRXI) KGFMs. For consistency, we set the test batch size to 16 and used Llama-2-7b as the backbone for all LLM-based KGFMs, conducting experiments on a single NVIDIA A100 GPU.
>
> **For KRLM and MKGL, their inference time differs by approximately one second between FB15k237 and NELL-v1, which are acceptable to humans**. However, PROLINK needs a long prompt to guide Llama2-7b to generate the potential target entity types of a query according to the relational context, which leads to it needs to spend longer inference time and larger memory on small-scale NELL-V1.
>
> | Dataset |KRLM (Ours)|MKGL|**PROLINK**|ULTRA|MOTIF|TRIX|
> |-|-|-|-|-|-|-|
> |FB15k237|2.2308±0.0329 [30.11GB]|1.9843±0.0411 [27.75GB]|The dataset is not provided by the preprocessing code|0.1436±0.0090 [2.6GB]|0.2508±0.0123 [2.63GB]|0.2212±0.00987 [2.6GB]|
> |NELL-V1|1.1817±0.0672 [29.32GB]|0.9916±0.0620 [26.82GB]|4.3536±0.0411 [36.42GB]|0.0129±0.0001 [2.5GB]|0.0211±0.0001 [2.5GB]|0.0132±0.0000 [2.5GB]|
>
> In addition, we have also counted the inference time of each component in KRLM in the table below. We found that the main module that affects the inference latency of KRLM on different scales of KGs is the KRL tokenizer, because it contains an ULTRA module, which needs to read the complete KG for structural context learning of entities and relations.
>
> | Dataset |KRL tokenizer|LLM with knowledge memory (KRL attention layers)|Next-entity predictor (Knowledge decoder)|
> |-|-|-|-|
> |FB15k237|1.2413±0.0179|1.1206±0.2408|0.0961±0.0200|
> |NELL-V1|0.0293±0.0019|1.0554±0.0535|0.0862±0.0047|

---

> ### Author Response · Authors · 2025-11-19
> **Response to Weakness 3**
>
> > **Weakness:** It’s not clear how the dynamic knowledge memory is constructed, updated, and queried. How are conflicts between LLM priors and KG entries resolved?
>
> **Response:** We apologize for causing confusion to you. The following is our detailed explanation of the dynamic knowledge memory. We hope these clarifications address your concerns:
>
> **(I) Overall Pipeline:** Given a KG $\mathcal{G}=(\mathcal{E}, \mathcal{R}, \mathcal{T})$ consisting of an entity set $\mathcal{E}$, a relation set $\mathcal{R}$, and a triplet set $\mathcal{T}$, as well as a query triplet $(e_h, r_q, ?)$, we first employ a GNN-based knowledge encoder (Eq. 1 in our paper) to obtain the structural embeddings for all relations and entities, denoted as $\mathbf{R}\in\mathbb{R}^{|\mathcal{R}|\times d}$ and $\mathbf{E}\in\mathbb{R}^{|\mathcal{E}|\times d}$, respectively.
>
> We then take the embedding $\mathbf{r}_q\in\mathbf{R}$ of relation $r_q$ and concatenate it with each entity embedding in $\mathbf{E}$; a linear layer is applied on top of the concatenation to compute the probability score of each entity being the correct target entity. Based on these scores, we select the top-$K$ entities and concatenate their structural embeddings to form a memory matrix $\mathbf{E}_m\in\mathbb{R}^{K\times d}$. This memory is fed into the KRL attention layer to support contextual learning over the KRL instruction. In particular, the operation of the $n$-th KRL attention layer over the memory is as follows (excerpted from Eq. 6 in the paper):
>
> $\mathbf{H}^{(0)}=\mathbf{T}$,
>
> $\mathbf{A}=\text{softmax}(\frac{\mathbf{H}^{(n-1)}\mathbf{M}_{Q}^{(n)}\mathbf{E}_m^T||(\mathbf{H}^{(n-1)}\mathbf{W}_Q^{(n)}[\mathbf{H}^{(n-1)}\mathbf{W}_K^{(n)}]^T+\mathbf{MASK})}{\sqrt{F}})$,
>
> $\mathbf{H}^{(n)}=\mathbf{A}[\mathbf{E}_{m}\mathbf{M}_V^{(n)}||\mathbf{H}^{(n-1)}\mathbf{W}_V^{(n)}]$.
>
> Here, $\mathbf{T}$ denotes the token embedding sequence of the KRL instruction. $\mathbf{W}_Q^{(n)}$, $\mathbf{W}_K^{(n)}$, and $\mathbf{W}_V^{(n)}$ are the pretrained weights of the LLM, and $\mathbf{MASK}$ is the causal attention mask. $\mathbf{M}_Q^{(n)}$ and $\mathbf{M}_V^{(n)}$ are the encoding parameters of $\mathbf{E}_m$, enabling the knowledge memory to be deeply coupled with the LLM’s context learning of the KRL instruction.
>
> **(II) Construction and Update:** As described in **Point (I)**, our knowledge memory is formed by the structural embeddings of the top-$K$ entities. During training, these top-$K$ entities change as the parameters of the knowledge encoder and scoring module are updated. Consequently, the evolution of the knowledge memory is closely tied to the optimization of these components and the overall model loss.
>
> **(III) Memory query mechanism:** The memory querying process is analogous to self-attention. Referring to the self-attention weight computation ($\mathbf{H}^{(n-1)}\mathbf{W}_Q^{(n)}[\mathbf{H}^{(n-1)}\mathbf{W}_K^{(n)}]^T$) in the above formula, the term
>
> $\mathbf{H}^{(n-1)}\mathbf{M}_{Q}^{(n)}\mathbf{E}_m^T$
>
> can be regarded as computing the **query–key attention scores between the KRL instruction and the embeddings in the knowledge memory**. These memory attention weights, together with the self-attention weights of the KRL instruction, are jointly normalized and then used to weight the value storage composed of both the memory and the KRL instruction. The final weighted aggregation yields the query result as $\mathbf{H}^{(n)}$.
>
> **(IV) How are conflicts between LLM priors and KG entries resolved:** We highlight the advantages of our approach to mitigating KG–LLM knowledge conflicts by comparing KRLM with existing LLM-based KGFMs that rely on explicit KG retrieval.
>
> - **Deeply coupled implicit retrieval in the knowledge memory.** As discussed in **Point (III)**, KRLM injects entities’ structural embeddings directly into the LLM’s attention layers. **This trainable attention-based implicit retrieval mechanism enables the model to autonomously coordinate KG memory with the pretrained LLM priors**, thereby alleviating the conflict.
> - **Knowledge memory construction.** KRLM uses multi-layer GNNs to encode entity structural embeddings as the retrieved memory. This allows the knowledge memory to store richer KG contextual semantics, providing broader and more expressive evidence for the above coordination of the KG memory and LLM priors.
> - In contrast, **existing LLM-based KGFMs with explicit retrieval** (e.g., PROLIKN [1]) inject limited-hop KG paths into the LLM purely in textual form. This design ① restricts the depth of KG context available to the model, and ② forces structural knowledge to be interpreted under the pretrained textual semantics, which often misaligns with the inherent structural representations of KG paths, ultimately leading to the *knowledge distortion* described in our introduction.
>
> [1] Wang K, Xu Y, Wu Z, et al. LLM as Prompter: Low-resource Inductive Reasoning on Arbitrary Knowledge Graphs. Findings of ACL 2024.

---

> ### Author Response · Authors · 2025-11-19
> **Response to Weakness 4**
>
> > **Weakness:** The paper should specify how KRL embeddings, attention layers, and predictor parameters are initialized (random, vocabulary-tied, or from pretrained adapters), and whether the backbone LLM is fully fine-tuned, LoRA-adapted, or frozen. Include sensitivity to model scale and initialization choices.
>
> **Response:** We apologize for the ambiguous model initialization declaration. The following are the detailed initialization settings of different components in KRLM.
>
> **(I) Embedding of a KRL instruciton:** As illustrated in **Response to Weakness 2**, a KRL instruction consists of the LLM pretrained tokens, word-level tokens for entities and relations, and KG representations.
>
> - **LLM pretrained tokens**. We initialize the embeddings of these tokens by directly looking up the LLM’s pre-trained embedding table, which provides $F$-dimensional vectors.
> - **Word-level tokens for entities/relations**. Following Section 4.1 in our paper, an entity `Michael Jackson` is first converted into the string *<Entity: Michael Jackson>*. This string is then tokenized using the LLM’s tokenizer, and the corresponding embedding sequence is projected to a $d$-dimensional space using **a trainable matrix**. We then apply the Principal Attribute Aggregation (PAA) pooling to this sequence. The pooled results are concatenated and further encoded by **another trainable matrix**, producing the final word-level token embedding for the entity. **Both the trainable matrices are initialized using the default settings of pytorch.nn.Linear()**.
> - **KG representations for entities and relations** are generated by the GNN-based knowledge encoder described in **Point (I) of Response to Weakness 3**. The encoder includes a 6-layer relation GNN and a 6-layer entity GNN. Each GNN layer consists of a non-parameter DistMult message-passing module, a sum aggregation function, and **a trainable linear updating function implemented via pytorch.nn.Linear with default initialization**. The relation GNN additionally maintains **an edge-embedding table initialized by the default settings of pytorch.nn.Embedding**. The encoder also includes **a trainable scoring function (implemented using pytorch.nn.Linear)** that selects the top-50 candidate entities to generate a knowledge memory.
>
> **(II) Initialization of the KRL attention layer:** A KRL attention layer consists of the backbone LLM’s self-attention module and our proposed knowledge memory, with the detailed formulation provided in the equations in **Response to Weakness 3**. Specifically, **the self-attention parameters ($\mathbf{W}_Q^{(n)}$, $\mathbf{W}_K^{(n)}$, and $\mathbf{W}_V^{(n)}$) are initialized from the pretrained LLM and kept frozen during training.** **The knowledge memory $\mathbf{E}_m$ is constructed from the entity GNN embeddings** produced by the knowledge encoder. To enable $\mathbf{E}_m$ to deeply couple in the context learning for KRL instructions, **we introduce two trainable projection matrices, $\mathbf{M}_Q^{(n)}$ and $\mathbf{M}_V^{(n)}$, implemented using the default initialization via torch.nn.Linear**.
>
> **(III) Initialization of the next-entity predictor:** The predictor consists of a stacked PAA module and an entity-GNN module, both initialized in the same way in **(I) Embedding of a KRL instruction**. In addition, **we define a scoring function using the default initialization of torch.nn.Linear**, which is used to estimate the score of each candidate target entity.
>
> **(IV) Experimental analysis of model scale and initialization choices:** As shown in the above initialization details of each module, only the LLM backbone is initialized with the pretrained weights. All other trainable components follow the default initialization schemes of Pytorch. Ablation studies and parameter-sensitivity analyses regarding the PAA module, the GNN-based knowledge encoder/entity predictor, and the scale of the knowledge memory are reported in Table 2 and Figure 4 of our paper. Overall, the findings show that: ① the PAA module achieves significant performance gains over other pooling methods; ② using 6 GNN layers yields the best results; ③ a knowledge-memory size of 50 is optimal.
>
> In addition, **under the unified training of the knowledge memory component**, we conduct further studies on different training strategies for the LLM backbone, including frozen (our KRLM), LoRA-adapted ($r$=32, $\alpha$=16, dropout=0.05, and applied to *q_proj* and *v_proj*), and fully fine-tuned variants. All three variants are trained end-to-end on four lightweight datasets with a batch size of 4 per GPU × 4 A100 GPUs. Their Hits@10 performance and the trainable parameters are summarized in the table below.
>
> |Dataset|Frozen (our KRLM) [18.49M]|LoRA-adapted [35.28M]|Fully fine-tuning [6.74B]|
> |-|-|-|-|
> |FB-V1|.705|.708|OOM|
> |WN-V1|.801|.811|OOM|
> |NL-0|.591|.588|OOM|
> |NL-100|.688|.686|OOM|
>
> Clearly, keeping the LLM backbone frozen during training offers the best cost-effectiveness.

---

> ### Author Response · Authors · 2025-11-19
> **Response to Weakness 5:  Distortion/Hallucination metrics definition and (I) Supplementary experiments on sparse KGs**
>
> > **Weakness:** The central claim, coordinated KRL reduces LLM knowledge distortion, needs direct measurement. It is needed to add controlled ablation studies that vary KG sparsity, conflicting triples, and noisy relations, reporting distortion/hallucination metrics.
>
> **Response:** Thank you for your insightful suggestion. Let's first define the distortion and hallucination metrics clearly:
>
> **Distortion rate (DR):** This metric is used to measure the misjudgment rate of the model before and after changes in KG structure, reflecting the model's ability to autonomously coordinate with KG context. For a query triplet $q=(h, r, ?)\in\mathcal{T}$, let $t$ be the ground truth. Suppose the model assigns a ranking score $s_1^{(q)}$ to $t$ on a clean KG and a score $s_2^{(q)}$ on a noisy KG. If $s_2^{(q)}>s_1^{(q)}$, the distortion rate for this query is recorded as $s_2^{(q)}-s_1^{(q)}$. The overall DR of the model on the noisy KG is given by $\frac{\sum_{q\in\mathcal{T}}\max(0,s_2^{(q)}-s_1^{(q)})}{|\mathcal{T}|}$, with lower values indicating better performance.
>
> **Hallucination rate (HR):** This metric measures the proportion of LLM-based KGFMs to generate out-of-scope answers. For a query triplet $q=(h, r, ?)\in\mathcal{T}$, let $S_q$ denote the candidate set of target entities. A hallucination is considered to occur when the result $p_q$ generated by the LLM satisfies $p_q\notin S_q$. The overall HR of the model on a KG is defined as:
>
> $\frac{\sum_{q\in\mathcal{T}}\mathbb{I}(p_q\notin S_q)}{|\mathcal{T}|}$,
>
> where a lower value indicates better performance.
>
> It should be noted that, since the LLM-based KGFM under comparison (PROLINK) uses a LLM to predict the type of the target entity to guide ULTRA’s reasoning over the KG, for PROLINK, $S_q$ refers to the set of entity types. During evaluation, we consider two modes: Relaxed (**HR_R**) and Strict (**HR_S**). In the Relaxed mode, $S_q$ includes all entity types in the KG, while in the Strict mode, $S_q$ is restricted to the target entity types that are directly obtainable for relation $r$ in the background KG.
>
> Based on the above metrics, we evaluate the model from three perspectives: **KG sparsity, conflicting triplets, and noisy relations**:
>
> **(I) Sparse KG reasoning:** We have collected the graph densities ($\frac{\text{triplet number}}{\text{entity number}\times(\text{entity number}-1)}$) of all KG datasets used in our experiments. Among them, **FB15k-237** exhibits the highest density of $1.29\times10^{-3}$. To further examine the behavior of our KRLM on sparse KGs, we additionally collected **FB15k237_10** [1], a sparse dataset extracted from FB15k237, with the density of $2.11\times10^{-4}$. **Then, we test the query triplets of FB15k237_10 using the background KGs of FB15k237 and FB15k237_10, respectively**. The table below reports the performance of structural learning-based (ULTRA) and LLM-based (PROLINK) KGFMs under the pre-trained setting.
>
> |FB15k237_10 testing triplets under different background KGs|ULTRA|PROLINK (Llama2-7b as 10-shot prompt backbone)|KRLM|
> |-|-|-|-|
> |FB15k-237|MRR: 0.469; Hits@10: 0.668; **HR_R: 0.00**; **HR_S: 0.00**|MRR: 0.471; Hits@10: 0.668; HR_R: 0.006; HR_S: 0.012|MRR: 0.479; Hits@10: 0.665; **HR_R: 0.00**; **HR_S: 0.00**|
> |FB15k237_10|MRR: 0.248; Hits@10: 0.398; DR: 471.42; **HR_R: 0.00**; **HR_S: 0.00**|MRR: 0.238; Hits@10: 0.383; DR: 612.78; HR_R: 0.002; HR_S: 0.008|MRR: 0.243; Hits@10: 0.409; **DR: 297.01**; **HR_R: 0.00**; **HR_S: 0.00**|
>
> Evidently, sparse KGs significantly constrain the reasoning of models due to the limited contextual evidence they can provide, leading to failures on query triplets that would otherwise be manageable. In this scenario, the structural learning capability of GNN modules becomes particularly crucial, enabling ULTRA and KRLM to capture implicit structural contexts in sparse KGs and thereby mitigate reasoning errors. In contrast, PROLINK’s explicit prompt-based contextual learning mechanism struggles to extract information highly relevant to the ground truth from the limited number of available KG paths. Moreover, in sparse KGs, PROLINK tends to reduce the out-of-scope answers, which we attribute to the constrained KG context helping to focus the LLM’s reasoning process, allowing it to rely solely on limited prompt evidence when generating answers.
>
> [1] Lv X, Han X, Hou L, et al. Dynamic Anticipation and Completion for Multi-Hop Reasoning over Sparse Knowledge Graph. EMNLP. 2020: 5694-5703.

---

> ### Author Response · Authors · 2025-11-19
> **Response to Weakness 5 (II) Supplementary experiments on the conflicting triplets**
>
> **(II) Reasoning on conflicting triplets:** We expand conflicting triplets in NELL-V1. The specific construction process is as follows: ① For each $(h, r)$ in the complete KG, obtain the set of tail entities $T_{hr}$, and for each relation $r$, obtain the set of tail entities $T_r$; ② Randomly select $k\\%$ of the correct triplets from the background KG $\mathcal{G}=(\mathcal{E},\mathcal{R},\mathcal{T})$ to be modified into conflicting triplets; ③ For each selected correct triplet $(h,r,t) \in \mathcal{T}$, randomly select an entity $c$ from the set $T_r[r] - T_{hr}[(h,r)]$ to construct a conflicting triplet $(h,r,c)$; ④ Expand the original set $\mathcal{T}$ by adding the conflicting triplets, resulting in $\mathcal{T}^{'}$, thereby obtaining the background KG $\mathcal{G}^{'}=(\mathcal{E},\mathcal{R},\mathcal{T}^{'})$ for reasoning. An example of a conflicting triplet is as follows:
>
> Correct triplet: `<TV station: KETS, company economicsector, academic field: media>`
>
> Conflicting triplet: `<TV station: KETS, company economicsector, academic field: news>`
>
> This construction method ensures that the conflicting entity $c$ is unrelated to $(h, r)$ while still conforming to the tail entity type of relation $r$, thereby creating conflicting triplets that are semantically valid but factually incorrect to the greatest extent. The table below records the performance of ULTRA, PROLINK, and our KRLM in the scenario of conflicting triplets.
>
> |NELL-V1 testing triplets under different conflicting triplet rate ($k\\%$)|ULTRA|PROLINK (Llama2-7b as 10-shot prompt backbone)|KRLM|
> |-|-|-|-|
> |$k=0$|MRR: 0.785; Hits@10: 0.913; HR_R: 0.00; HR_S: 0.00|MRR: 0.726; Hits@10: 0.883; HR_R: 0.005; HR_S: 0.008|MRR: 0.652; Hits@10: 0.887; HR_R: 0.00; HR_S: 0.00|
> |$k=10$|MRR: 0.665; Hits@10: 908; DR: 10.22; **HR_R: 0.00**; **HR_S: 0.00**|MRR: 0.626; Hits@10: 0.870; DR: 8.05; HR_R: 0.005; HR_S: 0.009|MRR: 0.641; Hits@10: 0.882; **DR: 2.2**; **HR_R: 0.00**; **HR_S: 0.00**|
> |$k=50$|MRR: 0.629; Hits@10: 0.8814; DR: 16.23; **HR_R: 0.00**; **HR_S: 0.00**|MRR: 0.413; Hits@10: 0.641; DR: 24.52; HR_R: 0.008; HR_S: 0.010|MRR: 0.583; Hits@10: 0.875; **DR: 14.39**; **HR_R: 0.00**; **HR_S: 0.00**|
>
> The experimental results indicate that the more conflicting triplets a KG contains, the greater the constraints it imposes on the model's reasoning. Specifically, in PROLINK, where explicit KG paths are used for prompting, conflicting triplets directly interfere with the LLM's reasoning, leading to a sharp increase in DR. In contrast, the generalizable structural representations learned by ULTRA demonstrate stronger resistance to conflicting triplets compared to explicit LLM prompting. Moreover, our KRLM incorporates both these generalizable structural representations and LLM priors within the KRL attention layer, thereby mitigating the knowledge distortion. Additionally, we observed that the reasoning hallucinations in PROLINK do not fluctuate drastically with the proportion of conflicting triplets. We attribute this to the fact that the constructed conflicting triplets still conform to the head and tail entity types required by the relations.

---

> > ### Author Response · Authors · 2025-11-19
> > **Response to Weakness 5 (III) Supplementary experiments on the noisy relations**
> >
> > **(III) Reasoning on noisy relations:** We inject noisy relations into NELL-V1 through the following operations: ① For each $(h, t)$ in the full KG, we obtain the corresponding relation set $R_{ht}$; ② From the background KG $\mathcal{G}=(\mathcal{E},\mathcal{R},\mathcal{T})$, we randomly select $k\\%$ of the correct triplets to undergo relational noising; ③ For each selected triplet $(h,r,t)\in\mathcal{T}$, we randomly choose a relation $r^{'}$ from the set $\mathcal{R}-R_{ht}[(h,t)]$ to construct a noisy triplet $(h,r^{'},t)$; ④ The constructed noisy triplets are added to $\mathcal{T}$ to form $\mathcal{T}^{'}$, thereby obtaining the background KG $\mathcal{G}^{'}=(\mathcal{E},\mathcal{R},\mathcal{T}^{'})$ for inference. An example of a noisy relational triplet is as follows:
> >
> > Correct triplet: `<TV station: WGCU, subpart of, company: PBS>`
> >
> > Conflicting triplet: `<TV station: WGCU, organization terminated person, company: PBS>`
> >
> > This construction ensures that the noisy relation $r^{'}$ has no semantic association with $(h, t)$. The table below records the performance of ULTRA, PROLINK, and our KRLM in the presence of relation noise.
> >
> >
> > |NELL-V1 testing triplets under different noisy relation rate ($k\\%$)|ULTRA|PROLINK (Llama2-7b as 10-shot prompt backbone)|KRLM|
> > |-|-|-|-|
> > |$k=0$|MRR: 0.785; Hits@10: 0.913; HR_R: 0.00; HR_S: 0.00|MRR: 0.726; Hits@10: 0.883; HR_R: 0.005; HR_S: 0.008|MRR: 0.652; Hits@10: 0.887; HR_R: 0.00; HR_S: 0.00|
> > |$k=10$|MRR: 0.539; Hits@10: 0.716; DR: 19.94; **HR_R: 0.00**; **HR_S: 0.00**|MRR: 0.515; Hits@10: 0.771; DR: 14.15; HR_R: 0.005; HR_S: 0.085|MRR: 0.576; Hits@10: 0.758; **DR: 14.0**; **HR_R: 0.00**; **HR_S: 0.00**|
> > |$k=50$|MRR: 0.507; Hits@10: 0.637; DR: 22.49; **HR_R: 0.00**; **HR_S: 0.00**|MRR: 0.483; Hits@10: 0.650; DR: 20.08; HR_R: 0.006; HR_S: 0.110|MRR: 0.533; Hits@10: 0.741; **DR: 17.40**; **HR_R: 0.00**; **HR_S: 0.00**|
> >
> > Obviously, as the relation noise rate increases, the reasoning performance of the model declines significantly. Moreover, compared to the results of **(II) Reasoning on conflicting triplets**, relation noise has a more pronounced adverse impact on models such as ULTRA and KRLM, which utilize GNNs for relational structure learning. This is because the noise disrupts the relational context that these models induce, at the semantic level. As for PROLINK, relation noise interferes with its prompt-based LLM reasoning in a manner similar to how conflicting triplets do. Furthermore, since the entity types associated with noisy relations fundamentally contradict real-world knowledge, the resulting HR_S is significantly higher than PROLINK’s metrics in scenarios involving sparse knowledge graphs and conflicting triplets.

---

> ### Author Response · Authors · 2025-11-19
> **Response to Question 1**
>
> > **Question:** How exactly does the structure-aware predictor enforce structural constraints?
>
> **Response:** Thank you for your comment. Below is our explanation of how the structure-aware predictor operates:
>
> Our method uses the hidden state of the last token (`<Relation: name of relation>`) in the KRL instruction to predict the word-level token (`<Entity: name of entity>`) of the target entity. This converts KG reasoning into an LLM-style next-token prediction, i.e., *next-entity prediction*. **This design avoids the risk of generating out-of-scope entities commonly observed in existing LLM-based KGR models** (refer to the HR metric in **Response to Weakness 5**). The structural constraints of our approach are reflected in two aspects:
>
> **(I) Entity-space constraint:** Most prior LLM-based KGR methods inherit the LLM’s next-token prediction mechanism, generating entities as sequences of vocabulary tokens. Since the LLM vocabulary (e.g., Llama2-7B has 32k tokens) is typically much larger than the number of entities in a KG benchmark and an entity name may require multiple tokens, LLMs may generate the textual name of an entity that falls outside the gold entity set. (This does not necessarily mean the generated entity is factually wrong, but it makes evaluation unfair.)
>
> To address this, KRLM aggregates the MLP head $\mathbf{P}\in\mathbb{R}^{4096\times 32000}$ in the next-token predictor of Llama2-7b into a compressed one $\mathbf{P}\in\mathbb{R}^{4096\times |\mathcal{E}|}$, whose size matches the KG’s entity set $\mathcal{E}$. The hidden state of the last KRL token is then compared with this compressed MLP head to select the top-1 target entity. This guarantees that predictions always lie within the entity set and therefore remain evaluable.
>
> **(II) Structural context constraint:** Under the **entity-space constraint**, the compressed MLP head stores each target entity’s word-level embedding, allowing basic in-domain entity prediction. However, we further want the MLP head to incorporate the KG structural context of a given query triplet $(h,r,?)$ to assist in model prediction.
>
> Consequently, as described in Eq. 8 in our paper, we feed the word-level embedding of the head entity $h$ into NBFNet [1], a GNN-based KG encoder, to propagate messages over the KG and obtain contextual embeddings for all entities. These embeddings form an $h$-specific MLP head, which is then used for predicting the target entity. To verify its effectiveness, we include the “-KDe” ablation in Table 2 in our paper, which demonstrates that adding **structural-context constraint** significantly outperforms using only **entity-space constraint**.
>
> [1] Zhu Z, Zhang Z, Xhonneux L P, et al. Neural bellman-ford networks: A general graph neural network framework for link prediction. NeurIPS, 2021: 29476-29490.

---

### Official Review · Reviewer_WYHS · 2025-11-10

**Soundness:** 4
**Presentation:** 3
**Contribution:** 3
**Rating:** 6
**Confidence:** 3

**Summary:**

This paper introduces the KRLMl, a novel framework designed to address the problem of knowledge distortion in LLM-based KGR models. Specifically, KRLM aims to integrate the inherent knowledge of LLMs with the structural context of KGs to improve the accuracy and reliability of fact reasoning in open-domain KGs. The proposed model shows strong performance in both zero-shot and fine-tuned inductive KGR tasks, outperforming existing models in several benchmark datasets.

**Strengths:**

1. The paper proposes a sophisticated mechanism for integrating the structural knowledge of KGs with the intrinsic knowledge of LLMs.
2. The experiment results indicate KRLM's superiority in both zero-shot reasoning and fine-tuning scenarios, outperforming several state-of-the-art models.
3. The model is well-grounded in both theory and practice, with clear explanations of the new modules

**Weaknesses:**

1. The reasoning complexity of KRLM, particularly during fine-tuning and inference, is a significant concern. While the model shows excellent performance, the computational overhead may limit its practical deployment in large-scale real-time applications. The authors briefly mention the computational complexity but do not provide enough detail.
2. The paper does not sufficiently address how the model would perform or be adapted to environments with limited computational resources or sparse KGs.
3. Tranditional knowledge graph reasoning methods need to be dicussed.

**Questions:**

The paper assumes that the knowledge memory will contain relevant information for reasoning. However, in cases where the relevant entities or relations are not part of the memory, will the model's performance dramatically drops?

---

> ### Author Response · Authors · 2025-11-18
> **Response to Weakness 1**
>
> > **Weakness:** The reasoning complexity of KRLM, particularly during fine-tuning and inference, is a significant concern. While the model shows excellent performance, the computational overhead may limit its practical deployment in large-scale real-time applications. The authors briefly mention the computational complexity but do not provide enough detail.
>
> **Response:** Thank you for your insightful suggestion. According to this comment, we carry out more detailed experimental statistics on the basis of theoretical complexity analysis, including inference and training costs. The specific statistical analysis is as follows:
>
> **(1) Inference cost analysis:** We selected two datasets with the highest (FB15k237) and lowest (NELL-V1) graph densities within our experimental scope as benchmarks and included MKGL and PROLINK, the latest LLM-based KGFMs, as comparative baselines. The following table reports the inference time of both LLM-based (KRLM, MKGL, PROLINK) and ULTRA-like (ULTRA, MOTIF, TRXI) KGFMs. For consistency, we set the test batch size to 16 and used Llama-2-7b as the backbone for all LLM-based KGFMs, conducting experiments on a single NVIDIA A100 GPU.
>
> ULTRA-like KGFMs require loading the entire KG as the source for inference, while LLM-based KGFMs follow the ULTRA+LLM hybrid framework. Consequently, all publicly accessible KGFMs we used are inevitably affected by the scale of the underlying KG. In addition, since the original PROLINK paper does not release data-processing scripts for FB15k237, we only counted its inference time on NELL-V1.
>
> The table below reports the detailed inference costs, including inference time (seconds per batch) and GPU memory consumption. As noted in the reviewer’s comment, existing KGFMs exhibit sensitivity to the KG size. **For KRLM and MKGL, their inference time differs by approximately one second between FB15k237 and NELL-v1, which are acceptable to humans**. However, PROLINK needs a long prompt to guide Llama2-7b to generate the potential target entity types of a query according to the relational context, which leads to it needing to spend a longer inference time and larger memory on small-scale NELL-V1.
>
> | Dataset |KRLM (Ours)|MKGL|**PROLINK**|ULTRA|MOTIF|TRIX|
> |-|-|-|-|-|-|-|
> |FB15k237 (272115 triplets)|2.2308±0.0329 [30.11GB]|1.9843±0.0411 [27.75GB]|-|0.1436±0.0090 [2.6GB]|0.2508±0.0123 [2.63GB]|0.2212±0.00987 [2.6GB]|
> |NELL-V1 (833 triplets)|1.1817±0.0672 [29.32GB]|0.9916±0.0620 [26.82GB]|4.3536±0.0411 [36.42GB]|0.0129±0.0001 [2.5GB]|0.0211±0.0001 [2.5GB]|0.0132±0.0000 [2.5GB]|
>
> **(2) Training cost analysis:** We calculated the trainable parameters of MKGL and our KRLM, as well as the training time on the FB15k237 dataset with a uniform batch of 4 per GPU. The statistical results are shown in the following table.
>
> KRLM requires embedding GNN in the tokenizer and next-token predictor of LLM, which slightly increases the parameters. However, it is consistent with MKGL in the main fine-tuning parameters of LLM (KRL attention layer V.S. LoRA). To ensure generalization, KRLM requires additional cost to construct a relational graph for real-time perturbed KGs in each batch, resulting in a training time of about 12 minutes longer per epoch than MKGL.
>
> |Model (Llama2-7b as backbone)|trainable parameters|training time per epoch|
> |-|-|-|
> |MKGL|18 M (16.78 M for LoRA)|1 h 8 min / 4 X A100 GPU|
> |KRLM (Ours)| 18.49 M (16.78 M for the knowledge memory in the KRL attention layer) |1 h 20 min / 4 X A100 GPU|
>
> **Although KRLM incurs additional training costs, it offers substantially stronger generalization compared to MKGL**. Specifically, KRLM requires only a single pre-training phase on a large-scale KG, after which it can perform **training-free zero-shot reasoning** on entirely new KGs (**refer to KRLM (PT) in Tables 1, 12, and 13 in our submitted paper**). In contrast, MKGL is not a fully generalizable KGFM in the strict sense. While it can effectively recognize unseen entities within each inductive dataset, it cannot transfer zero-shot across different inductive datasets. Consequently, MKGL must be retrained for every new dataset, which significantly increases its deployment overhead.

---

> ### Author Response · Authors · 2025-11-18
> **Response to Weakness 2**
>
> > **Weakness:** The paper does not sufficiently address how the model would perform or be adapted to environments with limited computational resources or sparse KGs.
>
> **Response:** This is a constructive comment. The following are our replies from the perspective of sparse KG reasoning and limited computing resource scenarios:
>
> **(1) Sparse KG reasoning:** We have collected the graph densities of all 28 KG benchmarks used in our experiments, using the standard formula $\frac{\text{triplet number}}{\text{entity number}\times(\text{entity number}-1)}$. Among them, **FB15k-237**, which is adopted for pre-training, exhibits the highest density of $1.29\times10^{-3}$. The densities of the remaining datasets mainly fall within the range of $10^{-4}$ to $10^{-5}$, indicating that they are overall much sparser than FB15k-237.
>
> To further examine the behavior of **our KRLM** on sparse KGs, we additionally collected three datasets [1], FB15k237_10, FB15k237_20, and FB15k237_50, with sparsity levels on the order of $10^{-4}$. The table below reports the performance of structural learning-based (ULTRA, MOTIF, TRIX) and LLM-based (MKGL, PROLINK) KGFMs under the pre-trained (PT) zero-shot inference setting.
>
> |Dataset (density)|Supervised SOTA (data statistics from ULRAT [2])|ULTRA (PT)|MOTIF (PT)|TRIX (PT)|MKGL|PROLINK (Llama2-7b as 10-shot prompt backbone)|KRLM (PT)|
> |-|-|-|-|-|-|-|-|
> |FB15k237_10 ($2.11\times10^{-4}$)|MRR: 0.219; Hits@10: 0.337|**MRR: 0.248**; Hits@10: 0.398|MRR: 0.236; Hits@10: 0.384|MRR: 0.246; Hits@10: 0.393|-|MRR: 0.238; Hits@10: 0.383|MRR: 0.243; **Hits@10: 0.409**|
> |FB15k237_20 ($3.14\times10^{-4}$)|MRR: 0.247; Hits@10: 0.391|**MRR: 0.272**; **Hits@10: 0.436**|MRR: 0.259; Hits@10: 0.422|MRR: 0.269; Hits@10: 0.430|-|MRR: 0.262; Hits@10: 0.404|MRR: 0.269; Hits@10: 0.424|
> |FB15k237_50 ($6.79\times10^{-4}$)|MRR: 0.293; Hits@10: 0.458|MRR: 0.324; Hits@10: 0.526|MRR: 0.312; Hits@10: 0.508|MRR: 0.321; Hits@10: 0.521|-|MRR: 0.324; Hits@10: 0.529|**MRR: 0.328**; **Hits@10: 0.526**|
> |FB15k-237 ($1.29\times10^{-3}$)|**MRR: 0.415**; **Hits@10: 0.599**|MRR: 0.368; Hits@10: 0.564|MRR: 0.357; Hits@10: 0.550|MRR: 0.366; Hits@10: 0.559|MRR: 0.410; Hits@10: 0.591|-|MRR: 0.381; Hits@10: 0.554|
>
> Overall, **existing KGFM models perform significantly better than supervised SOTA KG reasoning models on sparse KGs**, but they do not show clear advantages on dense ones. We attribute this to the relational GNN module in KGFM (Eq. 1 in our paper), which is able to induce more generalizable structural semantics from the KG and thus provides additional information for reasoning over sparse KGs. After injecting the inherent knowledge of LLMs, LLM-based KGFMs can further supply dense semantic support to sparse KGs, leading to additional performance gains. Moreover, the strong performance of our KRLM on sparse inductive datasets (e.g., the WN-vX, NL-X, and WK-X series shown in **Tables 12 and 13** in our paper) further demonstrates KRLM’s potential for handling sparse KGs.
>
> [1] Lv X, Han X, Hou L, et al. Dynamic Anticipation and Completion for Multi-Hop Reasoning over Sparse Knowledge Graph. EMNLP. 2020: 5694-5703.
>
> [2] Galkin M, Yuan X, Mostafa H, et al. Towards Foundation Models for Knowledge Graph Reasoning. ICLR 2024.
>
> **(2) Limited computing resource scenarios:** Computational constraints are a widespread issue in LLM-based methods, especially when fine-tuning is required. However, according to our computational cost analysis reported in **Response to Weakness 1**, although KRLM requires four 40GB A100 GPUs during pre-training, its strong zero-shot reasoning ability enables it to perform **training-free KG reasoning** of general-scale KG within 30GB memory. Compared with MKGL, the LLM-based KGFM baseline in the paper, which must be retrained on each dataset, **our KRLM exhibits a clear advantage in low-resource model transferability**.
>
> That said, alleviating compute bottlenecks has been one of our recent research priorities. One promising direction is to use ULTRA as a relation tokenizer and employ a smaller LLM, fine-tuned to treat relation embeddings as atomic tokens, as a rule generator. The generated candidate KG rules can then be processed using a neuro-symbolic embedding model for lightweight fuzzy-logical reasoning. At present, we have built a primary rule generator prototype on WN18RR using Qwen2.5-0.5B, achieving an average accuracy of around 70% in rule generation. We believe this provides a promising path toward mitigating the compute bottlenecks faced by LLMs in large-scale KG reasoning.
>
> If you are interested in this topic, we would be delighted to further discuss it!

---

> ### Author Response · Authors · 2025-11-18
> **Response to Weakness 3**
>
> > **Weakness:** Tranditional knowledge graph reasoning methods need to be dicussed.
>
> **Response:** We apologize for any confusion we have caused the reviewer. In fact, the **supervised SOTA** results reported in **Tables 1, 12, and 13** in our paper refer to traditional KG embedding methods (e.g., A*Net [1], NBFNet [2], RED-GNN [3], and InGram [4]) that require training **from scratch on each dataset**. Since we use a diverse set of datasets and the strongest traditional method varies across them, we follow the practice in the ULTRA manuscript [5] by aggregating these results and reporting them in **Tables 8 and 9**. For clarity, we collectively refer to these best-performing traditional approaches as **supervised SOTA** when comparing them with KGFMs.
>
> In the second paragraph of **Appendix H.1**, we have further analyzed why these traditional supervised methods struggle to outperform KGFMs on inductive datasets, discussing their limitations in generalization and emphasizing the advantages of KGFM.
>
> We apologize for any confusion caused, and we hope the above explanation addresses your concerns. Thank you for your support and understanding!
>
> [1] Zhu Z, Galkin M, Zhang Z, et al. Neural-symbolic models for logical queries on knowledge graphs. ICML, 2022: 27454-27478.
>
> [2] Zhu Z, Zhang Z, Xhonneux L P, et al. Neural bellman-ford networks: A general graph neural network framework for link prediction. NeurIPS, 2021: 29476-29490.
>
> [3] Zhang Y, Yao Q. Knowledge graph reasoning with relational digraph. ACM WWW 2022: 912-924.
>
> [4] Lee J, Chung C, Whang J J. InGram: Inductive knowledge graph embedding via relation graphs. ICML, 2023: 18796-18809.
>
> [5] Galkin M, Yuan X, Mostafa H, et al. Towards Foundation Models for Knowledge Graph Reasoning. ICLR 2024.

---

> ### Author Response · Authors · 2025-11-18
> **Response to Question 1**
>
> > **Question:** The paper assumes that the knowledge memory will contain relevant information for reasoning. However, in cases where the relevant entities or relations are not part of the memory, will the model's performance dramatically drops?
>
> **Response:** Thank you for your thorough review. **Your conjecture is fully aligned with the findings from our error analysis in Appendix J.7**! In Table 17 of our submitted manuscript, we divide the query triplets into two categories, **#Easy** and **#Hard**, depending on whether the correct target entity appears within the top-50 knowledge memory vectors. We then report our KRLM’s performance on these two subsets separately. The results show a sharp performance drop on **#Hard** queries. This bottleneck originates from the ULTRA component [1] inside KRLM (Eqs. 1 and 3 in our paper), which fails to effectively retrieve the correct candidate entity embeddings as memory storage, preventing KRLM from accessing the correct target during inference.
>
> **It is worth noting, however, that even when KRLM performs poorly on #Hard queries, the KRL attention layer still leverages the contextual semantics of entities in the knowledge memory to infer potential target candidates.** This mechanism helps improve the ranking of the ground-truth entity during the final prediction stage. A concrete example is illustrated in **Figure 7** of our paper: although ULTRA does not place the ground truth in the top-50 memory vectors, KRLM ultimately lifts the ground truth to rank 33.
>
> To further investigate how KRLM reasons about ground-truth entities that ULTRA fails to retrieve, we compute the MRR of ground-truth entities that appear inside (**#Easy**) or outside (**#Hard**) ULTRA’s top-50 retrieval results, and compare these with KRLM’s statistics in Table 17. As shown below, KRLM significantly outperforms ULTRA on all datasets for **#Easy** queries, and surpasses ULTRA on most **#Hard** queries as well.
>
> |Dataset|#Easy (KRLM)|#Easy (ULTRA)|#Hard (KRLM)|#Hard (ULTRA)|
> |-|-|-|-|-|
> |FB-V1|**0.658**|0.617|**0.010**|0.008|
> |FB-V2|**0.660**|0.606|**0.022**|0.007|
> |FB-V3|**0.674**|0.627|**0.011**|0.006|
> |FB-V4|**0.639**|0.599|**0.013**|0.007|
> |NELL-V1|**0.832**|0.785|**0.701**|0.000|
> |NELL-V2|**0.661**|0.598|**0.022**|0.007|
> |NELL-V3|**0.699**|0.600|**0.084**|0.008|
> |NELL-V4|**0.635**|0.586|**0.018**|0.008|
> |WN-V1|**0.827**|0.699|0.003|**0.006**|
> |WN-V2|**0.816**|0.750|**0.005**|0.003|
> |WN-V3|**0.650**|0.608|**0.006**|0.004|
> |WN-V4|**0.829**|0.626|**0.003**|**0.003**|
> |FB-25|**0.515**|0.471|**0.018**|0.007|
> |FB-50|**0.490**|0.444|**0.018**|0.006|
> |FB-75|**0.564**|0.497|**0.028**|0.007|
> |FB-100|**0.598**|0.533|**0.027**|0.008|
> |NL-0|**0.502**|0.441|**0.027**|0.006|
> |NL-25|**0.536**|0.521|**0.087**|0.008|
> |NL-50|**0.565**|0.513|**0.020**|0.008|
> |NL-75|**0.465**|0.454|**0.010**|**0.010**|
> |NL-100|**0.607**|0.539|**0.019**|0.008|
> |WK-25|**0.491**|0.430|**0.016**|0.007|
> |WK-50|**0.338**|0.312|**0.006**|0.005|
> |WK-75|**0.621**|0.567|**0.023**|0.007|
> |WK-100|**0.427**|0.399|**0.007**|0.005|
>
> [1] Galkin M, Yuan X, Mostafa H, et al. Towards Foundation Models for Knowledge Graph Reasoning. ICLR 2024.

---

### Author Response · Authors · 2025-12-01
**General Response to Reviewers, ACs, SACs, and PCs**

We sincerely thank all anonymous reviewers as well as the ACs, SACs, and PCs for their careful handling of our submission. We also appreciate the positive feedback from the reviewers on several aspects of our work, including **research motivation** (`Reviewers dW1V and LqgZ`), **model design** (`Reviewers WYHS, dW1V, ajSR, and LqgZ`), **theoretical and practical analysis** (`Reviewer WYHS`), and **the organization of comparative experiments** (`Reviewers WYHS, dW1V, and ajSR`). Based on the constructive suggestions provided by the reviewers, we have prepared point-by-point responses below. All revisions are highlighted in blue in the updated PDF. Here we highlight our major responses and revisions. We hope our responses can properly address the reviewers' concerns.

1. **Model Complexity Clarification** (`Reviewers WYHS, dW1V, ajSR, LqgZ`): We conducted a comprehensive analysis of the model computational complexity from the perspectives of training/inference cost, component-wise inference latency, and per-query token counts. **The results show that the training overhead of our model is not substantially higher than that of other LLM-based KGFMs. The inference latency of our method is significantly lower than that of LLM-based KGFMs that rely on explicit KG-path prompting**. The detailed analysis has been added to `Appendix G`.
2. **Regarding the adaptability to sparse KGs** (`Reviewers WYHS and dW1V`) and **limited computational resources** (`Reviewers WYHS, dW1V, and LqgZ`):
   - We conducted additional experiments on three sparse KG datasets derived from FB15k237 (FB15k237_10, FB15k237_20, and FB15k237_50). The results demonstrate that **KRLM effectively handles the lack of reasoning evidence in sparse KGs**. The corresponding analysis has been added to `Appendix J.4`.
   - **KRLM performs training-free, zero-shot inference on previously unseen KGs**. This substantially **reduces the computational cost required for model transfer**. A detailed discussion is included in `Appendix G.1`. Additional improvement directions have been incorporated into the future work section of the revised paper in `Appendix K`.
3. **Discussion of Component Details**
   - **Design details of the KRL instruction** (`Reviewer ajSR`): we added the discussions of instruction style, vocabulary, and length distribution in `Appendix B`. *These additions clarify the rationale and effectiveness of the proposed instruction format*.
   - **In-depth analysis of the core technical contributions (KRL attention layer with knowledge memory) along with theoretical explanations (`Reviewers dW1V and ajSR`) and error-case analyses (`Reviewer WYHS`):** `Appendix D` provides an analysis of the KRL attention layer from the perspective of attention coordination between the instruction context and KG entries, that is,  **the knowledge memory jointly influences the attention-weight scaling in contextual computation and the semantic correction of the aggregated representation**. `Appendix J.7` includes case studies and error analyses that present both quantitative and qualitative examinations of this module.
   - **Effectiveness analysis of the structure-aware predictor** (`Reviewer dW1V`): In `Appendix E`, we added a fine-grained effectiveness analysis of this component from the perspectives of the **entity-space constraint** and the **structural-context constraint**.
   - **Parameter initialization details** (`Reviewer dW1V`): `Appendix I` of the revised paper clarifies the initialization strategy for all model components. Specifically, **the LLM part uses the pretrained weights of Llama2-7b, and all remaining components are initialized using PyTorch’s built-in nn.Linear()**.
4. **Comparison of traditional KG embedding models** (`Reviewer WYHS`): The **supervised SOTA** results reported in `Tables 1, 12, and 13` denote the best-performing traditional KG embedding model on each dataset. Detailed descriptions of these models are provided in `Tables 8 and 9`.
5. **Quantitative evaluation of knowledge distortion** (`Reviewers dW1V and ajSR`): We introduce a new metric, **the distortion rate (DR)**, to assess the model’s robustness to knowledge distortion. The corresponding experiments demonstrate that **KRLM achieves strong resistance to distortion**. We have added this analysis to `Appendix J.5`.
6. **Qualitative examples for interpretable reasoning** (`Reviewer ajSR`): Existing KGFMs seldom explore interpretability in reasoning. We have incorporated potential solutions in `Appendix K`, where we employ an LLM-based rule generator combined with neuro-symbolic fuzzy reasoning to provide interpretability.
7. **Adaptability across different backbone LLMs** (`Reviewer LqgZ`): We have added the performance of our method under **Mistral-7B** and **Llama-3.1-8B** in `Appendix J.6`. **The experiments confirm that our approach maintains strong performance across different backbone LLMs**.

---

### Author Response · Authors · 2025-12-02
**Summary of Review Discussion Before the Leak Incident – 12291**

Dear Area Chair,

Due to the recent data leak incident on the platform and the subsequent paper reassignment, to assist in your assessment, we provide the status of the reviewers during the rebuttal period and a summary of the revisions we have made.

**We propose an LLM-based foundation model for knowledge graph reasoning, and the experiments on 25 inductive datasets demonstrate its strong generalization ability**. We received four reviews, including two scores of `6` (`Reviewers WYHS and ajSR`) and two scores of `4` (`Reviewers dW1V and LqgZ`). **During the initial rebuttal period, all reviewers provided positive feedback on our motivation, model design techniques, and breadth of comparative experiments (zero-shot and fine-tuning inference on 25 datasets)**. Their concerns primarily focused on several specific aspects, and we briefly summarize our responses to these points below:

1. **Model Complexity Clarification** (`Reviewers WYHS，dW1V，ajSR，LqgZ`)
   - We discuss the complexity of our method from four perspectives: **overall training cost**, **inference time**, **inference latency of each component**, and **instruction complexity**. The training overhead of our model is not substantially higher than that of other LLM-based KGFMs. The inference latency of our method is significantly lower than that of LLM-based KGFMs that rely on explicit KG-path prompting. The detailed analysis has been added to `Appendix G`.
   - We have added a potential solution to alleviate computational bottlenecks in `Appendix K`, which combines a lightweight LLM rule generator and neuro-symbolic fuzzy inference models to reduce computational costs.
2. **Adaptability analysis in more scenarios**
   - **Sparse KGs** (`Reviewer WYHS`): We evaluate zero-shot reasoning on three sparse KG datasets (FB237k\_10, FB237k\_20, and FB237k\_50). **KRLM achieves an average improvement of 3.35 \% in Hit@10 over existing KGFMs, which demonstrates its strong adaptability to sparse KGs**. The corresponding analysis has been added to `Appendix J.4`.
   - **Limited computational resources** (`Reviewer WYHS`): Our KRLM can perform **training-free, zero-shot inference on previously unseen KGs**. This substantially **reduces the computational cost required for model transfer**.
   - **Different backbone LLMs** (`Reviewer LqgZ`): We have added the performance of our method under **Mistral-7B** and **Llama-3.1-8B** in `Appendix J.6`. The experiments confirm that our approach maintains strong performance across different backbone LLMs.
3. **Theoretical analysis of the effectiveness of each component**
   - **Core technological contribution** (`Reviewers dW1V, ajSR`): Our core contribution lies in proposing **the KRL attention that coordinates KG entries (memory) and LLM intrinsic knowledge**. `Appendix D` provides an analysis of the KRL attention from the perspective of the extension of the self-attention mechanism, that is, **the knowledge memory jointly influences the attention-weight scaling in the instruction context and the semantic correction of the aggregated representation**.
   - We have added modeling principle analysis for **the KRL instruction tokenizer** (`Reviewers dW1V, ajSR`) and **the next-entity predictor** (`Reviewer dW1V`) in `Appendix B` and `Appendix E`, respectively, to demonstrate the effectiveness of each component.
4. **Comparison of traditional KG embedding models** (`Reviewer WYHS`)
   - Since our experiments involve 28 datasets, `Tables 1, 12, and 13` report the **supervised SOTA** results, which denote the best-performing traditional KG embedding model on each dataset. Detailed descriptions of these models are provided in `Tables 8 and 9`.
5. **Direct quantitative evaluation of the motivation for knowledge distortion** (`Reviewers dW1V and ajSR`)
   - We introduce a new metric, **the distortion rate (DR)**, to assess the model’s robustness when the KG reasoning evidence is incomplete. The corresponding experiments demonstrate that **KRLM achieves strong resistance to distortion**. We have added this analysis to `Appendix J.5`.
6. **Case study**
   - **Error analysis** (`Reviewer WYHS`): The visualization case of `Appendix J.7` in our **original submission** shows that if the knowledge memory does not hit the candidate entities, the reasoning of our model will decrease, **which is consistent with the conjecture of `Reviewer WYHS`**.
   - **Interpretable case analysis** (`Reviewer ajSR`): Existing KGFMs seldom explore interpretability in reasoning. We have incorporated potential solutions in `Appendix K`, where **we employ an LLM-based rule generator combined with neuro-symbolic fuzzy reasoning to provide interpretability**.

We appreciate your time and careful consideration of our submission. We hope this summary assists you in navigating our rebuttal situation. The revised PDF containing all these improvements has been uploaded.

Best regards,

on behalf of the authors of Paper 12291

---

### Meta-Review · Area_Chair_Uzcw · 2026-01-22

**Summary:**

This paper proposes an approach to inductive KG reasoning with LLMs that aims to better align LLM knowledge with KG reasoning. Reviewers initially raised concerns about efficiency, sparse KG performance, clarity, and novelty. The rebuttal thoroughly addressed these points with new experiments and clearer explanations. The framework is of interest to the KG reasoning community, and remaining issues are about framing rather than technical soundness.

**Reviewer Concerns:**

All addressed, in my opinion. The rebuttal was thorough.

**Reviewer Scores:**

I believe the reviewers concerned with the lack of a quantitative definition for knowlege distortion would have increased their scores after the authors introduced metrics and provided empirical results quantifying this phenomenon. Similarly, missing or unclear methodological details were thoroughly addressed, and more experimental results were provided; both would have increased scores.

---

### Decision · Program_Chairs · 2026-01-26

Accept (Poster)